# Learning in Observable POMDPs,
# without Computationally Intractable Oracles

**Noah Golowich**
MIT
nzg@mit.edu

**Ankur Moitra**
MIT
moitra@mit.edu

**Dhruv Rohatgi**
MIT
drohatgi@mit.edu

## Abstract

Much of reinforcement learning theory is built on top of oracles that are computationally hard to implement. Specifically for learning near-optimal policies in Partially Observable Markov Decision Processes (POMDPs), existing algorithms either need to make strong assumptions about the model dynamics (e.g. deterministic transitions) or assume access to an oracle for solving a hard optimistic planning or estimation problem as a subroutine. In this work we develop the first oracle-free learning algorithm for POMDPs under reasonable assumptions. Specifically, we give a quasipolynomial-time end-to-end algorithm for learning in "observable" POMDPs, where observability is the assumption that well-separated distributions over states induce well-separated distributions over observations. Our techniques circumvent the more traditional approach of using the principle of optimism under uncertainty to promote exploration, and instead give a novel application of barycentric spanners to constructing policy covers.

## 1 Introduction

Markov Decision Processes (MDPs) are a ubiquitous model in reinforcement learning that aim to capture sequential decision-making problems in a variety of applications spanning robotics to healthcare. However, modelling a problem with an MDP makes the often-unrealistic assumption that the agent has perfect knowledge about the state of the world. Partially Observable Markov Decision Processes (POMDPs) are a broad generalization of MDPs which capture an agent's inherent uncertainty about the state: while there is still an underlying state that updates according to the agent's actions, the agent never directly observes the state, but instead receives samples from a state-dependent observation distribution. The greater generality afforded by partial observability is crucial to applications in game theory [BS18], healthcare [Hau00, HF00b], market design [WME+22], and robotics [CKK96].

Unfortunately, this greater generality comes with steep statistical and computational costs. There are well-known statistical lower bounds [JKKL20, KAL16], which show that in the worst case, it is statistically intractable to find a near-optimal policy for a POMDP given the ability to play policies on it (the *learning* problem), even given unlimited computation. Furthermore, there are worst-case computational lower bounds [PT87, Lit94, BDRS96, LGM01, VLB12], which establish that it is computationally intractable to find a near-optimal policy even when given the exact parameters of the model (the substantially simpler *planning* problem).

Nevertheless there is a sizeable literature devoted to overcoming the *statistical* intractability of the learning problem by restricting to natural subclasses of POMDPs [KAL16, GDB16, ALA16, JKKL20, XCGZ21, KECM21a, KECM21b, LCSJ22]. There are far fewer works attempting to overcome *computational* intractability, and all make severe restrictions on either the model dynamics [JKKL20, KAL16] or the structure of the uncertainty [BDRS96, KECM21a]. The standard practice is to simply sidestep computational issues by assuming access to strong oracles such as ones that

36th Conference on Neural Information Processing Systems (NeurIPS 2022).

solve *Optimistic Planning* (given a constrained, non-convex set of POMDPs, find the maximum value achievable by any policy on any POMDP in the set) [JKKL20] or *Optimistic Maximum Likelihood Estimation* (given a set of action/observation sample trajectories, find a POMDP which obtains maximum value conditioned on approximately maximizing the likelihood of seeing those trajectories) [LCSJ22]. Unsurprisingly, these oracles are computationally intractable to implement. Is there any hope for giving computationally efficient, oracle-free learning algorithms for POMDPs under reasonable assumptions? The naïve approach would require exponential time, and thus even a quasi-polynomial time algorithm would represent a dramatic improvement.

A necessary first step towards solving the learning problem is having a computationally efficient planning algorithm. Few such algorithms have provable guarantees under reasonable model assumptions, but recently it was shown [GMR22] that there is a quasipolynomial-time planning algorithm for POMDPs which satisfy an *observability* assumption. Let $H \in \mathbb{N}$ be the horizon length of the POMDP, and for each state $s$ and step $h \in [H]$, let $\mathbb{O}_h(\cdot|s)$ denote the observation distribution at state $s$ and step $h$. Then observability is defined as follows:

**Assumption 1.1** ([EDKM07, GMR22])**.** *Let $\gamma > 0$. For $h \in [H]$, let $\mathbb{O}_h$ be the matrix with columns $\mathbb{O}_h(\cdot|s)$, indexed by states $s$. We say that the matrix $\mathbb{O}_h$ satisfies $\gamma$-observability if for each $h$, for any distributions $b, b'$ over states, $\|\mathbb{O}_h b - \mathbb{O}_h b'\|_1 \geq \gamma \|b - b'\|_1$. A POMDP satisfies* (one-step) $\gamma$-observability *if all $H$ of its observation matrices do.*

Compared to previous assumptions enabling computationally efficient planning, observability is much milder because it makes no assumptions about the dynamics of the POMDP, and it allows for natural observation models such as noisy or lossy sensors [GMR22]. It is known that statistically efficient learning is possible under somewhat weaker assumptions than observability [JKKL20], however these works rely on solving a planning problem that is computationally intractable. This raises the question: *Is observability enough to remedy both the computational and statistical woes of learning POMDPs?* In particular, can we get not only efficient planning but efficient learning too?

**Overview of results.** This work provides an affirmative answer to the questions above: we give an algorithm (`BaSeCAMP`, Algorithm 3) with quasi-polynomial time (and sample) complexity for learning near-optimal policies in observable POMDPs – see Theorem 3.1. While this falls just short of polynomial time, it turns out to be optimal in the sense that even for observable POMDPs there is a quasi-polynomial time lower bound for the (simpler) planning problem under standard complexity assumptions [GMR22]. A key innovation of our approach is an alternative technique to encourage exploration: whereas essentially all previous approaches for partially observable RL used the principle of *optimism under uncertainty* to encourage the algorithm to visit states [JKKL20, LCSJ22], we introduce a new framework based on the use of *barycentric spanners* [AK08] and *policy covers* [DKJ+19]. While each of these tools has previously been used in the broader RL literature to promote exploration (e.g. [LS17, FKQR21, DKJ+19, AHKS20, DGZ22, JKSY20, MHKL20]), they have not been used specifically in the study of POMDPs with imperfect observations,[1] and indeed our usage of them differs substantially from past instances.

The starting point for our approach is a result of [GMR22] (restated as Theorem 2.1) which implies that the dynamics of an observable POMDP $\mathcal{P}$ may be approximated by those of a *Markov decision process* $\mathcal{M}$ with a quasi-polynomial number of states. If we knew the transitions of $\mathcal{M}$, then we could simply use dynamic programming to find an optimal policy for $\mathcal{M}$, which would be guaranteed to be a near-optimal policy for $\mathcal{P}$. Instead, we must *learn* the transitions of $\mathcal{M}$, for which it is necessary to explore all (reachable) states of the underlying POMDP $\mathcal{P}$. A naïve approach to encourage exploration is to learn the transitions of $\mathcal{P}$ via forwards induction on the layer $h$, using, at each step $h$, our knowledge of the learned transitions at steps prior to $h$ to find a policy which visits each reachable state at step $h$. Such an approach would lead to a *policy cover*, namely a collection of policies which visits all reachable latent states.

A major problem with this approach is that the latent states are not observed: instead, we only see observations. Hence a natural approach might be to choose policies which lead to all possible *observations* at each step $h$. This approach is clearly insufficient, since, e.g., a single state could emit a uniform distribution over observations. Thus we instead compute the following stronger concept: for each step $h$, we consider the set $\mathcal{X}$ of all possible distributions over observations at

---

[1] Policy covers have been used in the special case of *block MDPs* [DKJ+19, MHKL20], namely where different states produce disjoint observations.

step $h$ under any general policy, and attempt to find a *barycentric spanner* of $\mathcal{X}$, namely a small subset $\mathcal{X}' \subset \mathcal{X}$ so that all other distributions in $\mathcal{X}$ can be expressed as a linear combination of elements of $\mathcal{X}'$ with bounded coefficients. By playing a policy which realizes each distribution in such a barycentric spanner $\mathcal{X}'$, we are able to explore all reachable latent states, despite having no knowledge about which states we are exploring. This discussion omits a key technical aspect of the proof, which is the fact that we can only compute a barycentric spanner for a set $\mathcal{X}$ corresponding to an empirical estimate $\widehat{\mathcal{M}}$ of $\mathcal{M}$. A key innovation in our proof is a technique to dynamically use such barycentric spanners, even when $\widehat{\mathcal{M}}$ is inaccurate, to improve the quality of the estimate $\widehat{\mathcal{M}}$. We remark that a similar dynamic usage of barycentric spanners appeared in [FKQR21]; we discuss in the appendix why the approach of [FKQR21], as well as related approaches involving nonstationary MDPs [WL21, WDZ22, WYDW21], cannot be applied here.

Taking a step back, few models in reinforcement learning (beyond tabular or linear MDPs) admit computationally efficient end-to-end learning algorithms – indeed, our main contribution is a way to circumvent the daunting task of implementing any of the various constrained optimistic planning oracles assumed in previous optimism-based approaches. We hope that our techniques may be useful in other contexts for avoiding computational intractability without resorting to oracles.

## 2 Preliminaries

For sets $\mathcal{T}, \mathcal{Q}$, let $\mathcal{Q}^{\mathcal{T}}$ denote the set of mappings from $\mathcal{T} \to \mathcal{Q}$. Accordingly, we will identify $\mathbb{R}^{\mathcal{T}}$ with $|\mathcal{T}|$-dimensional Euclidean space, and let $\Delta(\mathcal{T}) \subset \mathbb{R}^{\mathcal{T}}$ consist of distributions on $\mathcal{T}$. For $d \in \mathbb{N}$ and a vector $\mathbf{v} \in \mathbb{R}^d$, we denote its components by $\mathbf{v}(1), \ldots, \mathbf{v}(d)$. For integers $a \leq b$, we abbreviate a sequence $(x_a, x_{a+1}, \ldots, x_b)$ by $x_{a:b}$. If $a > b$, then we let $x_{a:b}$ denote the empty sequence. *Sometimes we refer to negative indices of a sequence $x_{1:n}$: in such cases the elements with negative indices may be taken to be aribtrary, as they will never affect the value of the expression. See Appendix B.1 for clarification.* For $x \in \mathbb{R}$, we write $[x]_+ = \max\{x, 0\}$, and $[x]_- = -\min\{x, 0\}$. For sets $\mathcal{S}, \mathcal{T}$, the notation $\mathcal{S} \subset \mathcal{T}$ allows for the possibility that $\mathcal{S} = \mathcal{T}$.

### 2.1 Background on POMDPs

In this paper we address the problem of learning finite-horizon *partially observable Markov decision processes (POMDPs)*. Formally, a POMDP $\mathcal{P}$ is a tuple $\mathcal{P} = (H, \mathcal{S}, \mathcal{A}, \mathcal{O}, b_1, R, \mathbb{T}, \mathbb{O})$, where: $H \in \mathbb{N}$ is a positive integer denoting the *horizon* length; $\mathcal{S}$ is a finite set of *states* of size $S := |\mathcal{S}|$; $\mathcal{A}$ is a finite set of *actions* of size $A := |\mathcal{A}|$; $\mathcal{O}$ is a finite set of *observations* of size $O := |\mathcal{O}|$; $b_1$ is the initial distribution over states; and $R, \mathbb{T}, \mathbb{O}$ are given as follows. First, $R = (R_1, \ldots, R_H)$ denotes a tuple of *reward functions*, where, for $h \in [H]$, $R_h : \mathcal{O} \to [0, 1]$ gives the reward received as a function of the observation at step $h$. (It is customary in the literatue [JKKL20, LCSJ22] to define the rewards as being a function of the observations as opposed to being observed by the algorithm as separate information.) Second, $\mathbb{T} = (\mathbb{T}_1, \ldots, \mathbb{T}_H)$ is a tuple of *transition kernels*, where, for $h \in [H], s, s' \in \mathcal{S}, a \in \mathcal{A}$, $\mathbb{T}_h(s'|s, a)$ denotes the probability of transitioning from $s$ to $s'$ at step $h$ when action $a$ is taken. For each $a \in \mathcal{A}$, we will write $\mathbb{T}_h(a) \in \mathbb{R}^{S \times S}$ to denote the matrix with $\mathbb{T}_h(a)_{s, s'} = \mathbb{T}_h(s|s', a)$. Third, $\mathbb{O} = (\mathbb{O}_1, \ldots, \mathbb{O}_H)$ is a tuple of *observation matrices*, where for $h \in [H], s \in \mathcal{S}, o \in \mathcal{O}$, $(\mathbb{O}_h)_{o,s}$, also written as $\mathbb{O}_h(o|s)$, denotes the probability observing $o$ while in state $s$ at step $h$. Thus $\mathbb{O}_h \in \mathbb{R}^{O \times S}$ for each $h$. Sometimes, for disambiguation, we will refer to the states $\mathcal{S}$ as the *latent states* of the POMDP $\mathcal{P}$.

The interaction (namely, a single *episode*) with $\mathcal{P}$ proceeds as follows: initially a state $s_1 \sim b_1$ is drawn from the initial state distribution. At each step $1 \leq h < H$, an action $a_h \in \mathcal{A}$ is chosen (as a function of previous observations and actions taken), $\mathcal{P}$ transitions to a new state $s_{h+1} \sim \mathbb{T}_h(\cdot|s_h, a_h)$, a new observation is observed, $o_{h+1} \sim \mathbb{O}_{h+1}(\cdot|s_{h+1})$, and a reward of $R_{h+1}(o_{h+1})$ is received (and observed). We emphasize that the underlying states $s_{1:H}$ are never observed directly. As a matter of convention, we assume that no observation is observed at step $h = 1$; thus the first observation is $o_2$.

### 2.2 Policies, value functions

A *deterministic policy* $\sigma$ is a tuple $\sigma = (\sigma_1, \ldots, \sigma_H)$, where $\sigma_h : \mathcal{A}^{h-1} \times \mathcal{O}^{h-1} \to \mathcal{A}$ is a mapping from *histories* up to step $h$, namely tuples $(a_{1:h-1}, o_{2:h})$, to actions. We will denote the collection of histories up to step $h$ by $\mathscr{H}_h := \mathcal{A}^{h-1} \times \mathcal{O}^{h-1}$ and the set of deterministic policies by $\Pi^{\text{det}}$,

meaning that $\Pi^{\mathrm{det}} = \prod_{h=1}^{H} \mathcal{A}^{\mathscr{H}_h}$. A *general policy* $\pi$ is a distribution over deterministic policies; the set of general policies is denoted by $\Pi^{\mathrm{gen}} := \Delta(\prod_{h=1}^{H} \mathcal{A}^{\mathscr{H}_h})$. Given a general policy $\pi \in \Pi^{\mathrm{gen}}$, we denote by $\sigma \sim \pi$ the draw of a deterministic policy from the distribution $\pi$; to execute a general policy $\pi$, a sample $\sigma \sim \pi$ is first drawn and then followed for an episode of the POMDP. For a general policy $\pi$ and some event $\mathcal{E}$, write $\mathbb{P}_{a_{1:H-1}, o_{2:H}, s_{1:H} \sim \pi}^{\mathcal{P}}(\mathcal{E})$ to denote the probability of $\mathcal{E}$ when $s_{1:H}, a_{1:H-1}, o_{2:H}$ is drawn from a trajectory following policy $\pi$ for the POMDP $\mathcal{P}$. At times we will compress notation in the subscript, e.g., write $\mathbb{P}_{\pi}^{\mathcal{P}}(\mathcal{E})$ if the definition of $s_{1:H}, a_{1:H-1}, o_{2:H}$ is evident. In similar spirit, we will write $\mathbb{E}_{a_{1:H-1}, o_{2:H}, s_{1:H} \sim \pi}^{\mathcal{P}}[\cdot]$ to denote expectations.

Given a general policy $\pi \in \Pi^{\mathrm{gen}}$, define the *value function* for $\pi$ at step 1 by $V_1^{\pi, \mathcal{P}}(\emptyset) = \mathbb{E}_{o_{1:H} \sim \pi}^{\mathcal{P}}\left[\sum_{h=2}^{H} R_h(o_h)\right]$, namely as the expected reward received by following $\pi$.

Our objective is to find a policy $\pi$ which maximizes $V_1^{\pi, \mathcal{P}}(\emptyset)$, in the *PAC-RL model* [KS02]: in particular, the algorithm does not have access to the transition kernel, reward function, or observation matrices of $\mathcal{P}$, but can repeatedly choose a general policy $\pi$ and observe the following data from a single trajectory drawn according to $\pi$: $(a_1, o_2, R_2(o_2), a_2, \ldots, a_{H-1}, o_H, R_H(o_H))$. The challenge is to choose such policies $\pi$ which can sufficiently explore the environment.

Finally, we remark that *Markov decision processes* (MDPs) are a special case of POMDPs where $\mathcal{O} = \mathcal{S}$ and $\mathbb{O}_h(o|s) = \mathbb{1}[o = s]$ for all $h \in [H]$, $o, s \in \mathcal{S}$. For the MDPs we will consider, the initial state distribution will be left unspecified (indeed, the optimal policy of an MDP does not depend on the initial state distribution). Thus, we consider MDPs $\mathcal{M}$ described by a tuple $\mathcal{M} = (H, \mathcal{S}, \mathcal{A}, R, \mathbb{T})$.

## 2.3 Belief contraction

A prerequisite for a computationally efficient *learning* algorithm in observable POMDPs is a computationally efficient *planning* algorithm, i.e. an algorithm to find an approximately optimal policy when the POMDP is known. Recent work [GMR22] obtains such a planning algorithm taking quasipolynomial time; we now introduce the key tools used in [GMR22], which are used in our algorithm as well.

Consider a POMDP $\mathcal{P} = (H, \mathcal{S}, \mathcal{A}, \mathcal{O}, b_1, R, \mathbb{T}, \mathbb{O})$. Given some $h \in [H]$ and a history $(a_{1:h-1}, o_{2:h}) \in \mathscr{H}_h$, the *belief state* $\mathbf{b}_h^{\mathcal{P}}(a_{1:h-1}, o_{2:h}) \in \Delta(\mathcal{S})$ is given by the distribution of the state $s_h$ conditioned on taking actions $a_{1:h-1}$ and observing $o_{2:h}$ in the first $h$ steps. Formally, the belief state is defined inductively as follows: $\mathbf{b}_1^{\mathcal{P}}(\emptyset) = b_1$, and for $2 \le h \le H$ and any $(a_{1:h-1}, o_{2:h}) \in \mathcal{H}_h$,

$$\mathbf{b}_h^{\mathcal{P}}(a_{1:h-1}, o_{2:h}) = U_{h-1}^{\mathcal{P}}(\mathbf{b}_{h-1}^{\mathcal{P}}(a_{1:h-2}, o_{2:h-1}); a_{h-1}, o_h),$$

where for $\mathbf{b} \in \Delta(\mathcal{S}), a \in \mathcal{A}, o \in \mathcal{O}, U_h^{\mathcal{P}}(\mathbf{b}; a, o) \in \Delta(\mathcal{S})$ is the distribution defined by

$$U_h^{\mathcal{P}}(\mathbf{b}; a, o)(s) := \frac{\mathbb{O}_{h+1}(o|s) \cdot \sum_{s' \in \mathcal{S}} \mathbf{b}(s') \cdot \mathbb{T}_h(s|s', a)}{\sum_{x \in \mathcal{S}} \mathbb{O}_{h+1}(o|x) \sum_{s' \in \mathcal{S}} \mathbf{b}(s') \cdot \mathbb{T}_h(x|s', a)}.$$

We call $U_h^{\mathcal{P}}$ the *belief update operator*. The belief state $\mathbf{b}_h^{\mathcal{P}}(a_{1:h-1}, o_{2:h})$ is a sufficient statistic for the sequence of future actions and observations under any deterministic policy. In particular, the optimal policy can be expressed as a function of the belief state, rather than the entire history. Thus, a natural approach to plan a near-optimal policy is to find a small set $\mathscr{B} \subset \Delta(\mathcal{S})$ of distributions over states such that each possible belief state $\mathbf{b}_h^{\mathcal{P}}(a_{1:h-1}, o_{2:h})$ is close to some element of $\mathscr{B}$. Unfortunately, this is not possible, even in observable POMDPs [GMR22, Example D.2]. The main result of [GMR22] circumvents this issue by showing that there is a subset $\mathscr{B} \subset \Delta(\mathcal{S})$ of quasipolynomial size (depending on $\mathcal{P}$) so that $\mathbf{b}_h^{\mathcal{P}}(a_{1:h-1}, o_{2:h})$ is close to some element of $\mathscr{B}$ *in expectation* under any given policy. To state the result of [GMR22], we need to introduce *approximate belief states*:

**Definition 2.1** (Approximate belief state). Fix a POMDP $\mathcal{P} = (H, \mathcal{S}, \mathcal{A}, \mathcal{O}, b_1, R, \mathbb{T}, \mathbb{O})$. For any distribution $\mathscr{D} \in \Delta(\mathcal{S})$, as well as any choices of $1 \le h \le H$ and $L \ge 0$, the approximate belief state $\mathbf{b}_h^{\mathrm{apx}, \mathcal{P}}(a_{h-L:h-1}, o_{h-L+1:h}; \mathscr{D})$ is defined as follows, via induction on $L$: in the case that $L = 0$, then we define

$$\mathbf{b}_h^{\mathrm{apx}, \mathcal{P}}(\emptyset; \mathscr{D}) := \begin{cases} b_1 & : h = 1 \\ \mathscr{D} & : h > 1, \end{cases}$$

and for the case that $L > 0$, define, for $h > L$,

$$\mathbf{b}_h^{\mathrm{apx},\mathcal{P}}(a_{h-L:h-1}, o_{h-L+1:h}; \mathscr{D}) := U_{h-1}^{\mathcal{P}}(\mathbf{b}_{h-1}^{\mathrm{apx},\mathcal{P}}(a_{h-L:h-2}, o_{h-L+1:h-1}; \mathscr{D}); a_{h-1}, o_h).$$

We extend the above definition to the case that $h \leq L$ by defining $\mathbf{b}_h^{\mathrm{apx},\mathcal{P}}(a_{h-L:h-1}, o_{h-L+1:h}; \mathscr{D}) := \mathbf{b}_h^{\mathrm{apx},\mathcal{P}}(a_{\max\{1,h-L\}:h-1}, o_{\max\{2,h-L+1\}:h}; \mathscr{D})$. In words, the approximate belief state $\mathbf{b}_h^{\mathrm{apx},\mathcal{P}}(a_{h-L:h-1}, o_{h-L+1:h}; \mathscr{D})$ is obtained by applying the belief update operator starting from the distribution $\mathscr{D}$ at step $h - L$, if $h - L > 1$ (and otherwise, starting from $b_1$, at step 1). At times, we will drop the superscript $\mathcal{P}$ from the above definitions and write $\mathbf{b}_h, \mathbf{b}_h^{\mathrm{apx}}$.

The main technical result of [GMR22], stated as Theorem 2.1 below (with slight differences, see Appendix B), proves that if the POMDP $\mathcal{P}$ is $\gamma$-observable for some $\gamma > 0$, then for a wide range of distributions $\mathscr{D}$, for sufficiently large $L$, the approximate belief state $\mathbf{b}_h^{\mathrm{apx},\mathcal{P}}(a_{h-L:h-1}, o_{h-L+1:h}; \mathscr{D})$ will be close to (i.e., "contract to") the true belief state $\mathbf{b}_h^{\mathcal{P}}(a_{1:h-1}, o_{2:h})$.

**Theorem 2.1** (Belief contraction; Theorems 4.1 and 4.7 of [GMR22]). *Consider any $\gamma$-observable POMDP $\mathcal{P}$, any $\epsilon > 0$ and $L \in \mathbb{N}$ so that $L \geq C \cdot \min\left\{ \frac{\log(1/(\epsilon\phi))\log(\log(1/\phi)/\epsilon)}{\gamma^2}, \frac{\log(1/(\epsilon\phi))}{\gamma^4} \right\}$. Fix any $\pi \in \Pi^{\mathrm{gen}}$, and suppose that $\mathscr{D} \in \Delta^{\mathcal{S}}$ satisfies $\frac{\mathbf{b}_h^{\mathcal{P}}(a_{1:h-L-1}, o_{2:h-L})(s)}{\mathscr{D}(s)} \leq \frac{1}{\phi}$ for all $(a_{h-1}, o_{2:h})$. Then*

$$\mathbb{E}_{(a_{1:h-1}, o_{2:h}) \sim \pi}^{\mathcal{P}} \left\| \mathbf{b}_h^{\mathcal{P}}(a_{1:h-1}, o_{2:h}) - \mathbf{b}_h^{\mathrm{apx},\mathcal{P}}(a_{h-L:h-1}, o_{h-L+1:h}; \mathscr{D}) \right\|_1 \leq \epsilon.$$

### 2.4 Visitation distributions

For a POMDP $\mathcal{P} = (H, \mathcal{S}, \mathcal{A}, \mathcal{O}, b_1, R, \mathbb{T}, \mathbb{O})$, policy $\pi \in \Pi^{\mathrm{gen}}$, and step $h \in [H]$, the (latent) *state visitation distribution* at step $h$ is $d_{\mathsf{S},h}^{\mathcal{P},\pi} \in \Delta(\mathcal{S})$ defined by $d_{\mathsf{S},h}^{\mathcal{P},\pi}(s) := \mathbb{P}_{s_{1:H} \sim \pi}^{\mathcal{P}}(s_h = s)$, and the *observation visitation distribution* at step $h$ is $d_{\mathsf{O},h}^{\mathcal{P},\pi} \in \Delta(\mathcal{O})$ defined by $d_{\mathsf{O},h}^{\mathcal{P},\pi} := \mathbb{P}_{o_{1:H} \sim \pi}^{\mathcal{P}}(o_h = \cdot) = \mathbb{O}_h \cdot d_{\mathsf{S},h}^{\mathcal{P},\pi}$. As will be discussed in Section 4.1, Theorem 2.1 implies that the transitions of the POMDP $\mathcal{P}$ can be approximated by those of an MDP $\mathcal{M}$ whose states consist of $L$-tuples of actions and observations. Thus we will often deal with such MDPs $\mathcal{M}$ of the form $\mathcal{M} = (H, \mathcal{Z}, \mathcal{A}, R, \mathbb{T})$ where the set of states has the product structure $\mathcal{Z} = \mathcal{A}^L \times \mathcal{O}^L$. We then define $\mathsf{o} : \mathcal{Z} \to \mathcal{O}$ by $\mathsf{o}(a_{1:L}, o_{1:L}) = o_L$. For such MDPs, we define the observation visitation distributions by $d_{\mathsf{O},h}^{\mathcal{M},\pi}(o) := \mathbb{P}_{z_{1:H} \sim \pi}^{\mathcal{M}}(\mathsf{o}(z_h) = o)$. Finally, for $o \in \mathcal{O}$, we let $e_o \in \mathbb{R}^{\mathcal{O}}$ denote the $o$th unit vector; thus, for instance, we have $d_{\mathsf{O},h}^{\mathcal{P},\pi}(o) = \langle e_o, d_{\mathsf{O},h}^{\mathcal{P},\pi} \rangle$.

## 3 Main result: learning observable POMDPs in quasipolynomial time

Theorem 3.1 below states our main guarantee for BaSeCAMP (**Ba**rycentric **S**pann**e**r policy **C**over with **A**pproximate **M**DP; Algorithm 3).

**Theorem 3.1.** *Given any $\alpha, \beta, \gamma > 0$ and $\gamma$-observable POMDP $\mathcal{P}$, BaSeCAMP with parameter settings as described in Section C.1 outputs a policy which is $\alpha$-suboptimal with probability at least $1 - \beta$, using time and sample complexity bounded by $(OA)^{CL} \log(1/\beta)$, where $C > 1$ is a constant and $L = \min\left\{ \frac{\log(HSO/(\alpha\gamma))}{\gamma^4}, \frac{\log^2(HSO/(\alpha\gamma))}{\gamma^2} \right\}$.*

It is natural to ask whether the complexity guarantee of Theorem 3.1 can be improved further. [GMR22, Theorem 6.4] shows that under the Exponential Time Hypothesis, there is no algorithm for planning in $\gamma$-observable POMDPs which runs in time $(SAHO)^{o(\log(SAHO/\alpha)/\gamma)}$ and produces $\alpha$-suboptimal policies, *even if the POMDP is known*. Thus, up to polynomial factors in the exponent, Theorem 3.1 is optimal. It is plausible, however, that there could be an algorithm which runs in quasipolynomial time yet only needs *polynomially* many samples; we leave this question for future work.

# 4 Algorithm description

## 4.1 Approximating $\mathcal{P}$ with an MDP.

A key consequence of observability is that, by the belief contraction result of Theorem 2.1, the POMDP $\mathcal{P}$ is well-approximated by an MDP $\mathcal{M}$ of quasi-polynomial size. In more detail, we will apply Theorem 2.1 with $\phi = 1/S$, $\mathscr{D} = \text{Unif}(\mathcal{S})$, and some $L = \text{poly}(\log(S/\epsilon)/\gamma)$ sufficiently large so as to satisfy the requirement of the theorem statement. The MDP $\mathcal{M}$ has state space $\mathcal{Z} := \mathcal{A}^L \times \mathcal{O}^L$, horizon $H$, action space $\mathcal{A}$, and transitions $\mathbb{P}_h^{\mathcal{M}}(\cdot|z_h, a_h)$ which are defined via a belief update on the approximate belief state $\mathbf{b}_h^{\text{apx},\mathcal{P}}(z_h; \text{Unif}(\mathcal{S}))$[2]: in particular, for a state $z_h = (a_{h-L:h-1}, o_{h-L+1:h}) \in \mathcal{Z}$ of $\mathcal{M}$, action $a_h$, and subsequent observation $o_{h+1} \in \mathcal{O}$, define

$$\mathbb{P}_h^{\mathcal{M}}((a_{h-L+1:h}, o_{h-L+2:h+1})|z_h, a_h) := e_{o_{h+1}}^{\top} \cdot \mathbb{O}_{h+1}^{\mathcal{P}} \cdot \mathbb{T}_h^{\mathcal{P}}(a_h) \cdot \mathbf{b}_h^{\text{apx},\mathcal{P}}(z_h; \text{Unif}(\mathcal{S})). \quad (1)$$

The above definition should be compared with the probability of observing $o_{h+1}$ given history $(a_{1:h}, o_{2:h})$ and policy $\pi$ when interacting with the POMDP $\mathcal{P}$, which is

$$\mathbb{P}_{o_{h+1} \sim \pi}^{\mathcal{P}}(o_{h+1}|a_{1:h}, o_{2:h}) = e_{o_{h+1}}^{\top} \cdot \mathbb{O}_{h+1}^{\mathcal{P}} \cdot \mathbb{T}_h^{\mathcal{P}}(a_h) \cdot \mathbf{b}_h^{\mathcal{P}}(a_{1:h-1}, o_{2:h}). \quad (2)$$

Theorem 2.1 gives that $\left\| \mathbf{b}_h^{\mathcal{P}}(a_{1:h-1}, o_{2:h}) - \mathbf{b}_h^{\text{apx},\mathcal{P}}(a_{h-L:h-1}, o_{h-L+1:h}; \text{Unif}(\mathcal{S})) \right\|_1$ is small in expectation under any general policy $\pi$, which, using (1) and (2), gives that, for all $\pi \in \Pi^{\text{gen}}$ and $h \in [H]$,

$$\mathbb{E}_{a_{1:h}, o_{2:h} \sim \pi} \sum_{o_{h+1} \in \mathcal{O}} \left| \mathbb{P}_h^{\mathcal{M}}(o_{h+1}|z_h, a_h) - \mathbb{P}_h^{\mathcal{P}}(o_{h+1}|a_{1:h}, o_{2:h}) \right| \leq \epsilon. \quad (3)$$

(Above we have written $z_h = (a_{h-L:h-1}, o_{h-L+1:h})$ and, via abuse of notation, $\mathbb{P}_h^{\mathcal{M}}(o_{h+1}|z_h, a_h)$ in place of $\mathbb{P}_h^{\mathcal{M}}((a_{h-L+1:h}, o_{h-L+2:h+1})|z_h, a_h)$.) The inequality (3) establishes that the dynamics of $\mathcal{P}$ under any policy may be approximated by those of the MDP $\mathcal{M}$. Crucially, this implies that there exists a deterministic Markov policy for $\mathcal{M}$ which is near-optimal among general policies for $\mathcal{P}$; the set of such Markov policies for $\mathcal{M}$ is denoted by $\Pi_{\mathsf{Z}}^{\text{markov}}$. Because of the Markovian structure of $\mathcal{M}$, such a policy can be found in time polynomial in the size of $\mathcal{M}$ (which is quasi-polynomial in the underlying problem parameters), if $\mathcal{M}$ is known. Of course, $\mathcal{M}$ is not known.

**Approximately learning the MDP $\mathcal{M}$.** These observations suggest the following model-based approach of trying to learn the transitions of $\mathcal{M}$. Suppose that we know a sequence of general policies $\pi^1, \ldots, \pi^H$ (abbreviated as $\pi^{1:H}$) so that for each $h$, $\pi^h$ visits a uniformly random state of $\mathcal{P}$ at step $h - L$ (i.e. $d_{\mathsf{S},h-L}^{\mathcal{P},\pi^h} = \text{Unif}(\mathcal{S})$). Then we can estimate the transitions of $\mathcal{M}$ as follows: play $\pi^h$ for $h-L-1$ steps and then play $L+1$ random actions, generating a trajectory $(a_{1:h}, o_{2:h+1})$. Conditioned on $z_h = (a_{h-L:h-1}, o_{h-L+1:h})$ and final action $a_h$, the last observation of this sample trajectory, $o_{h+1}$, would give an unbiased draw from the transition distribution $\mathbb{P}_h^{\mathcal{M}}(\cdot|z_h, a_h)$. Repeating this procedure would allow estimation of $\mathbb{P}_h^{\mathcal{M}}$ (see Lemma E.1).

This idea is formalized in the procedure `ConstructMDP` (Algorithm 1): given a sequence of general policies $\pi^1, \ldots, \pi^H$ (abbreviated $\pi^{1:H}$), `ConstructMDP` constructs an MDP, denoted $\widehat{\mathcal{M}} = \widehat{\mathcal{M}}(\pi^{1:H})$, which empirically approximates $\mathcal{M}$ using the sampling procedure described above. For technical reasons, $\widehat{\mathcal{M}}$ actually has state space $\overline{\mathcal{Z}} := \mathcal{A}^L \cdot \overline{\mathcal{O}}^L$, where $\overline{\mathcal{O}} := \mathcal{O} \cup \{o^{\text{sink}}\}$ and $o^{\text{sink}}$ is a special "sink observation" so that, after $o^{\text{sink}}$ is observed, all future observations are also $o^{\text{sink}}$. Furthermore, we remark that $d_{\mathsf{S},h-L}^{\mathcal{P},\pi^h}$ does not have to be exactly uniform – it suffices if $\pi^h$ visits all states of $\mathcal{P}$ at step $h - L$ with non-negligible probability.

## 4.2 Exploration via barycentric spanners

The above procedure for approximating $\mathcal{M}$ with $\widehat{\mathcal{M}}$ omits a crucial detail: *how can we find the "exploratory policies" $\pi^{1:H}$?* Indeed a major obstacle to finding such policies is that *we never directly*

---

[2]For simplicity, descriptions of the reward function of $\mathcal{M}$ are omitted; we refer the reader to the appendix for the full details of the proof

*observe the states of* $\mathcal{P}$. By repeatedly playing a policy $\pi$ on $\mathcal{P}$, we can estimate the induced observation visitation distribution $d_{\mathsf{O},h-L}^{\mathcal{P},\pi}$, which is related to the state visitation distribution via the equality $\mathbb{O}_{h-L}^{\dagger} \cdot d_{\mathsf{O},h-L}^{\mathcal{P},\pi} = d_{\mathsf{S},h-L}^{\mathcal{P},\pi}$. Unfortunately, the matrix $\mathbb{O}_{h-L}$ is still unknown, and in general *unidentifiable*.

On the positive side, we can attempt to learn $\mathcal{M}$ layer by layer – in particular, when learning the $h$th layer, we can assume that we have learned previous layers, i.e., $d_{\mathsf{O},h-L}^{\widehat{\mathcal{M}},\pi}$ approximates $d_{\mathsf{O},h-L}^{\mathcal{M},\pi}$, and therefore $d_{\mathsf{O},h-L}^{\mathcal{P},\pi}$. Even though $\widehat{\mathcal{M}}$ does not have underlying latent states, we can define "formal" latent state distributions on $\widehat{\mathcal{M}}$ in analogy with $\mathcal{P}$, i.e. $d_{\mathsf{S},h-L}^{\widehat{\mathcal{M}},\pi} := \mathbb{O}_{h-L}^{\dagger} \cdot d_{\mathsf{O},h-L}^{\widehat{\mathcal{M}},\pi}$. But this does not seem helpful, again because $\mathbb{O}_{h-L}$ is unknown. Our first key insight is that a policy $\pi^h$, for which $d_{\mathsf{S},h-L}^{\widehat{\mathcal{M}},\pi^h}$ puts non-negligible mass on all states, can be found (when it exists) via knowledge of $\widehat{\mathcal{M}}$ and the technique of *Barycentric Spanners* – all without ever explicitly computing $d_{\mathsf{S},h-L}^{\widehat{\mathcal{M}},\pi^h}$.

**Barycentric spanners.** Suppose we knew that the transitions of our empirical estimate $\widehat{\mathcal{M}}$ approximate those of $\mathcal{M}$ up to step $h$, and we want to find a policy $\pi^h$ for which the (formal) latent state distribution $d_{\mathsf{S},h-L}^{\widehat{\mathcal{M}},\pi^h}$ is non-negligible on all states. Unfortunately, the set of achievable latent state distributions $\{\mathbb{O}_{h-L}^{\dagger} \cdot d_{\mathsf{O},h-L}^{\widehat{\mathcal{M}},\pi} : \pi \in \Pi^{\mathrm{gen}}\} \subset \mathbb{R}^S$ is defined implicitly, via the unknown observation matrix $\mathbb{O}_{h-L}$. But we do have access to $\mathcal{X}_{\widehat{\mathcal{M}},h-L} := \{d_{\mathsf{O},h-L}^{\widehat{\mathcal{M}},\pi} : \pi \in \Pi^{\mathrm{gen}}\} \subset \mathbb{R}^O$, the set of achievable distributions over the observation at step $h - L$. In particular, for any reward function on observations at step $h - L$, we can efficiently (by dynamic programming) find a policy $\pi$ that maximizes reward on $\widehat{\mathcal{M}}$ over all policies. In other words, we can solve *linear optimization* problems over $\mathcal{X}$. By a classic result [AK08], we can thus efficiently find a *barycentric spanner* for $\mathcal{X}_{\widehat{\mathcal{M}},h-L}$:

**Definition 4.1** (Barycentric spanner). Consider a subset $\mathcal{X} \subset \mathbb{R}^d$. For $B \geq 1$, a set $\mathcal{X}' \subset \mathcal{X}$ of size $d$ is a *B-approximate barycentric spanner* of $\mathcal{X}$ if each $x \in \mathcal{X}$ can be expressed as a linear combination of elements in $\mathcal{X}'$ with coefficients in $[-B, B]$.

Using the guarantee of [AK08] (restated as Lemma D.1) applied to the set $\mathcal{X}_{\widehat{\mathcal{M}},h-L}$, we can find, in time polynomial in the size of $\widehat{\mathcal{M}}$, a 2-approximate barycentric spanner $\widetilde{\pi}^1, \ldots, \widetilde{\pi}^O$ of $\mathcal{X}_{\widehat{\mathcal{M}},h-L}$; this procedure is formalized in `BarySpannerPolicy` (Algorithm 2). Thus, for any policy $\pi$, the observation distribution $d_{\mathsf{O},h-L}^{\widehat{\mathcal{M}},\pi}$ induced by $\pi$ is a linear combination of the distributions $\{d_{\mathsf{O},h-L}^{\widehat{\mathcal{M}},\widetilde{\pi}^i} : i \in [O]\}$ with coefficients in $[-2, 2]$. Since $d_{\mathsf{S},h-L}^{\widehat{\mathcal{M}},\pi} = \mathbb{O}_{h-L}^{\dagger} \cdot d_{\mathsf{O},h-L}^{\widehat{\mathcal{M}},\pi}$ for all $\pi$, it follows that the formal latent state distribution $d_{\mathsf{S},h-L}^{\widehat{\mathcal{M}},\pi}$ induced by $\pi$ is a linear combination of the formal latent state distributions $\{d_{\mathsf{S},h-L}^{\widehat{\mathcal{M}},\widetilde{\pi}^i} : i \in [O]\}$ with the same coefficients. Now, intuitively, the randomized mixture policy $\pi_{\mathrm{mix}} = \frac{1}{O}(\widetilde{\pi}^1 + \cdots + \widetilde{\pi}^O)$ should explore every reachable state; indeed, we show the following guarantee for `BarySpannerPolicy`:

**Lemma 4.1** (Informal version & special case of Lemma D.2). *In the above setting, under some technical conditions, for all $s \in \mathcal{S}$ and $\pi \in \Pi^{\mathrm{gen}}$, it holds that $d_{\mathsf{S},h-L}^{\widehat{\mathcal{M}},\pi_{\mathrm{mix}}}(s) \geq \frac{1}{4O^2} \cdot d_{\mathsf{S},h-L}^{\widehat{\mathcal{M}},\pi}(s)$.*

But recall the original goal: a policy $\pi^h$ which explores $\mathcal{P}$ – not $\widehat{\mathcal{M}}$. Unfortunately, those states in $\mathcal{P}$ which can only be reached with probability that is small compared to the distance between $\mathcal{P}$ and $\widehat{\mathcal{M}}$ may be missed by $\pi_{\mathrm{mix}}$. When we use $\pi_{\mathrm{mix}}$ to compute the next-step transitions of $\widehat{\mathcal{M}}$, this will lead to additional error when applying belief contraction (Theorem 2.1), and therefore additional error between $\mathcal{P}$ and $\widehat{\mathcal{M}}$. If not handled carefully, this error will compound exponentially over layers.

### 4.3 The full algorithm via iterative discovery

The solution to the dilemma discussed above is to not try to construct our estimate $\widehat{\mathcal{M}}$ of $\mathcal{M}$ layer by layer, hoping that at each layer we can explore all reachable states of $\mathcal{P}$ despite making errors in

---

**Algorithm 1** ConstructMDP

---

1: **procedure** CONSTRUCTMDP($L, N_0, N_1, \pi^1, \ldots, \pi^H$)
2:     **for** $1 \leq h \leq H$ **do**
3:         Let $\widehat{\pi}^h$ be the policy which follows $\pi^h$ for the first $\max\{h - L - 1, 0\}$ steps and thereafter chooses uniformly random actions.
4:         Draw $N_0$ independent trajectories from the policy $\widehat{\pi}^h$:
5:         Denote the data from the $i$th trajectory by $a^i_{1:H-1}, o^i_{2:H}$, for $i \in [N_0]$.
6:         Set $z^i_h = (a^i_{h-L:h-1}, o^i_{h-L+1:h})$ for all $i \in [N_0], h \in [H]$.
7:         *// Construct the transitions $\mathbb{P}^{\widehat{\mathcal{M}}}_h(z_{h+1}|z_h, a_h)$ as follows:*
8:         **for** each $z_h = (a_{h-L:h-1}, o_{h-L+1:h}) \in \mathcal{Z}, a_h \in \mathcal{A}$ **do**
9:             *// Define $\mathbb{P}^{\widehat{\mathcal{M}}}_h(\cdot|z_h, a_h)$ to be the empirical distribution of $z^i_{h+1}|z^i_h, a^i_h$, as follows:*
10:            For $o_{h+1} \in \mathcal{O}$, define $\varphi(o_{h+1}) := |\{i : (a^i_{\max\{1,h-L\}:h}, o^i_{\max\{2,h-L+1\}:h+1}) = (a_{\max\{1,h-L\}:h}, o_{\max\{2,h-L+1\}:h+1})\}|.$
11:            **if** $\sum_{o_{h+1}} \varphi(o_{h+1}) \geq N_1$ **then**
12:                Set $\mathbb{P}^{\widehat{\mathcal{M}}}_h((a_{h-L+1:h}, o_{h-L+2:h+1})|z_h, a_h) := \frac{\varphi(o_{h+1})}{\sum_{o'_{h+1}} \varphi(o'_{h+1})}$ for all $o_{h+1} \in \mathcal{O}$.
13:                Set $R^{\widehat{\mathcal{M}}}_h(z_h, a_h) := R^{\mathcal{P}}_h(o^i_h)$ for some $i$ with $o^i_h = o_h$.    ▷ $R^{\mathcal{P}}_h(o^i_h)$ *is observed.*
14:            **else**
15:                Let $\mathbb{P}^{\widehat{\mathcal{M}}}_h(\cdot|z_h, a_h)$ put all its mass on $(a_{h-L+1:h}, (o_{h-L+2:h}, o^{\text{sink}}))$.
16:         **for** each $z_h = (a_{h-L:h-1}, o_{h-L+1:h}) \in \overline{\mathcal{Z}}\backslash\mathcal{Z}$ and $a_h \in \mathcal{A}$ **do**
17:            Let $\mathbb{P}^{\widehat{\mathcal{M}}}_h(\cdot|z_h, a_h)$ put all its mass on $(a_{h-L+1:h}, (o_{h-L+2:h}, o^{\text{sink}}))$.
18:     Let $\widehat{\mathcal{M}}$ denote the MDP $(\overline{\mathcal{Z}}, H, \mathcal{A}, R^{\widehat{\mathcal{M}}}, \mathbb{P}^{\widehat{\mathcal{M}}})$.
19:     **return** the MDP $\widehat{\mathcal{M}}$, which we denote by $\widehat{\mathcal{M}}(\pi^{1:H})$.

---

---

**Algorithm 2** BarySpannerPolicy

---

1: **procedure** BARYSPANNERPOLICY($\widehat{\mathcal{M}}, h$)   ▷ $\widehat{\mathcal{M}}$ *is MDP on state space* $\overline{\mathcal{Z}}$, *horizon* $H$; $h \in [H]$
2:     **if** $h \leq L$ **then return** an arbitrary general policy.
3:     Let $\mathscr{O}$ be the linear optimization oracle which given $r \in \mathbb{R}^{\mathcal{O}}$, returns $\arg\max_{\pi \in \Pi^{\text{markov}}_{\mathsf{Z}}} \langle r, d^{\pi, \widehat{\mathcal{M}}}_{\mathsf{O}, h-L} \rangle$ and $\max_{\pi \in \Pi^{\text{markov}}_{\mathsf{Z}}} \langle r, d^{\pi, \widehat{\mathcal{M}}}_{\mathsf{O}, h-L} \rangle$ ▷ *Note that $\mathscr{O}$ can be implemented in time $\widetilde{O}(|\overline{\mathcal{Z}}| \cdot HO)$ using dynamic programming*
4:     Using the algorithm of [AK08] with oracle $\mathscr{O}$, compute policies $\{\pi^1, \ldots, \pi^O\}$ so that $\{d^{\pi^i, \widehat{\mathcal{M}}}_{\mathsf{O}, h-L} : i \in [O]\}$ is a 2-approximate barycentric spanner of $\{d^{\pi, \widehat{\mathcal{M}}}_{\mathsf{O}, h-L} : \pi \in \Pi^{\text{markov}}_{\mathsf{Z}}\}$. ▷ *This algorithm requires only $O(O^2 \log O)$ calls to $\mathscr{O}$*
5:     **return** the general policy $\frac{1}{O} \cdot \sum_{i=1}^{O} \pi^i$.

---

earlier layers of $\widehat{\mathcal{M}}$. Instead, we have to be able to go back and "fix" errors in our empirical estimates at earlier layers. This task is performed in our main algorithm, BaSeCAMP (Algorithm 3). For some $K \in \mathbb{N}$, BaSeCAMP runs for a total of $K$ iterations: at each iteration $k \in [K]$, BaSeCAMP defines $H$ general policies, $\overline{\pi}^{k,1}, \ldots, \overline{\pi}^{k,H} \in \Pi^{\text{gen}}$ (abbreviated $\overline{\pi}^{k,1:H}$; step 4). The algorithm's overall goal is that, for some $k$, for each $h \in [H]$, $\overline{\pi}^{k,h}$ explores all latent states at step $h - L$ that are reachable by any policy.

To ensure that this condition holds at some iteration, BaSeCAMP performs the following two main steps for each iteration $k$: first, it calls Algorithm 1 to construct an MDP, denoted $\widehat{\mathcal{M}}^{(k)}$, using the policies $\overline{\pi}^{k,1:H}$. Then, for each $h \in [H]$, it passes the tuple $(\widehat{\mathcal{M}}^{(k)}, h)$ to BarySpannerPolicy, which returns as output a general policy, $\pi^{k+1,h,0}$. Then, policies $\pi^{k+1,h}$ are produced (step 7) by averaging $\pi^{k+1,h',0}$, for all $h' \geq h$. The policies $\pi^{k+1,h}$ are then mixed into the policies $\overline{\pi}^{k+1,h}$ for the next iteration $k + 1$. It follows from properties of BarySpannerPolicy that, in the event that the policies $\overline{\pi}^{k,h}$ are not sufficiently exploratory, one of the new policies $\pi^{k+1,h}$ visits *some* latent state $(s, h') \in \mathcal{S} \times [H]$, which was not previously visited by $\overline{\pi}^{k,1:H}$ with significant probability. Thus, after a total of $K = O(SH)$ iterations $k$, it follows that we must have visited all (reachable) latent

states of the POMDP. At the end of these $K$ iterations, BaSeCAMP computes an optimal policy for each $\widehat{\mathcal{M}}^{(k)}$ and returns the best of them (as evaluated on fresh trajectories drawn from $\mathcal{P}$; step 12).

---

**Algorithm 3** BaSeCAMP (**Ba**rycentric **S**panner policy **C**over with **A**pproximate **MDP**)

---

1: **procedure** BaSeCAMP($L, N_0, N_1, \alpha, \beta, K$)
2:     Initialize $\pi^{1,1}, \ldots, \pi^{1,H}$ to be arbitrary policies.
3:     **for** $k \in [K]$ **do**
4:         For each $h \in [H]$, set $\overline{\pi}^{k,h} = \frac{1}{k} \sum_{k'=1}^{K} \pi^{k',h}$.
5:         Run ConstructMDP($L, N_0, N_1, \overline{\pi}^{k,1:H}$) and let its output be $\widehat{\mathcal{M}}^{(k)}$.
6:         For each $h \in [H]$, let $\pi^{k+1,h,0}$ be the output of BarySpannerPolicy($\widehat{\mathcal{M}}^{(k)}, h$).
7:         For each $h \in [H]$, define $\pi^{k+1,h} := \frac{1}{H-h+1} \sum_{h'=h}^{H} \pi^{k+1,h',0}$.
8:     *// Choose the best optimal policy amongst all $\widehat{\mathcal{M}}^{(k)}$*
9:     **for** $k \in [K]$ **do**
10:         Let $\pi_\star^k$ denote an optimal policy of $\widehat{\mathcal{M}}^{(k)}$.
11:         Execute $\pi_\star^k$ for $\frac{100 H^2 \log K/\beta}{\alpha^2}$ trajectories and let the mean reward across them be $\widehat{r}^k$.
12:     Let $k^\star = \arg\max_{k \in [K]} \widehat{r}^k$.
13:     **return** $\pi_\star^{k^\star}$.

---

## 5 Proof Outline

We now briefly outline the proof of Theorem 3.1; further details may be found in the appendix. The high-level idea of the proof is to show that the algorithm BaSeCAMP makes a given amount of progress for each iteration $k$, as specified in the following lemma:

**Lemma 5.1** ("Progress lemma": informal version of Lemma I.2). *Fix any iteration $k$ in Algorithm 3, step 3. Then, for some parameters $\delta, \phi$ with $\alpha \gg \delta \gg \phi > 0$, one of the following statements holds:*

1. *Any $(s, h)$ with $d_{\mathsf{S}, h-L}^{\mathcal{P}, \overline{\pi}^{k,h}}(s) < \phi$ satisfies $d_{\mathsf{S}, h-L}^{\mathcal{P}, \pi}(s) \leq \delta$ for all general policies $\pi$.*

2. *There is some $(h, s) \in [H] \times \mathcal{S}$ so that:*
$$d_{\mathsf{S}, h-L}^{\mathcal{P}, \overline{\pi}^{k,h}}(s) < \phi \cdot H^2 S, \quad \text{yet, for all } k' > k: \quad d_{\mathsf{S}, h-L}^{\mathcal{P}, \overline{\pi}^{k',h}}(s) \geq \phi \cdot H^2 S.$$

Given Lemma 5.1, the proof of Theorem 3.1 is fairly straightforward. In particular, each $(h, s) \in [H] \times \mathcal{S}$ can only appear as the specified pair in item 2 of the lemma for a single iteration $k$. Thus, as long as $K > HS$, item 1 must hold for some value of $k$, say $k^\star \in [K]$. In turn, it is not difficult to show from this that the MDP $\widehat{\mathcal{M}}^{(k^\star)}$ is a good approximation of $\mathcal{P}$, in the sense that for any general policy $\pi$, the values of $\pi$ in $\widehat{\mathcal{M}}^{(k^\star)}$ and in $\mathcal{P}$ are close (Lemma H.3). Thus, the optimal policy $\pi_\star^{k^\star}$ of $\widehat{\mathcal{M}}^{(k^\star)}$ will be a near-optimal policy of $\mathcal{P}$, and steps 9 through 12 of BaSeCAMP will identify either the policy $\pi_\star^{k^\star}$ or some other policy $\pi_\star^{k'}$ which has even higher reward on $\mathcal{P}$.

**Proof of the progress lemma.** The bulk of the proof of Theorem 3.1 consists of the proof of Lemma 5.1, which we proceed to outline. Suppose that item 1 of the lemma does not hold, meaning that there is some $\pi \in \Pi^{\mathrm{gen}}$ and some $(h, s) \in [H] \times \mathcal{S}$ so that $d_{\mathsf{S}, h-L}^{\mathcal{P}, \overline{\pi}^{k,h}}(s) < \phi$ yet $d_{\mathsf{S}, h-L}^{\mathcal{P}, \pi}(s) > \delta$; i.e., $\overline{\pi}^{k,h}$ does *not* explore $(s, h - L)$, but the policy $\pi$ does. Roughly speaking, BaSeCAMP ensures that item 2 holds in this case, using the following two steps:

**(A)** First, we show that $\langle e_s, \mathbb{O}_{h-L}^\dagger \cdot d_{\mathsf{O}, h-L}^{\widehat{\mathcal{M}}^{(k)}, \pi} \rangle \geq \delta'$, where $\delta'$ is some parameter satisfying $\delta \gg \delta' \gg \phi$. In words, the estimate of the underlying state distribution provided by $\widehat{\mathcal{M}}^{(k)}$ *also* has the property that some policy $\pi$ visits $(s, h - L)$ with non-negligible probability (namely, $\delta'$). While this statement would be straightforward if $\widehat{\mathcal{M}}^{(k)}$ were a close approximation $\mathcal{P}$, this is *not* necessarily the case (indeed, if it were the case, then item 1 of Lemma 5.1 would hold). To circumvent this issue, we introduce a family of intermediate POMDPs indexed by $H' \in [H]$ and denoted $\overline{\mathcal{P}}_{\phi, H'}(\overline{\pi}^{k,1:H})$,

which we call *truncated POMDPs*. Roughly speaking, the truncated POMDP $\overline{\mathcal{P}}_{\phi,H'}(\overline{\pi}^{k,1:H})$ diverts transitions away from all states at step $H' - L$ which $\overline{\pi}^{k,H'}$ does not visit with probability at least $\phi$. This modification is made to allow Theorem 2.1 to be applied to $\overline{\mathcal{P}}_{\phi,H'}(\overline{\pi}^{k,1:H})$ and any general policy $\pi$. By doing so, we may show a *one-sided* error bound between $\overline{\mathcal{P}}_{\phi,H'}(\overline{\pi}^{k,1:H})$ and $\widehat{\mathcal{M}}^{(k)}$ (Lemma G.4) which, importantly, holds even when the policies $\overline{\pi}^{k,1:H}$ may fail to explore some states. It is this one-sided error bound which implies the lower bound on $\langle e_s, \mathbb{O}_{h-L}^{\dagger} \cdot d_{\mathsf{O},h-L}^{\widehat{\mathcal{M}}^{(k)},\pi} \rangle$.

**(B)** Second, we show that the policy $\pi^{k+1,h,0}$ produced by `BarySpannerPolicy` in step 6 explores $(s, h - L)$ with sufficient probability. To do so, we first use Lemma 4.1 to conclude that $\langle e_s, \mathbb{O}_{h-L}^{\dagger} \cdot d_{\mathsf{O},h-L}^{\widehat{\mathcal{M}}^{(k)},\pi^{k+1,h,0}} \rangle \geq \frac{\delta'}{4O^2}$. The more challenging step is to use this fact to conclude a lower bound on $d_{\mathsf{S},h-L}^{\mathcal{P},\pi^{k+1,h,0}}(s)$; unfortunately, the one-sided error bound between $\mathcal{P}$ and $\widehat{\mathcal{M}}^{(k)}$ that we used in the previous paragraph goes in the wrong direction here. The solution is to use Lemma H.3, which has the following consequence: either $d_{\mathsf{S},h-L}^{\mathcal{P},\pi^{k+1,h,0}}(s)$ is not too small, or else the policy $\pi^{k+1,h,0}$ visits some state at a step *prior to* $h - L$ which was not sufficiently explored by any of the policies $\overline{\pi}^{k,1:H}$ (see Section C for further details). In either case $\pi^{k+1,h,0}$ visits a state that was not previously explored, and the fact that $\pi^{k+1,h,0}$ is mixed in to $\overline{\pi}^{k',h'}$ for $k' > k, h' \leq h$ (steps 4, 7 of `BaSeCAMP`) allows us to conclude item 2 of Lemma 5.1.

## 6  Conclusion

In this paper we have demonstrated the first quasipolynomial-time (and quasipolynomial-sample) algorithm for learning observable POMDPs. Several interesting directions for future work remain:

- It is straightforward to show that a $\gamma$-observable POMDP is $\Omega(\gamma/\sqrt{S})$ weakly-revealing in the sense of [LCSJ22, Assumption 1]. Thus, the results of [JKKL20, LCSJ22] imply that $\gamma$-observable POMDPs can be learned with polynomially many samples, albeit by computationally inefficient algorithms. Thus, as discussed following Theorem 3.1, it is natural to wonder whether we can achieve the best of both worlds: is there a quasipolynomial-time algorithm that only needs polynomially many samples, or can one show a computational-statistical tradeoff?

- It is also natural to ask whether an analogue of Assumption 1.1 for the $\ell_2$ norm, namely that $\|\mathbb{O}_h b - \mathbb{O}_h b'\|_2 \geq \gamma \|b - b'\|_2$ for all $h \in [H]$, is sufficient for computationally efficient learnability. Even the planning version of this question (where the parameters of the POMDP are known and the problem is to find a near-optimal policy) is open.

### Acknowledgments and Disclosure of Funding

N.G. is supported by a Fannie & John Hertz Foundation Fellowship and an NSF Graduate Fellowship. A.M. is supported by a Microsoft Trustworthy AI Grant, NSF Large CCF-1565235, a David and Lucile Packard Fellowship and an ONR Young Investigator Award. D.R. is supported by an Akamai Presidential Fellowship and a U.S. DoD NDSEG Fellowship.

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
