6:         Set $z_h^i = (a_{h-L:h-1}^i, o_{h-L+1:h}^i)$ for all $i \in [N_0], h \in [H]$.
7:         *// Construct the transitions $\mathbb{P}_h^{\widehat{\mathcal{M}}}(z_{h+1}|z_h, a_h)$ as follows:*
8:         **for** each $z_h = (a_{h-L:h-1}, o_{h-L+1:h}) \in \mathcal{Z}$, $a_h \in \mathcal{A}$ **do**
9:             *// Define $\mathbb{P}_h^{\widehat{\mathcal{M}}}(\cdot|z_h, a_h)$ to be the empirical distribution of $z_{h+1}^i|z_h^i, a_h^i$, as follows:*
10:             For $o_{h+1} \in \mathcal{O}$, define $\varphi(o_{h+1}) := |\{i : (a_{\max\{1,h-L\}:h}^i, o_{\max\{2,h-L+1\}:h+1}^i) = (a_{\max\{1,h-L\}:h}, o_{\max\{2,h-L+1\}:h+1})\}|$.
11:             **if** $\sum_{o_{h+1}} \varphi(o_{h+1}) \geq N_1$ **then**
12:                 Set $\mathbb{P}_h^{\widehat{\mathcal{M}}}((a_{h-L+1:h}, o_{h-L+2:h+1})|z_h, a_h) := \frac{\varphi(o_{h+1})}{\sum_{o'_{h+1}} \varphi(o'_{h+1})}$ for all $o_{h+1} \in \mathcal{O}$.
13:                 Set $R_h^{\widehat{\mathcal{M}}}(z_h, a_h) := R_h^{\mathcal{P}}(o_h^i)$ for some $i$ with $o_h^i = o_h$.     ▷ $R_h^{\mathcal{P}}(o_h^i)$ *is observed.*
14:             **else**
15:                 Let $\mathbb{P}_h^{\widehat{\mathcal{M}}}(\cdot|z_h, a_h)$ put all its mass on $(a_{h-L+1:h}, (o_{h-L+2:h}, o^{\text{sink}}))$.
16:         **for** each $z_h = (a_{h-L:h-1}, o_{h-L+1:h}) \in \overline{\mathcal{Z}} \backslash \mathcal{Z}$ and $a_h \in \mathcal{A}$ **do**
17:             Let $\mathbb{P}_h^{\widehat{\mathcal{M}}}(\cdot|z_h, a_h)$ put all its mass on $(a_{h-L+1:h}, (o_{h-L+2:h}, o^{\text{sink}}))$.
18:     Let $\widehat{\mathcal{M}}$ denote the MDP $(\overline{\mathcal{Z}}, H, \mathcal{A}, R^{\widehat{\mathcal{M}}}, \mathbb{P}^{\widehat{\mathcal{M}}})$.
19:     **return** the MDP $\widehat{\mathcal{M}}$, which we denote by $\widehat{\mathcal{M}}(\pi^{1:H})$.

---

**Algorithm 2** BarySpannerPolicy

1: **procedure** BARYSPANNERPOLICY($\widehat{\mathcal{M}}, h$)   ▷ $\widehat{\mathcal{M}}$ *is MDP on state space* $\overline{\mathcal{Z}}$*, horizon $H$; $h \in [H]$*
2:     **if** $h \leq L$ **then return** an arbitrary general policy.
3:     Let $\mathscr{O}$ be the linear optimization oracle which given $r \in \mathbb{R}^{\mathcal{O}}$, returns $\arg\max_{\pi \in \Pi_{\mathsf{Z}}^{\text{markov}}} \langle r, d_{\mathsf{O},h-L}^{\pi,\widehat{\mathcal{M}}} \rangle$ and $\max_{\pi \in \Pi_{\mathsf{Z}}^{\text{markov}}} \langle r, d_{\mathsf{O},h-L}^{\pi,\widehat{\mathcal{M}}} \

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

# A  Additional Related Work

**Learning POMDPs (without computational efficiency).**   Over the past several decades several algorithmic paradigms have been developed for the problem of learning the optimal policy in POMDPs. [LLC09] introduced the *regionalized policy representation* (later developed in [LLC11, LLC13, CLC09]), a model-free Bayesian method which attempts to learn the optimal policy directly by learning a model which aggregates belief states into regions and then marginalizing out over the regions. Several papers [PV08, RCdP07, RPCdK11, KOA19] have also applied model-based Bayesian methods to learning POMDPs, by augmenting the state space with empirical estimates of transition and observation probabilities and redefining the belief state updates in the natural way. See also [DVPWR15] for related Bayesian methods. [AYA18] adapts policy gradient methods to the problem of policy optimization in POMDPs. Various other heuristics have been considered as well (e.g., [SBS05, CJJ$^+$21, CJJT22, JJW$^+$17]). More recently, several works have used recurrent neural networks to parametrize history-dependent policies and value functions for learning POMDPs [Sch90, NES22, MGK21, HDT20]. Essentially all of these results come with no formal guarantees (i.e., achieving PAC bounds computationally efficiently) regarding optimality of the learned policies.

The lack of provable guarantees for learning POMDPs without additional assumptions is unavoidable: [JKKL20, KAL16] show that exponentially many samples are needed to learn an approximately optimal policy in general POMDPs. Furthermore, [MR05] proved the related result that learning the parameters of an HMM is as hard as learning parity with noise. If given exponentially many samples, however, it is possible to prove positive results: in the infinite-horizon nonepisodic setting, [EDKM05] introduces an algorithm which provably outputs a near-optimal policy in time (and sample complexity) scaling exponentially in a certain parameter which depends on a horizon-like quantity of the POMDP.

Closer to our work, by placing appropriate assumptions on the POMDP, many works have proven polynomial sample complexity bounds. Several works [GDB16, ALA16, XCGZ21] have used *spectral methods* (see [HKZ12, AGH$^+$14]) to learn POMDPs with polynomial sample complexity; while, in a sense, these papers do address the question of exploration, they do so by making strong assumptions on the underlying *transitions*, namely that the transitions must be full-rank and furthermore must satisfy a mixing-type condition which implies that the Markov chain induced by any policy mixes quickly. Such assumptions on the transitions were removed in the recent work [JKKL20], which modified the technique of spectral methods to obtain polynomial sample complexity for learning POMDPs that satisfy an $\ell_2$ version of Assumption 1.1: in particular, their sample complexity bound depends polynomially on the inverse of the smallest singular value of any of the observation matrices. Later work [LCSJ22] presented a simpler algorithm for learning such POMDPs, based on maximum likelihood estimation; further, the result of [LCSJ22] applies also to the more general class of *weakly revealing POMDPs*, which includes the *overcomplete* case, namely where there are more states than observations. [JJN21] applies the technique of posterior sampling to learn POMDPs in the Bayesian setting, though with the strong assumption that either different models to be learned are separated in KL divergence, or that the models sampled in the course of the algorithm are good estimates of the true model. [EJKM22] establishes polynomial sample complexity bounds in the case when the latent state is *decodable* from a small suffix of the history. [KECM21b] shows that latent MDPs, a special case of overcomplete POMDPs, are learnable when the individual MDPs have transitions that are well-separated. We emphasize that all of the above works do *not* provide end-to-end computational guarantees on learning POMDPs.

**Learning POMDPs (with computational efficiency).**   Prior to our work, the only computationally efficient learning algorithms for POMDPs applied to very special cases. [JKKL20, KAL16] proved polynomial-time learning results assuming deterministic transitions of the POMDP. [KECM21a] obtains results for learning latent MDPs which are a mixture of 2 underlying models; since the uncertainty in the system can be modeled by a single-dimensional parameter, their results are computationally efficient.

**Planning algorithms in RL.**   The problem of *planning* in POMDPs (namely, finding the optimal policy when the model is known) has been extensively studied, with various proposed heuristics (e.g., [Mon82, CLZ97, HF00a, Hau00, RG02, TK03, PB04, SV05, PGT06, RPPCD08, SV10, SS12, SYHL13, GHL19, Han98, MKKC99, KMN99, LYX11, AYA18]), and a few provably efficient

algorithms [BDRS96, KECM21a, GMR22]. Most closely related to our work is [GMR22], which shows a quasipolynomial-time planning algorithm for observable POMDPs.

**Computational-statistical considerations in RL.** Several recent works have shown that certain learning problems in RL can be solved with few (i.e., polynomial) samples yet have superpolynomial *computational* complexity, under standard assumptions. This is the case for finding a near-optimal policy in $\gamma$-observable POMDPs with $\gamma = 1/\operatorname{poly}(O)$, which requires exponential time under ETH [GMR22] yet can be learned with polynomially many samples [JKKL20]; finding stationary coarse correlated equilibria in general-sum Markov games [DGZ22]; and finding the optimal policy in MDPs which have linear $Q^\star$ and $V^\star$ functions [KLLM22]. The latter result establishes a computational-statistical gap of stronger nature in the sense that (unlike the previous two examples) the planning version of the problem can be solved efficiently. In contrast to the above lower bounds for computationally efficient learning, Theorem 3.1 shows that learning $\gamma$-observable POMDPs with constant $\gamma$ can be done computationally (quasi)-efficiently.

As a whole, the issue of developing computationally efficient algorithms for RL problems is wide open, with a plethora of intriguing open questions (see, e.g., [WAJ$^+$21, JKA$^+$17, FKQR21]).

**Nonstationary MDPs and related approaches.** In light of the fact that the dynamics of a $\gamma$-observable POMDP $\mathcal{P}$ can be approximated by those of an MDP $\mathcal{M}$ with quasipolynomially-sized state space (see Section 4.1), a natural approach to learn an optimal policy of $\mathcal{P}$ is to use one of many recent results (e.g., [WL21, WYDW21, WDZ22]) which establishes regret bounds for learning in MDPs when an adversary is allowed to corrupt the trajectories for some episodes. Such an adversary may be used to model the fact that we can simulate trajectory draws from $\mathcal{P}$ by viewing them as trajectory draws from $\mathcal{M}$ with a certain fraction of episodes being corrupted. In particular, [WYDW21] shows that if $C$ episodes (out of $T$ total episodes) are corrupted, then there is an efficient algorithm with regret upper bounded by $\operatorname{poly}(H) \cdot \tilde{O}(\sqrt{ZAT} + Z^2A + CZA)$, where $Z$ denotes the number of states of the MDP $\mathcal{M}$. In our application, we would have $Z = (OA)^L$ (with $L$ defined as in Theorem 3.1) and $C \geq \alpha \cdot T$ (as Theorem 3.1 gives an upper bound on the probability of an episode being corrupted which is no smaller than $\alpha$). Thus $CZA \geq \alpha T \cdot (OA)^{\log(1/\alpha)} \gg T$, meaning that the bound from [WYDW21] is vacuous. Furthermore, this is not just an artifact of their analysis: [WYDW21] showed a lower bound of $\Omega(CZA)$ on the regret when an adversary is allowed to corrupt $C$ episodes. Thus, in a sense, the adversarial model studied by [WL21, WYDW21, WDZ22] is too strong to prove nonvacuous results in our setting.

Another related approach is the `PC-IGW.Bilinear` algorithm (as well as its `IGW-Spanner` subroutine) of [FKQR21], which use similar techniques (namely, a policy cover and a barycentric spanner) to ours. Moreover, since POMDPs fit into the framework of *decision-making with structured observations* of [FKQR21], one can attempt to use the `PC-IGW.Bilinear` algorithm together with E2D [FKQR21] to attempt to efficiently learn the optimal policy in POMDPs. However, there is a significant issue: the full E2D meta-algorithm of [FKQR21] requires a *model-estimation oracle* (the output of which is passed to `PC-IGW.Bilinear`). One might hope that we can implement this oracle efficiently by using subtrajectories of length $L$ to estimate the transitions of the MDP $\mathcal{M}$ that approximates $\mathcal{P}$. However (using the notation of the previous paragraph) the estimation error would necessarily scale as $\alpha T$. Furthermore, in order to ensure that the output of the estimation oracle belongs to the class of models, the model class must be augmented to contain all MDPs on the state space $\mathcal{Z}$; doing so leads to a bound on the decision estimation coefficient which is at least $\frac{Z}{\gamma}$. The resulting regret bound of [FKQR21, Theorem 4.1] scales as $\inf_{\gamma>0} \frac{TZ}{\gamma} + \gamma \cdot \mathbf{Est}$, where $\mathbf{Est} \geq \alpha T$ denotes the estimation error. This expression is minimized when $\gamma = \sqrt{TZ/\mathbf{Est}}$, and yields a regret bound which is at least $\sqrt{TZ \cdot \mathbf{Est}} \geq T \cdot \sqrt{\alpha \cdot (OA)^{\log 1/\alpha}} \gg T$, which is again vacuous. Finding a general model of learning in "approximate MDPs" which yields nonvacuous regret bounds in our setting is an interesting open problem.

We also remark that the use of barycentric spanners in [FKQR21] differs significantly from ours. In [FKQR21], a weaker notion, known as *approximate G-optimal design*, suffices for their application, yet in order to compute such a G-optimal design efficiently in their setting, it turns out to be more convenient to search for the stronger notion of barycentric spanner. Replacing the barycentric spanner in our algorithm (Algorithm 2) with a G-optimal design does not seem to sufffice for our needs.

# B  Additional preliminaries

In this section we state some additional preliminaries which are useful in the proof and formalize some notions which were introduced informally in the main body.

**Extending the state and observation spaces.**  Fix a POMDP $\mathcal{P} = (\mathcal{S}, \mathcal{A}, \mathcal{O}, H, b_1, R, \mathbb{T}, \mathbb{O})$, as in Theorem 3.1. For technical reasons, we will augment the state spce $\mathcal{S}$ with a special "sink state" $s^{\text{sink}}$, and augment the observation space $\mathcal{O}$ with a special "sink observation" $o^{\text{sink}}$. We write $\overline{\mathcal{S}} := \mathcal{S} \cup \{s^{\text{sink}}\}$ and $\overline{\mathcal{O}} := \mathcal{O} \cup \{o^{\text{sink}}\}$, and call the spaces $\overline{\mathcal{S}}, \overline{\mathcal{O}}$, the *extended state and observation spaces*, respectively. We extend the various components of $\mathcal{P}$ to the extended state and observation sequences as follows: $b_1$ remains unchanged, $R_h(o^{\text{sink}}) = 0$ for all $h$, $\mathbb{T}_h^{\mathcal{P}}(s^{\text{sink}}|s^{\text{sink}}, a) = 1$ for all $a \in \mathcal{A}, h \in [H]$ (and all transitions from states $s \in \mathcal{S}$ remain unchanged), and $\mathbb{O}_h^{\mathcal{P}}(o^{\text{sink}}|s^{\text{sink}}) = 1$ for all $h \in [H]$ (and all observation distributions at states $s \in \mathcal{S}$ remain unchanged). It is clear that the distribution over trajectories given any fixed policy are the same in the original and extended POMDP, and the extended POMDP satisfies $\gamma$-observability if the original POMDP does. Thus, it is without loss of generality to replace $\mathcal{P}$ with the extended POMDP $\mathcal{P} = (\overline{\mathcal{S}}, \mathcal{A}, \overline{\mathcal{O}}, H, b_1, R, \mathbb{T}, \mathbb{O})$, as we shall do in the entirety of the proof (we do so since we will introduce other POMDPs which, unlike $\mathcal{P}$, will at times transition into $s^{\text{sink}}$, unlike the original POMDP $\mathcal{P}$; see Section E.3). Thus, we have that the belief states $\mathbf{b}_h^{\mathcal{P}}(a_{1:h-1}, o_{2:h})$, as well as the approximate belief states $\mathbf{b}_h^{\text{apx}, \mathcal{P}}(a_{h-L:h-1}, o_{h-L+1:h}; \mathcal{D})$, belong to $\mathbb{R}^{\overline{\mathcal{S}}}$. Moreover, it is immediate from the block structure of $\mathbb{O}_h^{\mathcal{P}} \in \mathbb{R}^{\overline{\mathcal{O}} \times \overline{\mathcal{S}}}$ that its pseudoinverse $(\mathbb{O}_h^{\mathcal{P}})^\dagger \in \mathbb{R}^{\overline{\mathcal{S}} \times \overline{\mathcal{O}}}$ has the same block structure, i.e., $(\mathbb{O}_h^{\mathcal{P}})^\dagger_{s^{\text{sink}}, o} = (\mathbb{O}_h^{\mathcal{P}})^\dagger_{s, o^{\text{sink}}} = 0$ for all $s \in \mathcal{S}, o \in \mathcal{O}$.

Finally, we remark that for some choices of $a_{1:h-1}, o_{2:h}$, the belief state $\mathbf{b}_h^{\mathcal{P}}(a_{1:h-1}, o_{2:h})$ or the approximate belief state $\mathbf{b}_h^{\text{apx}, \mathcal{P}}(a_{h-L:h-1}, o_{h-L+1:h})$ may be undefined (for instance, if $o_{h-L+1:h} \notin \overline{\mathcal{O}}^L$ and $\mathcal{D}(s^{\text{sink}}) = 0$, then $\mathbf{b}_h^{\text{apx}, \mathcal{P}}(a_{h-L:h-1}, o_{h-L+1:h}; \mathcal{D})$ is undefined since the transitions of $\mathcal{P}$ never lead to $s^{\text{sink}}$). In such a case, as a matter of convention, we will define the belief state to put all its mass on $s^{\text{sink}}$.

**MDPs with Z-structure.**  At times, we will write $\mathbb{T}^{\mathcal{P}}, \mathbb{O}^{\mathcal{P}}, R^{\mathcal{P}}$ in place of $\mathbb{T}, \mathbb{O}, R$, respectively, to clarify that the corresponding transition matrices, observation matrices, and reward function are for the POMDP $\mathcal{P}$. Furthermore, we will consider a fixed value $L \in \mathbb{N}$ (the particular value of $L$, as a function of the parameters of the problem, is given in Section C.1; it is chosen so as to satisfy the requirements of the belief contraction theorem (Theorem B.2)). Given $\mathcal{P}$, write $\mathcal{Z} := \mathcal{A}^L \times \mathcal{O}^L$ and $\overline{\mathcal{Z}} := \mathcal{A} \times \overline{\mathcal{O}}^L$; the spaces $\mathcal{Z}, \overline{\mathcal{Z}}$ should be interpreted as representing a sequence of $L$ consecutive action-observation pairs for a trajectory drawn from $\mathcal{P}$. Throughout the course of the proof, we will consider several *Markov Decision Processes* (MDPs) whose state space is $\overline{\mathcal{Z}}$. In particular, for various choices of the probability transition matrices $\mathbb{P}_h(z_{h+1}|z_h, a_h)$ (for $z_h, z_{h+1} \in \overline{\mathcal{Z}}$) and reward functions $R = (R_1, \ldots, R_H)$, $R_h : \overline{\mathcal{Z}} \to \mathbb{R}$, we will consider MDPs of the form $\mathcal{M} = (\overline{\mathcal{Z}}, \mathcal{A}, H, R, \mathbb{P})$ (we will typically leave the initial state distribution unspecified, as it does not affect the optimal policy of an MDP).

For such an $\mathcal{M}$, we say that $\mathcal{M}$ *has Z-structure* if: (a) $R_h(z)$ is only a function of $\mathsf{o}(z)$ for all $h \in [H]$, and (b) the transitions $\mathbb{P}_h(\cdot|z_h, a_h)$ have the following property, for $z_h \in \overline{\mathcal{Z}}, a_h \in \mathcal{A}$: writing $z_h = (a_{h-L:h-1}, o_{h-L+1:h})$, $\mathbb{P}_h(z_{h+1}|z_h, a_h)$ is nonzero only for those $z_{h+1}$ of the form $z_{h+1} = (a_{h-L+1:h}, o_{h-L+2:h+1})$, where $o_{h+1} \in \overline{\mathcal{O}}$. Throughout this section, the MDPs with state space $\overline{\mathcal{Z}}$ that we consider all will have Z-structure.

Throughout, we will use the following notational convention: given a sequence of $L$ action-observation pairs for the POMDP $\mathcal{P}$, namely $(a_{h-L:h-1}, o_{h-L+1:h}) \in (\mathcal{A} \times \overline{\mathcal{O}})^L$, we will denote this sequence by $z_h = (a_{h-L:h-1}, o_{h-L+1:h})$. When the identity of the action-observation sequence is clear, we will use $z_h$ and $(a_{h-L:h-1}, o_{h-L+1:h})$ interchangeably. Furthermore, for a sequence $z_h = (a_{h-L:h-1}, o_{h-L+1:h})$, write $\mathsf{o}(z_h) := o_h$ to denote the final observation of this sequence.

We define $\Pi_{\mathsf{Z}}^{\text{markov}}$ to be the set of sequences $\pi = (\pi_1, \ldots, \pi_H)$, where each $\pi_h : \overline{\mathcal{Z}} \to \mathcal{A}$. $\Pi_{\mathsf{Z}}^{\text{markov}}$ may be identified as a subset of $\Pi^{\text{det}}$: a policy $\pi = (\pi_1, \ldots, \pi_H) \in \Pi_{\mathsf{Z}}^{\text{markov}}$ may be seen as a history-dependent deterministic policy via the mapping $(a_{1:h-1}, o_{2:h}) \mapsto \pi_h(a_{h-L:h-1}, o_{h-L+1:h})$

(i.e., by ignoring the portion of the history prior to step $h - L$). Given an MDP $\mathcal{M}$ with $\mathsf{Z}$-structure and a policy $\pi \in \Pi_{\mathsf{Z}}^{\mathrm{markov}}$, the definition of a trajectory drawn from $\mathcal{M}$ by following policy $\pi$ is standard (see Section 2.1). We next extend this definition to the case of a *general policy* $\pi \in \Pi^{\mathrm{gen}}$, for which a trajectory $z_{1:H}$ (which is equivalent to the data of $a_{1:H-1}, o_{2:H}$) is drawn as follows: first, a deterministic policy $\sigma \in \Pi^{\mathrm{det}}$ is drawn, $\sigma \sim \pi$, and then, at each step $h \in [H]$, given history $z_{1:h}$ (which is equivalent to the data of $a_{1:h-1}, o_{2:h}$, since $\mathcal{M}$ has $\mathsf{Z}$-structure), we take action $\sigma_h(a_{1:h-1}, o_{2:h})$. At times, we will abuse notation and write $\sigma_h(z_{1:h}) = \sigma_h(a_{1:h-1}, o_{2:h})$. Additionally, we write $\mathbb{P}_{z_{1:H} \sim \pi}^{\mathcal{M}}(\mathcal{E})$ to denote the probability of $\mathcal{E}$ when $z_{1:H}$ (which is equivalent to the data of $a_{1:H-1}, o_{2:H}$) is drawn from a trajectory following policy $\pi$ for the MDP $\mathcal{M}$. In similar spirit, we will write $\mathbb{E}_{z_{1:H} \sim \pi}^{\mathcal{M}}[\cdot]$ to denote expectations.

**Visitation distributions.** Next, for an element $z = (a_{1:L}, o_{1:L}) \in \overline{\mathcal{Z}}$, as well as $0 \le h$, write $\mathrm{suff}_h(z) := (a_{\max\{1,h-L+1\}:L}, o_{\max\{1,h-L+1\}:L}) \in \mathcal{A}^{\min\{h,L\}} \times \overline{\mathcal{O}}^{\min\{h,L\}}$ to denote the suffix of $z$ that consists of the last $h$ actions and observations in $z$, or all of $z$ if $h > L$. Recall from Section 2.4 the following notation for visitation distributions for the POMDP $\mathcal{P}$ and an MDP $\mathcal{M}$ with $\mathsf{Z}$-structure:

- Given a general policy $\pi \in \Pi^{\mathrm{gen}}$, $h \in [H]$, and $s \in \overline{\mathcal{S}}$, define $d_{\mathsf{S},h}^{\mathcal{P},\pi}(s) = \mathbb{P}_{s_{1:H} \sim \pi}^{\mathcal{P}}(s_h = s)$.

- For a general policy $\pi \in \Pi^{\mathrm{gen}}$, $h \in [H]$, and $o \in \overline{\mathcal{O}}$, define $d_{\mathsf{O},h}^{\mathcal{P},\pi}(o) = \mathbb{P}_{o_{1:H} \sim \pi}^{\mathcal{P}}(o_h = o)$.

- For a general policy $\pi \in \Pi^{\mathrm{gen}}$, $h \in [H]$, and $z \in \overline{\mathcal{Z}}$, define $d_{\mathsf{Z},h}^{\mathcal{P},\pi}(z) = \mathbb{P}_{a_{1:H}, o_{2:H} \sim \pi}^{\mathcal{P}}(\mathrm{suff}_{h-1}(a_{h-L:h-1}, o_{h-L+1:h}) = \mathrm{suff}_{h-1}(z))$.

- For a general policy $\pi \in \Pi^{\mathrm{gen}}$, $h \in [H]$, and $z \in \overline{\mathcal{Z}}$, define $d_{\mathsf{Z},h}^{\mathcal{M},\pi}(z) = \mathbb{P}_{z_{1:H} \sim \pi}^{\mathcal{M}}(\mathrm{suff}_{h-1}(z_h) = \mathrm{suff}_{h-1}(z))$.

- For a general policy $\pi \in \Pi^{\mathrm{gen}}$, $h \in [H]$, and $o \in \overline{\mathcal{O}}$, define $d_{\mathsf{O},h}^{\mathcal{M},\pi}(o) = \mathbb{P}_{z_{1:H} \sim \pi}^{\mathcal{M}}(\mathsf{o}(z_h) = o)$.

Notice that, for $h \le L$, $d_{\mathsf{Z},h}^{\mathcal{P},\pi}$ and $d_{\mathsf{Z},h}^{\mathcal{M},\pi}$ are not distributions, i.e., we will have $\sum_{z \in \overline{\mathcal{Z}}} d_{\mathsf{Z},h}^{\mathcal{P},\pi}(z) > 1$, since we consider the suffix $\mathrm{suff}_{h-1}(z)$ in the above definitions. This choice is made to ensure that, for $z = (a_{h-L:h-1}, o_{h-L+1:h})$, the values $d_{\mathsf{Z},h}^{\mathcal{P},\pi}(z), d_{\mathsf{Z},h}^{\mathcal{M},\pi}(z)$ do not depend on elements of $z$ with nonpositive indices when $h \le L$.

Given the definitions above, we will view the corresponding visitations as vectors in their respective spaces: in particular, $d_{\mathsf{S},h}^{\mathcal{P},\pi} \in \mathbb{R}^{\overline{\mathcal{S}}}$, $d_{\mathsf{O},h}^{\mathcal{P},\pi}, d_{\mathsf{O},h}^{\mathcal{M},\pi} \in \mathbb{R}^{\overline{\mathcal{O}}}$, and $d_{\mathsf{Z},h}^{\mathcal{P},\pi}, d_{\mathsf{Z},h}^{\mathcal{M},\pi} \in \mathbb{R}^{\overline{\mathcal{Z}}}$. Note that the vectors $d_{\mathsf{O},h}^{\mathcal{P},\pi} \in \mathbb{R}^{\overline{\mathcal{O}}}$ and $d_{\mathsf{S},h}^{\mathcal{P},\pi} \in \mathbb{R}^{\overline{\mathcal{S}}}$ satisfy $d_{\mathsf{O},h}^{\mathcal{P},\pi} = \mathbb{O}_h^{\mathcal{P}} \cdot d_{\mathsf{S},h}^{\mathcal{P},\pi}$. Furthermore, we will slightly overload notation as follows: for a state visitation vector $d$ in $\mathbb{R}^{\mathcal{T}}$ (for some set $\mathcal{T}$), and $\mathcal{T}' \subset \mathcal{T}$, we write $d(\mathcal{T}') := \sum_{t \in \mathcal{T}'} d(t)$.

### B.1 Negative indices.

We remark that our expressions will, at times, refer to state visitation distributions of the form $d_{\mathsf{S},h-L}^{\mathcal{P},\pi}$ (or $d_{\mathsf{O},h-L}^{\mathcal{P},\pi}$, etc.), for some fixed $L \in \mathbb{N}$ and $h \in [H]$. Of course, if $h \le L$, then this distribution is not defined. In all such cases, though, it will be the case that, for $h \le L$, the expression in question does not depend on the distribution $d_{\mathsf{S},h-L}^{\mathcal{P},\pi}$, and thus the expression is well-defined. In a similar manner, for a trajectory $(a_{1:H-1}, o_{2:H})$ drawn from a POMDP $\mathcal{P}$, we will at times consider expressions of the form $(a_{h-L:h-1}, o_{h-L+1:h})$, for $h \in [H]$. In the case that $h \le L$, we consider that the actions $a_{h-L}, \ldots, a_0$ and observations $o_{h-L+1}, \ldots, o_1$ to be chosen arbitrarily (to be concrete, one may consider that all $a_{h'} = a^\star$ and $o_{h'+1} = o^\star$ for $h' \le 0$ and some universal $a^\star \in \mathcal{A}, o^\star \in \mathcal{O}$) – they will not affect the value of the relevant expression, and we use this convention purely to simplify notation.

### B.2 Contextual decision processes

Both POMDPs and MDPs with $\mathsf{Z}$-structure are special cases of *contextual decision processes* [JKA+17]: a contextual decision process is a tuple $\mathcal{P} = (\mathcal{A}, \overline{\mathcal{O}}, H, R, \mathbb{P})$, where: $\mathcal{A}$ is the *ac-*

*tion space*, $\overline{\mathcal{O}}$ is the *observation space* (also known as the *context space*), $H$ is the horizon, $R = (R_1, \ldots, R_H)$ is a collection of reward functions, $R_h : \overline{\mathcal{O}} \to \mathbb{R}$, and $\mathbb{P}$ is the *transition kernel*. Formally, we have $\mathbb{P} = (\mathbb{P}_1, \ldots, \mathbb{P}_H)$, where $\mathbb{P}_h : \mathscr{H}_h \times \mathcal{A} \to \Delta(\overline{\mathcal{O}})$ is a mapping from the space of histories $\mathscr{H}_h$ (where $\mathscr{H}_h = \mathcal{A}^{h-1} \times \overline{\mathcal{O}}^{h-1}$) and actions to the space of distributions over the next observation (context). Given $h \geq 1$, a history $(a_{1:h-1}, o_{2:h})$ and an action $a_h \in \mathcal{A}$, we write $\mathbb{P}_h(o_{h+1}|a_{1:h}, o_{2:h})$ to denote the probability of observing $o_{h+1}$ given the history $(a_{1:h-1}, o_{2:h})$ and that we take action $a_h$. We will consider the space of general policies $\Pi^{\mathrm{gen}}$ (defined in Section 2 as the convex hull of $\Pi^{\mathrm{det}}$, which is the space of tuples $\pi = (\pi_1, \ldots, \pi_H)$, where $\pi_h : \mathscr{H}_h \to \mathcal{A}$) acting on a contextual decision process.

POMDPs and MDPs with Z-structure may seen to be special case of contextual decision processes, as follows:

- Given a POMDP $\mathcal{P} = (\overline{\mathcal{S}}, \mathcal{A}, \overline{\mathcal{O}}, H, b_1, R, \mathbb{T}, \mathbb{O})$, we view it as the contextual decision process $\mathcal{P} = (\mathcal{A}, \overline{\mathcal{O}}, H, R, \mathbb{P})$, where $\mathbb{P}$ is defined as follows: given a history $(a_{1:h-1}, o_{2:h})$ and an action $a_h \in \mathcal{A}$, define

$$\mathbb{P}_h^{\mathcal{P}}(o_{h+1}|a_{1:h}, o_{2:h}) := e_{o_{h+1}}^{\top} \mathbb{O}_{h+1}^{\mathcal{P}} \cdot \mathbb{T}_h^{\mathcal{P}}(a_h) \cdot \mathbf{b}_h^{\mathcal{P}}(a_{1:h-1}, o_{2:h}),$$

  which is exactly the probability (in the POMDP $\mathcal{P}$) of next observing $o_{h+1}$ given the history $(a_{1:h-1}, o_{2:h})$ and action $a_h$.

- Given an MDP with Z-structure, $\mathcal{M} = (\overline{\mathcal{Z}}, \mathcal{A}, H, R, \mathbb{P}^{\mathcal{M}})$, we view it as the contextual decision process $\mathcal{P} = (\mathcal{A}, \overline{\mathcal{O}}, H, R, \mathbb{P}^{\mathcal{P}})$, where we use the fact that (by Z-structure) $R = (R_1, \ldots, R_H)$ has the property that each $R_h(z)$ is only a function of $\mathsf{o}(z)$, and $\mathbb{P}^{\mathcal{P}}$ is defined as follows. We have $\mathbb{P}^{\mathcal{P}} = (\mathbb{P}_1^{\mathcal{P}}, \ldots, \mathbb{P}_H^{\mathcal{P}})$, where

$$\mathbb{P}_h^{\mathcal{P}}(o_{h+1}|a_{1:h}, o_{2:h}) := \mathbb{P}_h^{\mathcal{M}}(z_{h+1}|z_h, a_h),$$

  where $z_h = (a_{h-L:h-1}, o_{h-L+1:h})$ and $z_{h+1} = (a_{h-L+1:h}, o_{h-L+2:h+1})$, and we use the fact that $\mathcal{M}$ has Z-structure to guarantee that $\mathbb{P}_h^{\mathcal{P}}(\cdot|a_{1:h}, o_{2:h})$ is a probability distribution.

**Value functions.** Given a deterministic policy $\sigma \in \Pi^{\mathrm{det}}$ and a contextual decision process $\mathcal{P} = (\mathcal{A}, \overline{\mathcal{O}}, H, R, \mathbb{P})$, the value function $V_h^{\sigma, \mathcal{P}}(a_{1:h-1}, o_{2:h})$ is defined in an inductive manner:

$$V_h^{\sigma, \mathcal{P}}(a_{1:h-1}, o_{2:h}) := \mathbb{E}_{o_{h+1} \sim \mathbb{P}_h(\cdot|a_{1:h}, o_{2:h})} \left[ R_{h+1}(o_{h+1}) + V_{h+1}^{\sigma, \mathcal{P}}(a_{1:h}, o_{2:h+1}) \right], \qquad (4)$$

where we have written $a_h := \pi_h(a_{1:h-1}, o_{2:h})$ above. Then, for a general policy $\pi \in \Pi^{\mathrm{gen}}$, we define $V_h^{\pi, \mathcal{P}}(a_{1:h-1}, o_{2:h}) := \mathbb{E}_{\sigma \sim \pi} \left[ V_h^{\sigma, \mathcal{P}}(a_{1:h-1}, o_{2:h}) \right]$. We will often modify contextual decision processes by considering different choices of their reward function. Given a contextual decision process $\mathcal{P}$ and a reward function $R = (R_1, \ldots, R_H)$ (which may not be the reward function of $\mathcal{P}$), where $R_h : \mathcal{O} \to \mathbb{R}$ for each $h \in [H]$, we write $V_h^{\pi, \mathcal{P}, R}(\cdot)$ to denote the value under $\pi$ of $\mathcal{P}$ where its reward function is replaced by $R$. We will often consider the value function of a contextual decision process $\mathcal{P}$ which is actually either a POMDP or a MDP with Z-structure, in which case the following remarks apply:

- If $\mathcal{P}$ is a POMDP, then $V_1^{\pi, \mathcal{P}}(\emptyset) = \mathbb{E}_{(a_{1:H-1}, o_{2:H}) \sim \pi}^{\mathcal{P}} \left[ \sum_{h=2}^{H} R_h(o_h) \right]$, i.e., our definition of the value function $V_1^{\pi, \mathcal{P}}(\emptyset)$ given here by reverse induction corresponds with that given in Section 2.

- If $\mathcal{P}$ is a MDP with Z-structure, then, it is straightforward to see that $V_h^{\pi, \mathcal{P}}(a_{1:h-1}, o_{2:h})$ depends only on $z_h = (a_{h-L:h-1}, o_{h-L+1:h})$ (by the Markovian structure of $\mathcal{P}$), so we will often write $V_h^{\pi, \mathcal{P}}(z_h)$ in place of $V_h^{\pi, \mathcal{P}}(a_{1:h-1}, o_{2:h})$.

**Viewing $\mathbb{P}_h$ as an operator.** Consider a contextual decision process $\mathcal{P} = (\mathcal{A}, \overline{\mathcal{O}}, H, R, \mathbb{P})$. For $h \in [H]$, we will view $\mathbb{P}_h$ as the following operator: given a function $f : \mathscr{H}_{h+1} \to \mathbb{R}$, define $\mathbb{P}_h f : \mathscr{H}_h \times \mathcal{A} \to \mathbb{R}$ as:

$$(\mathbb{P}_h f)(a_{1:h}, o_{2:h}) := \mathbb{E}_{o_{h+1} \sim \mathbb{P}_h(\cdot|a_{1:h}, o_{2:h})} \left[ f(a_{1:h}, o_{2:h+1}) \right].$$

Thus we may alternatively write (4) as $V_h^{\sigma,\mathcal{P}}(a_{1:h-1}, o_{2:h}) = (\mathbb{P}_h(R_{h+1} + V_{h+1}^{\sigma,\mathcal{P}}))(a_{1:h}, o_{2:h})$, where we view $R_{h+1}$ as a function on $\mathscr{H}_{h+1}$ in the natural way, i.e., with abuse of notation, $R_{h+1}(a_{1:h}, o_{2:h+1}) := R_{h+1}(o_{h+1})$.

If $\mathcal{P}$ is an MDP with $\mathsf{Z}$-structure and $f(a_{1:h}, o_{2:h+1})$ only depends on the most recent $L$ actions and observations (i.e., $(a_{h-L+1:h}, o_{h-L+2:h+1})$), then $(\mathbb{P}_h f)(a_{1:h}, o_{2:h})$ only depends on $z_h := (a_{h-L:h-1}, o_{h-L+1:h})$ and $a_h$. Thus, we will often write $(\mathbb{P}_h f)(z_h, a_h) := (\mathbb{P}_h f)(a_{1:h}, o_{2:h})$.

**Lemma B.1** ([DLB17], Lemma E.15). *Consider any two CDPs $\mathcal{P}, \mathcal{P}'$, with reward functions $R^{\mathcal{P}}, R^{\mathcal{P}'}$ and transition kernels $\mathbb{P}^{\mathcal{P}}, \mathbb{P}^{\mathcal{P}'}$. Then for any deterministic policy $\pi$ and all $h \in [H]$, it holds that, for all $a_{1:h-1}, o_{2:h}$,*

$$V_h^{\pi,\mathcal{P}}(a_{1:h-1}, o_{2:h}) - V_h^{\pi,\mathcal{P}'}(a_{1:h-1}, o_{2:h})$$
$$= \mathbb{E}_{(a_{h:H-1}, o_{h+1:H})\sim\pi}^{\mathcal{P}} \left[ \sum_{t=h}^{H} \left( (\mathbb{P}_t^{\mathcal{P}} R_{t+1}^{\mathcal{P}})(a_{1:t}, o_{2:t}) - (\mathbb{P}_t^{\mathcal{P}'} R_{t+1}^{\mathcal{P}'})(a_{1:t}, o_{2:t}) \right. \right.$$
$$\left. \left. + ((\mathbb{P}_t^{\mathcal{P}} - \mathbb{P}_t^{\mathcal{P}'}) V_{t+1}^{\pi,\mathcal{P}'})(a_{1:t}, o_{2:t}) \right) \mid a_{1:h-1}, o_{2:h} \right].$$

*Proof.* The lemma is essentially an immediate consequence of [DLB17, Lemma E.15], since we may view $\mathcal{P}, \mathcal{P}'$ as MDPs on the state space of histories. For completeness, we write out the full proof in our setting.

We use reverse induction on $h$, noting that the lemma statement is immediate for $h = H + 1$ since $V_{H+1}^{\pi,\mathcal{P}}$ and $V_{H+1}^{\pi,\mathcal{P}'}$ are identically 0. Now assume that the statement of the lemma holds at time step $h + 1$. Fix any $(a_{1:h-1}, o_{2:h}) \in \mathscr{H}_h$, and let $a_h := \pi_h(a_{1:h-1}, o_{2:h})$. Then

$$V_h^{\pi,\mathcal{P}}(a_{1:h-1}, o_{2:h}) = (\mathbb{P}_h^{\mathcal{P}} R_{h+1}^{\mathcal{P}})(a_{1:h}, o_{2:h}) + (\mathbb{P}_h^{\mathcal{P}} V_{h+1}^{\pi,\mathcal{P}})(a_{1:h}, o_{2:h})$$
$$V_h^{\pi,\mathcal{P}'}(a_{1:h-1}, o_{2:h}) = (\mathbb{P}_h^{\mathcal{P}'} R_{h+1}^{\mathcal{P}'})(a_{1:h}, o_{2:h}) + (\mathbb{P}_h^{\mathcal{P}'} V_{h+1}^{\pi,\mathcal{P}'})(a_{1:h}, o_{2:h}),$$

and an analogous equality holds for $\mathcal{P}'$. Then we may compute, for any fixed $(a_{1:h-1}, o_{2:h}) \in \mathscr{H}_h$ (again letting $a_h := \pi_h(a_{1:h-1}, o_{2:h})$),

$$V_h^{\pi,\mathcal{P}}(a_{1:h-1}, o_{2:h}) - V_h^{\pi,\mathcal{P}'}(a_{1:h-1}, o_{2:h})$$
$$= (\mathbb{P}_h^{\mathcal{P}} R_{h+1}^{\mathcal{P}})(a_{1:h}, o_{2:h}) - (\mathbb{P}_h^{\mathcal{P}'} R_{h+1}^{\mathcal{P}'})(a_{1:h}, o_{2:h})$$
$$\quad + ((\mathbb{P}_h^{\mathcal{P}} - \mathbb{P}_h^{\mathcal{P}'}) V_{h+1}^{\pi,\mathcal{P}'})(a_{1:h}, o_{2:h}) + (\mathbb{P}_h^{\mathcal{P}} (V_{h+1}^{\pi,\mathcal{P}} - V_{h+1}^{\pi,\mathcal{P}'}))(a_{1:h}, o_{2:h})$$
$$= (\mathbb{P}_h^{\mathcal{P}} R_{h+1}^{\mathcal{P}})(a_{1:h}, o_{2:h}) - (\mathbb{P}_h^{\mathcal{P}'} R_{h+1}^{\mathcal{P}'})(a_{1:h}, o_{2:h}) + ((\mathbb{P}_h^{\mathcal{P}} - \mathbb{P}_h^{\mathcal{P}'}) V_{h+1}^{\pi,\mathcal{P}'})(a_{1:h}, o_{2:h})$$
$$\quad + \mathbb{E}_{(a_h, o_{h+1})\sim\pi}^{\mathcal{P}} \left[ (V_{h+1}^{\pi,\mathcal{P}} - V_{h+1}^{\pi,\mathcal{P}'})(a_{1:h}, o_{2:h+1}) \mid a_{1:h-1}, o_{2:h} \right]$$
$$= (\mathbb{P}_h^{\mathcal{P}} R_{h+1}^{\mathcal{P}})(a_{1:h}, o_{2:h}) - (\mathbb{P}_h^{\mathcal{P}'} R_{h+1}^{\mathcal{P}'})(a_{1:h}, o_{2:h}) + ((\mathbb{P}_h^{\mathcal{P}} - \mathbb{P}_h^{\mathcal{P}'}) V_{h+1}^{\pi,\mathcal{P}'})(a_{1:h}, o_{2:h})$$
$$\quad + \mathbb{E}_{(a_h, o_{h+1})\sim\pi}^{\mathcal{P}} \left[ \mathbb{E}_{(a_{h+1:H-1}, o_{h+2:H})\sim\pi}^{\mathcal{P}} \left[ \sum_{t=h+1}^{H} (\mathbb{P}_t^{\mathcal{P}} R_{t+1}^{\mathcal{P}})(a_{1:t}, o_{2:t}) - (\mathbb{P}_t^{\mathcal{P}'} R_{t+1}^{\mathcal{P}'})(a_{1:t}, o_{2:t}) \right. \right.$$
$$\quad + \left. \left. \sum_{t=h+1}^{H} ((\mathbb{P}_t^{\mathcal{P}} - \mathbb{P}_t^{\mathcal{P}'}) V_{t+1}^{\pi,\mathcal{P}'})(a_{1:t}, o_{2:t}) \mid a_{1:h}, o_{2:h+1} \right] \mid a_{1:h-1}, o_{2:h} \right]$$
$$= \mathbb{E}_{(a_{h:H-1}, o_{h+1:H})\sim\pi}^{\mathcal{P}} \left[ \sum_{t=h}^{H} (\mathbb{P}_t^{\mathcal{P}} R_{t+1}^{\mathcal{P}})(a_{1:t}, o_{2:t}) - (\mathbb{P}_t^{\mathcal{P}'} R_{t+1}^{\mathcal{P}'})(a_{1:t}, o_{2:t}) \right.$$
$$\quad + \left. \sum_{t=h}^{H} ((\mathbb{P}_h^{\mathcal{P}} - \mathbb{P}_h^{\mathcal{P}'}) V_{t+1}^{\pi,\mathcal{P}'})(a_{1:t}, o_{2:t}) \mid a_{1:h-1}, o_{2:h} \right],$$

which completes the inductive step. $\qquad\square$

## B.3 Belief contraction, expanded

Below we state a slight strengthening of Theorem 2.1 which will be needed in our proofs.

**Theorem B.2** (Belief contraction; Theorems 4.1 and 4.7 of [GMR22])**.** *Consider any $\gamma$-observable POMDP $\mathcal{P}$, any $\epsilon > 0$ and $L \in \mathbb{N}$ so that $L \geq C \cdot \min\left\{ \frac{\log(1/(\epsilon\phi))\log(\log(1/\phi)/\epsilon)}{\gamma^2}, \frac{\log(1/(\epsilon\phi))}{\gamma^4} \right\}$. Fix any $\pi \in \Pi^{\mathrm{gen}}$, and suppose that $\mathscr{D}', \mathscr{D} \in \Delta^{\mathcal{S}}$ satisfy $\frac{\mathscr{D}'(s)}{\mathscr{D}(s)} \leq \frac{1}{\phi}$ (where we take $0/0 = 0$). Then*

$$\mathbb{E}^{\mathcal{P}}_{\substack{s_{h-L} \sim \mathscr{D}' \\ (a_{h-L:h-1}, o_{h-L+1:h}) \sim \pi}} \left\| \mathbf{b}^{\mathrm{apx},\mathcal{P}}_h(a_{h-L:h-1}, o_{h-L+1:h}; \mathscr{D}') - \mathbf{b}^{\mathrm{apx},\mathcal{P}}_h(a_{h-L:h-1}, o_{h-L+1:h}; \mathscr{D}) \right\|_1 \tag{5}$$
$$\leq \epsilon.$$

Note that the theorems in [GMR22] only state the special case $\mathscr{D} = \mathrm{Unif}(\mathcal{S})$, in which case the precondition of the theorem is always satisfied with $\phi = 1/S$. However, the proof extends unchanged to this generality.

### B.4 Pseudoinverses

As previously stated, for a matrix $\mathbb{O} \in \mathbb{R}^{O \times S}$, we let $\mathbb{O}^{\dagger}$ denote the Moore-Penrose pseudoinverse of $\mathbb{O}$. The following lemma is straightforward:

**Lemma B.3.** *If $\mathbb{O} \in \mathbb{R}^{O \times S}$ is $\gamma$-observable, then for all $x \in \mathbb{R}^O$, $\|\mathbb{O}^{\dagger}x\|_1 \leq \frac{\sqrt{S}}{\gamma} \cdot \|x\|_1$. In particular, all entries of $\mathbb{O}^{\dagger}$ are bounded in magnitude by $\frac{\sqrt{S}}{\gamma}$.*

*Proof.* Write $y = \mathbb{O}^{\dagger}x$. Then

$$\|y\|_1 \leq \frac{1}{\gamma} \cdot \|\mathbb{O}y\|_1 \leq \frac{\sqrt{S}}{\gamma} \cdot \|\mathbb{O}y\|_2 = \frac{\sqrt{S}}{\gamma} \cdot \|\mathbb{O}\mathbb{O}^{\dagger}x\|_2 \leq \frac{\sqrt{S}}{\gamma} \cdot \|x\|_2 \leq \frac{\sqrt{S}}{\gamma} \cdot \|x\|_1.$$

where the first inequality follows from observability and the second follows from Cauchy-Schwartz. To establish the second statement, note that, for each $o \in [O]$, we have that $\|\mathbb{O}^{\dagger} \cdot e_o\|_1 \leq \frac{\sqrt{S}}{\gamma} \cdot \|e_o\|_1 = \frac{\sqrt{S}}{\gamma}$. $\qquad\square$

## C  Proof Overview and Organization

Below we describe the organization of the proof of Theorem 3.1 which obtains a guarantee for `BaSeCAMP` (Algorithm 3). At a high level, our task is to show that the true POMDP $\mathcal{P}$ and the MDPs with Z-structure $\widehat{\mathcal{M}}^{(k)}$ constructed in the course of `BaSeCAMP` are close in some sense; doing so is crucial to proving the main "progress lemma" (Lemma I.2, the formal version of Lemma 5.1) as discussed in Section 5:

- In Section D, we introduce the concept of barycentric spanners (Definition D.1) and prove our main lemma (Lemma D.2, the formal version of Lemma 4.1) which establishes a guarantee on the output of Algorithm 2 (`BarySpannerPolicy`).

- In Section E we introduce several intermediary contextual decision processes which we will use to relate the true POMDP $\mathcal{P}$ and the empirical MDPs $\widehat{\mathcal{M}}^{(k)}$ constructed in `BaSeCAMP`: these include an MDP $\widetilde{\mathcal{M}}(\pi^{1:H})$ with Z-structure (which depends on a sequence of general policies $\pi^{1:H}$; see Section E.1) and a collection of truncated POMDPs, $\overline{\mathcal{P}}_{\phi,h}(\pi^{1:H})$, $h \in [H]$ (which depend on a parameter $\phi > 0$ and a sequence of general policies $\pi^{1:H}$; see Section E.3). Omitting some details, the way we relate the various contextual decision processes is summarized as follows:

$$\mathcal{P} \overset{\text{Lems. F.1 \& F.2}}{\sim} \overline{\mathcal{P}}_{\phi,h}(\pi^{1:H}) \overset{\text{Lems. F.3 \& G.4}}{\sim} \widetilde{\mathcal{M}}(\pi^{1:H}) \overset{\text{Lems. E.1, F.4 \& G.4}}{\sim} \widehat{\mathcal{M}}^{(k)}. \tag{6}$$

- The truncated POMDPs $\overline{\mathcal{P}}_{\phi,h}(\pi^{1:H})$ constructed in Section E.3 are a special case of a more general notion of *truncation* of the given POMDP $\mathcal{P}$ (Definition F.1). In Section F we prove several properties of truncations of POMDPs, which are useful for establishing several steps in the relations (6).

- In Section G we use the results of Section F to establish a certain type of "one-sided" relation between $\overline{\mathcal{P}}_{\phi,h}(\pi^{1:H})$ and $\widehat{\mathcal{M}}^{(k)}$ (for each $k \in [K]$). In particular, the main result of Section G is Lemma G.4, which establishes the following: for any reward function $R$ which is non-negative in a certain sense, the value of any general policy $\pi$ in $\overline{\mathcal{P}}_{\phi,h}(\pi^{1:H})$ with the reward function $R$ is approximately bounded above by the value of $\pi$ in $\widehat{\mathcal{M}}^{(k)}$ with the reward function $R$; i.e., we have an upper bound on $V_1^{\pi,\overline{\mathcal{P}}_{\phi,h}(\pi^{1:H}),R}(\emptyset) - V_1^{\pi,\widehat{\mathcal{M}}^{(k)},R}(\emptyset)$. The inequality does not hold, though, in the opposite direction: this is because $\overline{\mathcal{P}}_{\phi,h}(\pi^{1:H})$ is obtained by redirecting some of the transitions of $\mathcal{P}$ into the sink state $s^{\text{sink}}$ (which will always lead to 0 reward), and lacking a bound on the probability of ending up in $s^{\text{sink}}$, it is impossible to show an inequality in the opposite direction.

- Our proof also requires a two-sided inequality that upper bounds $\left| V_1^{\pi,\mathcal{P},R}(\emptyset) - V_1^{\pi,\widehat{\mathcal{M}}^{(k)},R}(\emptyset) \right|$ for general policies $\pi$. Such an inequality is established in Section H, namely in Lemma H.3. The proof of Lemma H.3 proceeds somewhat differently than that of the one-sided inequalities, in that it does not make use of the truncated POMDPs $\overline{\mathcal{P}}_{\phi,h}(\pi^{1:H})$; in particular, as opposed to (6), the rough outline is as follows:

$$\mathcal{P} \;\overset{\text{Lems. H.2 \& H.3}}{\sim}\; \widetilde{\mathcal{M}}(\pi^{1:H}) \;\overset{\text{Lems. E.1, F.4, \& H.3}}{\sim}\; \widehat{\mathcal{M}}^{(k)}. \tag{7}$$

Unsurprisingly, there is a cost to the fact that this inequality is two-sided (which is of course stronger than being one-sided): the upper bound on $\left| V_1^{\pi,\mathcal{P},R}(\emptyset) - V_1^{\pi,\widehat{\mathcal{M}}^{(k)},R}(\emptyset) \right|$ depends on the probability, say $\rho(\pi)$, that $\pi$ visits the set of states which are "underexplored" (see Definition E.2) with respect to a certain set of exploratory policies used by BaSeCAMP. This additional term of $\rho(\pi)$ is dealt as follows when proving the "progress lemma", Lemma I.2: we will be applying Lemma H.3 with $\pi = \pi^{k+1,h,0}$ for some $h \in [H]$, namely one of the "exploratory policies" constructed in BaSeCAMP with the purpose of exploring states at step $h - L$. If the probability $\rho(\pi^{k+1,h,0})$ is too large, then, by the precise definition of $\rho(\pi^{k+1,h,0})$, it turns out that $\pi^{k+1,h,0}$ must explore some state *prior to* step $h - L$ (even though $\pi^{k+1,h,0}$ was constructed with the "purpose" of exploring states at step $h - L$!). We can then use this fact to conclude that item 2 of Lemma I.2 holds in this case.

- Finally, Section I analyzes BaSeCAMP, proving Theorem 3.1. As outlined in Section 5, the main step is to prove Lemma I.2, which establishes that BaSeCAMP makes a quantifiable amount of progress at each iteration $k$. Using Lemma I.2, Lemma I.3 establishes that, in fact, at some iteration $\overline{k}$ in BaSeCAMP, the item 1 of Lemma I.2 must hold, which, roughly speaking, means the POMDP $\mathcal{P}$ has been sufficiently explored. Lemma I.4 uses this fact together with Lemma H.3 to establish a *two-sided* upper bound between $\mathcal{P}$ and $\widehat{\mathcal{M}}^{(\overline{k})}$ (without the additional term involving underexplored states referred to above). From this result, the proof of Theorem 3.1 is straightforward.

### C.1 Definitions of hyperparameters

Below we give the values for the various hyperparameters employed by our algorithm and used in the proof, given a $\gamma$-observable POMDP with $S$ states, $A$ actions, $O$ observations, horizon $H$, and a desired accuracy level $\alpha$ and probability of failure $\beta$ (in the context of Theorem 3.1). $C^\star$ refers to a sufficiently large constant. Throughout the remainder of the appendix, when we will refer to the values of the constants below with any qualifiers, they will be assumed to take the below values.

- $\epsilon = \alpha \cdot \frac{\gamma}{O^2 H^5 S^{3/2} (C^\star)^2}$.
- $\phi = \frac{1}{C^\star} \cdot \frac{\gamma}{H^5 S^{7/2} O^2} \cdot \epsilon$.
- $L = C^\star \cdot \min \left\{ \frac{\log(1/(\epsilon\phi))\log(\log(1/\phi)/\epsilon)}{\gamma^2}, \frac{\log(1/(\epsilon\phi))}{\gamma^4} \right\}$.
- $\theta = \epsilon$ (see Lemma E.1).
- $\zeta = \frac{\epsilon\phi}{A^{2L} O^L}$ (see Lemma E.1).
- $\delta = C^\star \cdot \frac{O^2 H^3 \sqrt{S}}{\gamma} \cdot \epsilon$.

- $\delta' = \delta/2$.
- $K = 2HS$.
- $p = \beta/(2K)$.
- $N_1 = C^\star \cdot \frac{LA^{L+1}O^L \log(AO/p)}{\theta^2}$ (used in Algorithm 1).
- $N_0 = C^\star \cdot \frac{N_1 AL \log(OA/p)}{\zeta}$ (used in Algorithm 1).

We assume that $H$ is chosen sufficiently large so that $H > L$ (if not, then we can increase $H$ to be $L + 1$, which only affects the time and sample complexities by a constant factor in the exponent).

## D   Barycentric spanners

Definition D.1 below formally defines the notion of *barycentric spanner*.

**Definition D.1.** Consider a subset $\mathcal{X} \subset \mathbb{R}^d$. For $C \geq 1$, a set $\mathcal{X}' \subset \mathcal{X}$ of size $d$ is a *C-approximate barycentric spanner* of $\mathcal{X}$ if each $x \in \mathcal{X}$ can be expressed as a linear combination of elements in $\mathcal{X}'$ with coefficients in $[-C, C]$.

It is easy to show that (exact) barycentric spanners exist: it is without loss of generality to assume that $\mathcal{X}$ is full-rank, since otherwise we may apply a linear transformation to it so that the transformed set is full-rank (in a smaller-dimensional space). Then, given that $\mathcal{X}$ is full-rank, it may be verified that a collection of $d$ vectors $x_1, \ldots, x_d \in \mathcal{X}$ maximizing $|\det(x_1, \ldots, x_d)|$ is a (1-approximate) barycentric spanner. We will use the following result of Awerbuch & Kleinberg, showing that approximate barycentric spanners can be computed efficiently:

**Lemma D.1** ([AK08], Proposition 2.5). *Suppose $\mathcal{X} \subset \mathbb{R}^d$ and $\lambda > 1$. Given an oracle for optimizing linear functions over $\mathcal{X}$, a $\lambda$-approximate barycentric spanner for $\mathcal{X}$ may be computed in polynomial time using $O(d^2 \log_\lambda(d))$ calls to the optimization oracle.*

*Further, each element $x_i$ of the resulting approximate barycentric spanner is equal to the output of the optimization oracle for some input cost vector.*

We remark that the algorithm of [AK08] technically applies to the setting that $\mathcal{X}$ is full-dimensional. In general this will not be the case for sets $\mathcal{X}$ to which we will apply Lemma D.1. We may deal with sets $\mathcal{X} \subset \mathbb{R}^d$ which are not full-dimensional as follows: using at most $d$ oracle calls to the optimization oracle, we may compute a set of $k \leq d$ linearly independent vectors which spans $\mathcal{X}$. Using these $k$ vectors, we may compute a one-to-one linear mapping $T$ from $\mathcal{X}$ into $\mathbb{R}^k$, so that the image of $\mathcal{X}$ in $\mathbb{R}^k$, say $\mathcal{Y}$, is full-dimensional. We may then use Lemma D.1 to compute a $\lambda$-approximate barycentric spanner of $\mathcal{Y}$, and then map this back into $\mathcal{X}$ via $T^{-1}$, yielding a $\lambda$-approximate barycentric spanner of $\mathcal{X}$.

In Lemma D.2 below, we consider positive integers $S, O \in \mathbb{N}$, a matrix $\mathbb{M} \in \mathbb{R}^{S \times O}$, as well as a set $\mathcal{X} \subset \mathbb{R}^O$ indexed by some given set $\Pi$. In particular, there is a bijection between $\Pi$ and $\mathcal{X}$, which we denote as $\pi \mapsto x(\pi)$. In our usage of Lemma D.2, the set $\mathcal{X}$ will be the set of all possible observation visitation distributions (indexed by policies $\pi$) at a particular layer of the current empirical MDP with Z-structure $\widehat{\mathcal{M}}^{(k)}$. The matrix $\mathbb{M}$ will be equal to $\mathbb{O}_h^\dagger$ for some layer $h$ of the POMDP. Thus, for $\pi \in \Pi$, $\mathbb{M} \cdot x(\pi) = \mathbb{O}_h^\dagger \cdot x(\pi)$ represents an estimate of the state visitation distribution of $\mathcal{P}$ under policy $\pi$ at step $h$. In this context, the conclusion of Lemma D.2 should be interpreted as saying that the mixture policy defined as $\frac{\pi^1 + \cdots + \pi^O}{O}$ visits all states which can be visited with non-negligible probability.

**Lemma D.2.** *Fix $\eta > 0$, $S, O \in \mathbb{N}$, a matrix $\mathbb{M} \in \mathbb{R}^{S \times O}$, and a set $\mathcal{X} \subset \mathbb{R}^O$, indexed by some class $\Pi$. Assume that for each $x \in \mathcal{X}$, we have*

$$\sum_{s \in \mathcal{S}} [(\mathbb{M} \cdot x)(s)]_- \leq \frac{\eta}{4O^2}. \tag{8}$$

*Consider an oracle $\mathscr{O}$ which returns, given any $r \in \mathbb{R}^O$, some $\pi^\star(r) \in \arg\max_{\pi \in \Pi} \langle r, x(\pi) \rangle$ as well as $\langle r, x(\pi^*(r)) \rangle$. Then there is a polynomial-time algorithm which only has access to the oracle $\mathscr{O}$*

*(i.e., does not know $\mathbb{M}$), makes $O(O^2 \log O)$ oracle calls, and returns $\pi^1, \ldots, \pi^O \in \Pi$ satisfying*

$$\forall \mathcal{S}' \subset [S] \text{ s.t. } \max_{\pi \in \Pi} \sum_{s \in \mathcal{S}'} \langle e_s, \mathbb{M} \cdot x(\pi) \rangle \geq \eta, \qquad \sum_{s \in \mathcal{S}'} \left\langle e_s, \frac{\mathbb{M} \cdot (x(\pi^1) + \cdots + x(\pi^O))}{O} \right\rangle \geq \frac{\eta}{4O^2}. \tag{9}$$

*Proof.* By Lemma D.1, with $O(O^2 \log O)$ calls to the oracle $\mathscr{O}$, we may compute a set $\{\pi^1, \ldots, \pi^O\}$ so that $\{x(\pi^1), \ldots, x(\pi^O)\}$ is a 2-approximate barycentric spanner of $\mathcal{X}$.

Consider any $\mathcal{S}' \subset [S]$ so that, for some $\pi \in \Pi$, $\sum_{s \in \mathcal{S}'} \langle e_s, \mathbb{M} \cdot x(\pi) \rangle \geq \eta$. By the definition of barycentric spanner, we can write $x(\pi) = \sum_{o=1}^{O} \alpha_o \cdot x(\pi^o)$, for some reals $\alpha_o \in [-2, 2]$, $o \in [O]$. Then we have

$$\eta \leq \sum_{s \in \mathcal{S}'} \langle e_s, \mathbb{M} \cdot x(\pi) \rangle = \sum_{o=1}^{O} \alpha_o \cdot \left\langle \sum_{s \in \mathcal{S}'} e_s, \mathbb{M} \cdot x(\pi^o) \right\rangle.$$

Thus there must be some $o \in [O]$ so that $\left| \langle \sum_{s \in \mathcal{S}'} e_s, \mathbb{M} \cdot x(\pi^o) \rangle \right| \geq \frac{\eta}{2O}$, which implies, by the assumption (8), that $\langle \sum_{s \in \mathcal{S}'} e_s, \mathbb{M} \cdot x(\pi^o) \rangle \geq \frac{\eta}{2O}$. Since, for all $o' \in [O]$, we similarly have, by (8),

$$\left\langle \sum_{s \in \mathcal{S}'} e_s, \mathbb{M} \cdot x(\pi^{o'}) \right\rangle \geq - \sum_{s \in \mathcal{S}} \left[ \left( \mathbb{M} \cdot x(\pi^{o'}) \right)(s) \right]_- \geq -\frac{\eta}{4O^2},$$

it follows that

$$\left\langle \sum_{s \in \mathcal{S}'} e_s, \mathbb{M} \cdot \frac{x(\pi^1) + \cdots + x(\pi^O)}{O} \right\rangle \geq \frac{\eta}{2O^2} - \frac{\eta}{4O^2} = \frac{\eta}{4O^2}.$$

$\square$

# E  Intermediary Models for the Analysis

In Section 4.1, we gave an overview of the notion of an MDP $\mathcal{M}$ which approximates the POMDP $\mathcal{P}$, using approximate belief states initialized with the uniform distribution. We then suggested that, if appropriate policies $\pi^{1:H}$ can be found, then (using ConstructMDP, Algorithm 1), we can construct an MDP $\widehat{\mathcal{M}}(\pi^{1:H})$ which can approximate $\mathcal{M}$, and therefore $\mathcal{P}$. To make this argument more formal, we use an MDP which differs slightly from the "intermediate" MDP $\mathcal{M}$ introduced in Section 4.1 to relate $\widehat{\mathcal{M}}(\pi^{1:H})$ and $\mathcal{P}$. In particular, in Section E.1 we introduce an MDP $\widetilde{\mathcal{M}}(\pi^{1:H})$ for any set of policies $\pi^{1:H}$. At a high level, $\widetilde{\mathcal{M}}(\pi^{1:H})$ uses approximate belief states initialized with the visitation distributions induced by $\pi^{1:H}$. In Section E.2 we show by concentration arguments that $\widetilde{\mathcal{M}}(\pi^{1:H})$ is close to the "empirical MDP" $\widehat{\mathcal{M}}(\pi^{1:H})$ (constructed in Algorithm 1) for arbitrary $\pi^{1:H}$.

Subsequently, the core technical challenge will be to relate $\widetilde{\mathcal{M}}(\pi^{1:H})$ to $\mathcal{P}$ under various assumptions on $\pi^{1:H}$. As described in the proof overview, we will crucially use *truncated POMDPs* as an intermediary model. These are introduced in Section E.3.

## E.1  Construction of Approximate MDP $\widetilde{\mathcal{M}}(\pi^{1:H})$

Fix a POMDP $\mathcal{P}$, and consider any collection of $H$ general policies $\pi^1, \ldots, \pi^H$, which we abbreviate as $\pi^{1:H}$. For each $h \in [H]$, let $\widehat{\pi}^h$ denote the policy which follows $\pi^h$ for the first $\max\{h - L - 1, 0\}$ steps and then chooses a uniformly random action for steps $h - L$ through $h$. Write $\widehat{\pi} := \frac{1}{H} \cdot \sum_{h=1}^{H} \widehat{\pi}^h$.

Given the policies $\pi^{1:H}$ as above, we now define an MDP with Z-structure, denoted $\widetilde{\mathcal{M}}(\pi^{1:H}) = (\overline{\mathcal{Z}}, H, \mathcal{A}, \mathbb{P}^{\widetilde{\mathcal{M}}(\pi^{1:H})})$, as follows. To define the transitions $\mathbb{P}_h^{\widetilde{\mathcal{M}}(\pi^{1:H})}$, first define, for $\pi' \in \Pi^{\text{gen}}$,

$$\widetilde{\mathbf{b}}_h^{\pi'}(a_{h-L:h-1}, o_{h-L+1:h}) := \mathbf{b}_h^{\text{apx}, \mathcal{P}}(a_{h-L:h-1}, o_{h-L+1:h}; d_{\mathsf{S}, h-L}^{\pi', \mathcal{P}}).$$

(Recall the definition of $\mathbf{b}_h^{\mathrm{apx},\mathcal{P}}(\cdot)$ in Section 2.3; furthermore, recall that, if $h \leq L$, $\mathbf{b}_h^{\mathrm{apx},\mathcal{P}}(a_{h-L:h-1}, o_{h-L+1:h}; \mathscr{D})$ is defined as $\mathbf{b}_h^{\mathrm{apx},\mathcal{P}}(a_{1:h-1}, o_{2:h}; \mathscr{D})$, which does not depend on $\mathscr{D}$, as it is $\mathbf{b}_h^{\mathcal{P}}(a_{1:h-1}, o_{2:h})$. Thus, even for $h \leq L$, $\widetilde{\mathbf{b}}_h^{\pi'}$ is well-defined despite the fact that $d_{\mathsf{S},h-L}^{\pi^h,\mathcal{P}}$ is not.)

Given a sequence $a_{1:h}, o_{2:h+1}$, writing $z_h = (a_{h-L:h-1}, o_{h-L+1:h})$, $z_{h+1} = (a_{h-L+1:h}, o_{h-L+2:h+1})$, define, in the case that $z_h \in \mathcal{Z}$,

$$\mathbb{P}_h^{\widetilde{\mathcal{M}}(\pi^{1:H})}(z_{h+1}|z_h, a_h) = e_{o_{h+1}}^\top \cdot \mathbb{O}_{h+1} \cdot \mathbb{T}_h(a_h) \cdot \widetilde{\mathbf{b}}_h^{\pi^h}(a_{h-L:h-1}, o_{h-L+1:h}). \tag{10}$$

Note that $\widetilde{\mathcal{M}}(\pi^{1:H})$ has $\mathsf{Z}$-structure: given $z_h = (a_{h-L:h-1}, o_{h-L+1:h}), a_h$, under the transitions of $\widetilde{\mathcal{M}}(\pi^{1:H})$, all coordinates of $z_{h+1}$ except the last observation are full determined (namely, they are given by $(a_{h-L+1:h}, o_{h-L+2:h})$); thus, with slight abuse of notation, we will at times write $\mathbb{P}_h^{\widetilde{\mathcal{M}}(\pi^{1:H})}(o_{h+1}|z_h, a_h) = \mathbb{P}_h^{\widetilde{\mathcal{M}}(\pi^{1:H})}(z_{h+1}|z_h, a_h)$. If $z_h \notin \mathcal{Z}$, then we define $\mathbb{P}_h^{\widetilde{\mathcal{M}}(\pi^{1:H})}(o^{\mathrm{sink}}|z_h, a_h) = 1$. Finally, we remark that we have not specified a reward function or initial state distribution for $\widetilde{\mathcal{M}}(\pi^{1:H})$; such choices turn out not to affect the analysis.

## E.2  Concentration of Empirical Approximate MDP $\widehat{\mathcal{M}}(\pi^{1:H})$.

While we do not have algorithmic access to $\widetilde{\mathcal{M}}(\pi^{1:H})$, we can algorithmically construct an empirical approximation of it. This is precisely the (random) MDP $\widehat{\mathcal{M}}(\pi^{1:H}) = (\overline{\mathcal{Z}}, H, \mathcal{A}, R^{\widehat{\mathcal{M}}(\pi^{1:H})}, \mathbb{P}^{\widehat{\mathcal{M}}(\pi^{1:H})})$, defined as the output of Algorithm 1 when given input policies $\pi^{1:H}$. Notice that the MDP $\widehat{\mathcal{M}}(\pi^{1:H})$ does come with a reward function $R^{\widehat{\mathcal{M}}(\pi^{1:H})}$; we remark that, at some points in the proof, we will consider an alternative reward function (generally denoted by $R$, without the superscript) attached to the remaining data of $\widehat{\mathcal{M}}(\pi^{1:H})$, purely for purposes of analysis.

Lemma E.1 shows by Chernoff bounds that outside the set of states which are rarely visited by $\widehat{\pi}^{1:H}$ (defined in Definition E.1 below), we have that $\widehat{\mathcal{M}}(\pi^{1:H})$ closely approximates $\widetilde{\mathcal{M}}(\pi^{1:H})$.

To state Lemma E.1, we need to introduce some notation.

**Definition E.1.** For each $h \in [H]$, $\zeta > 0$, and a general policy $\pi'$, define the set $\mathcal{Z}_{h,\zeta}^{\mathrm{low}}(\pi') \subset \mathcal{Z}$, as follows:

$$\mathcal{Z}_{h,\zeta}^{\mathrm{low}}(\pi') := \left\{ z \in \mathcal{Z} \ : \ d_{\mathsf{Z},h}^{\mathcal{P},\pi'}(z) \leq \zeta \right\}.$$

**Lemma E.1.** *Fix any sequence of policies $\pi^{1:H}$ and the resulting $\widehat{\pi}^{1:H}$ as defined in Section E.1. There is an event $\mathcal{E}^{\mathrm{low}}$ that occurs with probability $1 - p$, so that under the event $\mathcal{E}^{\mathrm{low}}$, we have:*

1. *The output $\widehat{\mathcal{M}}(\pi^{1:H})$ of ConstructMDP with parameters $N_1, N_0$ satisfying*

$$N_1 \geq C_0 \cdot \left( \frac{LA^{L+1}O^L \log(AO/p)}{\theta^2} \right), \quad N_0 \geq C_0 \cdot \frac{N_1 AL \log(OA/p)}{\zeta} \tag{11}$$

   *(where $C_0$ is a sufficiently large constant) satisfies the following: for all $z_h \notin \mathcal{Z}_{h,\zeta}^{\mathrm{low}}(\widehat{\pi}^h)$ and all $a_h \in \mathcal{A}$, we have that $R_h^{\widehat{\mathcal{M}}(\pi^{1:H})}(o(z_h)) = R_h^{\mathcal{P}}(o(z_h))$ and*

$$\left\| \mathbb{P}_h^{\widehat{\mathcal{M}}(\pi^{1:H})}(\cdot|z_h, a_h) - \mathbb{P}_h^{\widetilde{\mathcal{M}}(\pi^{1:H})}(\cdot|z_h, a_h) \right\|_1 \leq \theta. \tag{12}$$

2. *For all $z_h \in \mathcal{Z}_{h,\zeta}^{\mathrm{low}}(\widehat{\pi}^h)$, if (12) does not hold, then under the transitions of $\widehat{\mathcal{M}}(\pi^{1:H})$, for all $a_h \in \mathcal{A}$, $z_h = (a_{h-L:h-1}, o_{h-L+1:h})$ transitions to $(a_{h-L+1:h}, (o_{h-L+2:h}, o^{\mathrm{sink}}))$ with probability 1.*

Notice that, for $h \leq L$, $z_h = (a_{h-L:h-1}, o_{h-L+1:h})$ includes actions (and observations) with nonpositive indices: importantly, such actions and observations do not affect the distribution of the next observation, namely $\mathbb{P}_h^{\widehat{\mathcal{M}}(\pi^{1:H})}(o_{h+1}|z_h, a_h)$ or $\mathbb{P}_h^{\widetilde{\mathcal{M}}(\pi^{1:H})}(o_{h+1}|z_h, a_h)$, which intuitively explains why (12) can hold even for such $h$.

*Proof of Lemma E.1.* First, note that (12) is immediate if $z_h \notin \mathcal{Z}$, since then

$$\mathbb{P}_h^{\widehat{\mathcal{M}}(\pi^{1:H})}(\cdot|z_h, a_h) = \mathbb{P}_h^{\widetilde{\mathcal{M}}(\pi^{1:H})}(\cdot|z_h, a_h) = o^{\text{sink}}$$

with probabiltiy 1. Thus for the remainder of the proof we may consider $z_h \in \mathcal{Z}$.

Consider some $p' \in (0, 1)$, to be specified below. Fix any $h \in [H]$: we will argue about the $h$th step in the for loop on step 2 of Algorithm 1. We will use the following key fact: for any $(z_h, a_h) \in \mathcal{Z} \times \mathcal{A}$, and for any iteration $i$ of ConstructMDP, the distribution of $z_{h+1}^i$ conditioned on $a_h^i = a_h$ and $\text{suff}_{h-1}(z_h^i) = \text{suff}_{h-1}(z_h)$ is the same as $\mathbb{P}_h^{\widetilde{\mathcal{M}}(\pi^{1:H})}(\cdot|z_h, a_h)$. To see that this fact holds, it suffices to note that, since $(z_h^i, a_h^i)$ is drawn according to the policy $\widehat{\pi}^h$, the distribution of $s_h$, conditioned on the observation-action sequence $(a_{\max\{1, h-L\}:h-1}, o_{\max\{2, h-L+1\}:h})$, is exactly $\widetilde{\mathbf{b}}_h^{\pi^h}(a_{h-L:h-1}, o_{h-L+1:h})$. Thus, given that action $a_h$ is taken at step $h$, the probability that the next observation is $o_{h+1}$ is given by $e_{o_{h+1}}^\top \cdot \mathbb{O}_{h+1} \cdot \mathbb{T}_h(a_h) \cdot \widetilde{\mathbf{b}}_h^{\pi^h}(a_{h-L:h-1}, o_{H-L+1:h})$, which, by definition, is exactly $\mathbb{P}_h^{\widetilde{\mathcal{M}}(\pi^{1:H})}((a_{h-L+1:h}, o_{h-L+2:h+1})|z_h, a_h)$.

Consider any $z_h \in \mathcal{Z}$ with $z_h \notin \mathcal{Z}_{h,\zeta}^{\text{low}}(\widehat{\pi}^h)$ and any $a_h \in \mathcal{A}$. By the Chernoff bound, with probability at least $1 - \exp(-\zeta N_0/(8A))$, there are at least $\frac{\zeta N_0}{2A} \geq N_1$ iterations $i \in [N_0]$ (in the for loop in step 4 of Algorithm 1) so that $a_h^i = a_h$ and $\text{suff}_{h-1}(z_h^i) = \text{suff}_{h-1}(z_h)$. Under such an event, since the reward function $R^{\mathcal{P}}$ of $\mathcal{P}$ is deterministic, it must be the case that $R_h^{\widehat{\mathcal{M}}(\pi^{1:H})}(\mathsf{o}(z_h)) = R_h^{\mathcal{P}}(\mathsf{o}(z_h))$ (step 13 of Algorithm 1).

Next, consider any $(z_h, a_h) \in \mathcal{Z} \times \mathcal{A}$ so that there are at least $N_1$ values of $i$ so that $a_h^i = a_h$ and $\text{suff}_{h-1}(z_h^i) = \text{suff}_{h-1}(z_h)$ (i.e., for the chosen value of $(z_h, a_h)$, the **if** statement at step 11 holds). By the fact established in the previous paragraph and [Can20, Theorem 1], with probability at least $1 - p'$, (12) holds as long as

$$N_1 \geq C \cdot \frac{|\mathcal{Z}| + \log(1/p')}{\theta^2} \tag{13}$$

for some constant $C > 1$. Thus, by a union bound over all $h \in [H]$, and all $(z_h, a_h) \in \mathcal{Z} \times \mathcal{A}$, under some event $\mathcal{E}^{\text{low}}$ that occurs with probability at least

$$1 - |\mathcal{Z}|A \cdot \exp(-\zeta N_0/(8A)) - |\mathcal{Z}|A \cdot p', \tag{14}$$

we have that both:

1. For all $h \in [H]$, $a_h \in \mathcal{A}$, and $z_h \notin \mathcal{Z}_{h,\zeta}^{\text{low}}(\widehat{\pi}^h)$, there are at least $N_1$ values of $i$ (in the $h$th step of the loop in step 4) so that $(a_h^i = a_h)$ and $\text{suff}_{h-1}(z_h^i) = \text{suff}_{h-1}(z_h)$;

2. For all $h \in [H]$, $a_h \in \mathcal{Z}$, $z_h \in \mathcal{Z}$ for which there are at least $N_1$ values of $i$ so that $(a_h^i = a_h)$ and $\text{suff}_{h-1}(z_h^i) = \text{suff}_{h-1}(z_h)$, (12) holds; and

3. Thus, if (12) does not hold, we must have that there are fewer than $N_1$ values of $i$ so that $(z_h^i, a_h^i) = (z_h, a_h)$, which means, by step 15, that $z_h = (a_{h-L:h-1}, o_{h-L+1:h})$ transitions to $(a_{h-L+1:h}, (o_{h-L+2:o^{\text{sink}}}))$ with probability 1.

The above points verify the two points in the lemma statement.

Finally we must verify that our settings $N_0, N_1$ satisfy the above constraints and lead to the probability in (14) being at least $1 - p$. In particular, we choose $p' = p/(2|\mathcal{Z}|A)$, so that (13) holds as long as we set

$$N_1 = \frac{C_0 L A^{L+1} O^L \log(AO/p)}{\theta^2},$$

for sufficiently large constant $C_0$. Finally, note that $|\mathcal{Z}|A \cdot \exp(-\zeta N_0/(8A)) \leq p/2$ if

$$N_0 > \log\left(\frac{2|\mathcal{Z}|A}{p}\right) \cdot \frac{8A}{\zeta},$$

which holds as long as the constant $C_0$ in (11) is sufficiently large. Thus, the probability of $\mathcal{E}^{\text{low}}$ in (14) is at least $1 - p$, as desired. $\qquad\square$

## E.3 Construction of Truncated POMDPs $\overline{\mathcal{P}}_{\phi,h}(\pi^{1:H})$.

One of the main technical challenges is accounting for states of $\mathcal{P}$ which the current policy set *underexplores*, i.e. fails to explore with non-trivial probability. To deal with this issue, given the POMDP $\mathcal{P}$ and a sequence of policies $\pi^{1:H}$, we introduce the notion of *truncated POMDPs*. We start with a description of how to construct a sequence of truncated POMDPs $\overline{\mathcal{P}}_{\phi,h}(\pi^{1:H})$ for $h \in [H]$, which will aid in our analysis relating $\mathcal{P}$ and $\widehat{\mathcal{M}}^{(k)}$. Note that (like the MDPs $\widetilde{\mathcal{M}}(\pi^{1:H})$) truncated POMDPs do not appear in our algorithms; they are solely an analytic tool.

We begin by making the following definition of underexplored sets of states.

**Definition E.2.** Given a real number $\phi \in (0,1)$, a POMDP $\mathcal{P}$, and a general policy $\pi'$, define the $\phi$-*underexplored set for* $\pi'$, denoted $\mathcal{U}_\phi^{\mathcal{P}}(\pi')$, to be

$$\mathcal{U}_\phi^{\mathcal{P}}(\pi') := \left\{ (h,s) \in [H] \times \mathcal{S} \ : \ d_{\mathsf{S},h}^{\mathcal{P},\pi'}(s) < \phi \right\}.$$

For some $h \in [H]$, we also define the $\phi$-*underexplored set at step* $h$, as:

$$\mathcal{U}_{\phi,h}^{\mathcal{P}}(\pi') := \left\{ s \in \mathcal{S} \ : \ (h,s) \in \mathcal{U}_\phi^{\mathcal{P}}(\pi') \right\}.$$

In words, $\mathcal{U}_\phi^{\mathcal{P}}(\pi')$ is the set of pairs $(h,s) \in [H] \times \mathcal{S}$ so that $\pi'$ does not put sufficient mass on state $s$ at step $h$.

Given a sequence of general policies $\pi^{1:H}$ as above together with a parameter $\phi > 0$ representing a threshold visitation probability and some $H' \in [H]$, we proceed to describe how to construct a POMDP $\overline{\mathcal{P}}_{\phi,H'}(\pi^{1:H}) = (H, \overline{\mathcal{S}}, \mathcal{A}, \overline{\mathcal{O}}, b_1, \overline{\mathbb{T}}^{H'}, \overline{\mathbb{O}}^{H'})$, whose observation space is $\overline{\mathcal{O}} = \mathcal{O} \cup \{o^{\mathrm{sink}}\}$, and whose state space is $\overline{\mathcal{S}} = \mathcal{S} \cup \{s^{\mathrm{sink}}\}$. The observation matrices of $\overline{\mathcal{P}}_{\phi,H'}(\pi^{1:H})$, denoted by $\overline{\mathbb{O}}_h^{H'}$, are equal to those of $\mathcal{P}$, i.e., we have $\overline{\mathbb{O}}_h^{H'} = \mathbb{O}_h^{\mathcal{P}}$ for all $h \in [H]$. The state transitions of $\overline{\mathcal{P}}_{\phi,H'}(\pi^{1:H})$, denoted by $\overline{\mathbb{T}}^{H'}$, are constructed inductively (with respect to $H'$) according to the following procedure:

1. First set $\overline{\mathcal{P}}_{\phi,1}(\pi^{1:H}), \ldots, \overline{\mathcal{P}}_{\phi,L}(\pi^{1:H})$ to have identical transitions $\overline{\mathbb{T}}^1, \ldots, \overline{\mathbb{T}}^L$ (respectively) to those of $\mathcal{P}$ out of any state $s \in \overline{\mathcal{S}}$ at each step $h \in [H]$.

2. For $H' = L+1, L+2, \ldots, H$:

   (a) Define the transition matrices $\overline{\mathbb{T}}^{H'}$ of $\overline{\mathcal{P}}_{\phi,H'}(\pi^{1:H})$ as follows:

   (b) Initially set $\overline{\mathbb{T}}^{H'} \leftarrow \overline{\mathbb{T}}^{H'-1}$.

   (c) Then, for each state $s \in \mathcal{U}_{\phi,H'-L}^{\overline{\mathcal{P}}_{\phi,H'-1}(\pi^{1:H})}(\pi^{H'})$:

   - For all $s' \in \mathcal{S}$ and $a \in \mathcal{A}$, add $\mathbb{T}_{H'-L-1}^{H'}(s|s',a)$ to $\overline{\mathbb{T}}_{H'-L-1}^{H'}(s^{\mathrm{sink}}|s',a)$ and set $\overline{\mathbb{T}}_{H'-L-1}^{H'}(s|s',a)$ to 0.

Note that for all $H'$, the above construction ensures that (at the end of the above procedure) $\overline{\mathbb{T}}_h^{H'}(s^{\mathrm{sink}}|s^{\mathrm{sink}}, a) = 1$ for all $h \in [H], a \in \mathcal{A}$.

The main point of the construction of the truncated POMDPs $\overline{\mathcal{P}}_{\phi,H'}(\pi^{1:H})$ is to ensure that the following lemma holds:

**Lemma E.2.** *Consider a sequence of general policies $\pi^{1:H}$ and $\phi > 0$. For each $H' \in [H]$, the truncated POMDP $\overline{\mathcal{P}}_{\phi,H'}(\pi^{1:H})$ satisfies the following: for all $(s,h) \in \mathcal{S} \times [H]$ with $L < h \le H'$, if there is some policy $\pi \in \Pi^{\mathrm{gen}}$ with $d_{\mathsf{S},h-L}^{\overline{\mathcal{P}}_{\phi,H'}(\pi^{1:H}),\pi}(s) > 0$, then $d_{\mathsf{S},h-L}^{\overline{\mathcal{P}}_{\phi,H'}(\pi^{1:H}),\pi^h}(s) \ge \phi$, i.e., $s \notin \mathcal{U}_{\phi,h-L}^{\overline{\mathcal{P}}_{\phi,H'}(\pi^{1:H})}(\pi^h)$.*

*Proof.* For each $h \in [H]$ let $\overline{\mathbb{T}}^h$ denote the state transitions of $\overline{\mathcal{P}}_{\phi,h}(\pi^{1:H})$.

Since the transitions of $\overline{\mathcal{P}}_{\phi,H'}(\pi^{1:H})$ and $\overline{\mathcal{P}}_{\phi,h}(\pi^{1:H})$ do not differ prior to step $h$, it suffices to consider the case that $H' = h$. So suppose that $d_{\mathsf{S},h-L}^{\overline{\mathcal{P}}_{\phi,h}(\pi^{1:H}),\pi}(s) > 0$ for some $\pi \in \Pi^{\mathrm{gen}}$. Then

by construction of $\overline{\mathcal{P}}_{\phi,h}(\pi^{1:H})$ it must be the case that $s \notin \mathcal{U}_{\phi,h-L}^{\overline{\mathcal{P}}_{\phi,h-1}(\pi^{1:H})}(\pi^h)$. Since $\mathbb{T}^h$ and $\mathbb{T}^{h-1}$ only differ in the transitions at step $h-L-1$ and furthermore for all $s' \in \overline{\mathcal{S}}$ and $a \in \mathcal{A}$, $\overline{\mathbb{T}}_{h-L-1}^{h-1}(s|s',a) = \overline{\mathbb{T}}_{h-L-1}^h(s|s',a)$, we have that $d_{\mathsf{S},h-L}^{\overline{\mathcal{P}}_{\phi,h-1}(\pi^{1:H}),\pi^h}(s) = d_{\mathsf{S},h-L}^{\overline{\mathcal{P}}_{\phi,h}(\pi^{1:H}),\pi^h}(s)$. Thus $s \notin \mathcal{U}_{\phi,h-L}^{\overline{\mathcal{P}}_{\phi,h}(\pi^{1:H})}(\pi^h)$, as desired. $\qquad\square$

## F    Generic truncated POMDPs

We next define a generic notion of truncation (of a POMDP $\mathcal{P}$), which is satisfied by the truncated POMDPs $\overline{\mathcal{P}}_{\phi,h}(\pi^{1:H})$ as constructed in the previous section (Lemma F.1 below).

**Definition F.1.** We say that a POMDP $\overline{\mathcal{P}}$ with extended state and observation spaces $\overline{\mathcal{S}}, \overline{\mathcal{O}}$ is a *truncation* of $\mathcal{P}$ if for all general policies $\pi \in \Pi^{\mathrm{gen}}$, $h \in [H]$, $a'_{1:h-1} \in \mathcal{A}^{h-1}, o'_{2:h} \in \mathcal{O}^h, s'_{1:h} \in \mathcal{S}^h$, it holds that

$$\mathbb{P}_{(a_{1:h-1},s_{1:h},o_{2:h})\sim\pi}^{\overline{\mathcal{P}}}((a_{1:h-1},s_{1:h},o_{2:h}) = (a'_{1:h-1},s'_{1:h},o'_{2:h}))$$
$$\leq \mathbb{P}_{(a_{1:h-1},s_{1:h},o_{2:h})\sim\pi}^{\mathcal{P}}((a_{1:h-1},s_{1:h},o_{2:h}) = (a'_{1:h-1},s'_{1:h},o'_{2:h})). \tag{15}$$

Furthermore, we require that $\mathbb{T}_h^{\overline{\mathcal{P}}}(s^{\mathrm{sink}}|s^{\mathrm{sink}},a) = 1$ for all $h \in [H]$ and $a \in \mathcal{A}$.

Note that an immediate consequence of (15) is that for all $(h,s) \in [H] \times \mathcal{S}$, $d_{\mathsf{S},h}^{\mathcal{P},\pi}(s) \geq d_{\mathsf{S},h}^{\overline{\mathcal{P}},\pi}(s)$.

**Lemma F.1.** *Fix any sequence of general policies $\pi^{1:H}$ and $\phi > 0$. Then for each $H', H'' \in [H]$ satisfying $H'' \leq H'$, the POMDP $\overline{\mathcal{P}}_{\phi,H'}(\pi^{1:H})$ as constructed above is a truncation of $\overline{\mathcal{P}}_{\phi,H''}(\pi^{1:H})$ (per Definition F.1).*

Since $\overline{\mathcal{P}}_{\phi,1}(\pi^{1:H}) = \overline{\mathcal{P}}$, an immediate consequence of Lemma F.1 is that for any $H' \in [H]$, $\overline{\mathcal{P}}_{\phi,H'}(\pi^{1:H})$ is a truncation of $\mathcal{P}$.

*Proof of Lemma F.1.* Fix any $H'' \leq H'$. Write $\overline{\mathcal{P}}_{H'} = \overline{\mathcal{P}}_{\phi,H'}(\pi^{1:H})$ and $\overline{\mathcal{P}}_{H''} = \overline{\mathcal{P}}_{\phi,H''}(\pi^{1:H})$. By convexity it suffices to consider the case that $\pi$ is a deterministic policy, i.e., $\pi \in \Pi^{\mathrm{det}}$. The proof uses induction on $h$ to verify (15), noting that the base case $h = 1$ is immediate since the initial state distributions of $\overline{\mathcal{P}}_{H''}$ and $\overline{\mathcal{P}}_{H'}$ are identical.

Fix any $h \in [H]$, and suppose that (15) holds at all steps up to (and including) step $h$. Let $\tau = (s_1,a_1,s_2,o_2,a_2,\ldots,s_{h-1},o_{h-1},a_{h-1},s_h,o_h)$ denote a trajectory drawn according to the policy $\pi$ up to step $h$. Our inductive assumption gives that $\mathbb{P}_\pi^{\overline{\mathcal{P}}_{H'}}(\tau) \leq \mathbb{P}_\pi^{\overline{\mathcal{P}}_{H''}}(\tau)$. Since $\pi$ is a deterministic policy, the action chosen by $\pi$ at step $h$ is given by $\pi_h(a_{1:h-1},o_{2:h})$, i.e., it is a function of $\tau$. Furthermore, for any $a \in \mathcal{A}$ and $s' \in \mathcal{S}$, we have that $\mathbb{T}_h^{\overline{\mathcal{P}}_{H''}}(s'|s_h,a) \geq \mathbb{T}_h^{\overline{\mathcal{P}}_{H'}}(s'|s_h,a)$ by definition of $\overline{\mathcal{P}}_{H'}, \overline{\mathcal{P}}_{H''}$. Since furthermore $\mathbb{O}_{h+1}^{\overline{\mathcal{P}}_{H''}}(o|s') = \mathbb{O}_{h+1}^{\overline{\mathcal{P}}_{H'}}(o|s')$ for all $o \in \overline{\mathcal{O}}$, $s' \in \overline{\mathcal{S}}$, we get that (15) holds at step $h + 1$, as desired. $\qquad\square$

The below lemma establishes some additional properties of the truncated POMDPS $\overline{\mathcal{P}}_{\phi,h}(\pi^{1:H})$.

**Lemma F.2.** *Fix any sequence of general policies $\pi^{1:H}$ and $\phi > 0$. Then for each $h \in [H]$, we have the following:*

1. *For any general policy $\pi \in \Pi^{\mathrm{gen}}$ and all $(h,s) \in [H] \times \mathcal{S}$ with $h > L$, it holds that*

$$d_{\mathsf{S},h-L}^{\pi,\mathcal{P}}(s) \leq d_{\mathsf{S},h-L}^{\pi,\overline{\mathcal{P}}_{\phi,h}(\pi^{1:H})}(s) + d_{\mathsf{S},h-L}^{\pi,\overline{\mathcal{P}}_{\phi,h}(\pi^{1:H})}(s^{\mathrm{sink}}).$$

2. *For any $H', H'' \in [H]$ satisfying $H'' \leq H'$, and any $(s,h) \in [H] \times \mathcal{S}$, it holds that $d_{\mathsf{S},h}^{\pi,\overline{\mathcal{P}}_{\phi,H'}(\pi^{1:H})}(s) \leq d_{\mathsf{S},h}^{\overline{\mathcal{P}}_{\phi,H''}(\pi^{1:H}),\pi}(s)$, and thus $d_{\mathsf{S},h}^{\overline{\mathcal{P}}_{\phi,H''}(\pi^{1:H}),\pi}(s^{\mathrm{sink}}) \leq d_{\mathsf{S},h}^{\overline{\mathcal{P}}_{\phi,H'}(\pi^{1:H}),\pi}(s^{\mathrm{sink}})$.*

3. *For any $h,h' \in [H]$ satisfying $h' \leq h$, as well as any $H' \in [H]$, it holds that $d_{\mathsf{S},h'}^{\overline{\mathcal{P}}_{\phi,H'}(\pi^{1:H}),\pi}(s^{\mathrm{sink}}) \leq d_{\mathsf{S},h}^{\overline{\mathcal{P}}_{\phi,H'}(\pi^{1:H}),\pi}(s^{\mathrm{sink}})$.*

*Proof.* For $h \in [H]$, write $\overline{\mathcal{P}}_h := \overline{\mathcal{P}}_{\phi,h}(\pi^{1:H})$. We begin with a proof of the first item. By linearity of expectation it suffices to consider the case that $\pi$ is a deterministic policy, i.e., we have $\pi = (\pi_1, \ldots, \pi_H)$, where each $\pi_h : \mathcal{A}^{h-1} \times \mathcal{O}^{h-1} \to \mathcal{A}$. Consider any trajectory up to step $h - L$, denoted $\tau = (s_1, a_1, s_2, o_2, a_2, \ldots, s_{h-L}, o_{h-L})$, which has positive probability under $\mathcal{P}$ when policy $\pi$ is used. For such a trajectory $\tau$ and $h' \leq h - L$, let $s_{h'}(\tau)$ denote the state in $\tau$ at step $h'$ (i.e., the state $s_{h'}$). By the construction of $\overline{\mathcal{P}}_h$, if $\tau$ also has positive probability under $\overline{\mathcal{P}}_h$ for the policy $\pi$, then $\mathbb{P}_\pi^{\mathcal{P}}(\tau) = \mathbb{P}_\pi^{\overline{\mathcal{P}}_h}(\tau)$. On the other hand, it is evident from construction of $\overline{\mathcal{P}}_h = \overline{\mathcal{P}}_{\phi,h}(\pi^{1:H})$ that

$$d_{\mathsf{S},h-L}^{\overline{\mathcal{P}}_h,\pi}(s^{\mathrm{sink}}) = \sum_{\tau:\mathbb{P}_\pi^{\overline{\mathcal{P}}_h}(\tau)=0,\ s_{h-L}(\tau)\in\mathcal{S}} \mathbb{P}_\pi^{\mathcal{P}}(\tau). \tag{16}$$

To see that the above equality holds, consider any trajectory $\tau = (s_1, a_1, s_2, o_2, a_2, \ldots, s_{h-L}, o_{h-L})$ with $s_{h-L} = s^{\mathrm{sink}}$ and which occurs with positive probability under $\overline{\mathcal{P}}_h, \pi$. Choose $h'$ as small as possible so that $s_{h'} = s^{\mathrm{sink}}$. We associate $\tau$ with the set $\mathcal{T}(\tau)$ of trajectories that agree with $\tau$ up to step $h' - 1$, and then transition to a state $\widetilde{s}_{h'}$ from which mass was diverted away in the construction of $\overline{\mathcal{P}}_{h'+L}$ (to be precise, the set of such states is $\mathcal{U}_{\phi,h'}^{\mathcal{P}_{h'+L-1}}(\pi^{h'+L})$). It is clear that all trajectories $\tau' \in \mathcal{T}(\tau)$ have $\mathbb{P}_\pi^{\overline{\mathcal{P}}_h}(\tau') = 0$, and moreover, $\mathbb{P}_\pi^{\overline{\mathcal{P}}_h}(\tau) = \sum_{\tau'\in\mathcal{T}(\tau)} \mathbb{P}_\pi^{\mathcal{P}}(\tau)$. Furthermore, any trajectory $\tau'$ with $s_{h-L}(\tau') \in \mathcal{S}$, $\mathbb{P}_\pi^{\mathcal{P}}(\tau') > 0$, and $\mathbb{P}_\pi^{\overline{\mathcal{P}}_h}(\tau') = 0$ must be in some (unique) $\mathcal{T}(\tau)$, for some $\tau$ with $s_{h-L}(\tau) = s^{\mathrm{sink}}$. Combining these facts verifies (16).

Then we have

$$d_{\mathsf{S},h-L}^{\mathcal{P},\pi}(s) = \sum_{\tau:s_{h-L}(\tau)=s} \mathbb{P}_\pi^{\mathcal{P}}(\tau) \leq \sum_{\tau:s_{h-L}(\tau)=s} \mathbb{P}_\pi^{\overline{\mathcal{P}}_h}(\tau) + d_{\mathsf{S},h-L}^{\overline{\mathcal{P}}_h,\pi}(s^{\mathrm{sink}})$$
$$= d_{\mathsf{S},h-L}^{\overline{\mathcal{P}}_h,\pi}(s) + d_{\mathsf{S},h-L}^{\overline{\mathcal{P}}_h,\pi}(s^{\mathrm{sink}}), \tag{17}$$

which completes the proof of the first item.

The second item of the lemma statement is a direct consequence of the fact that $\overline{\mathcal{P}}_{\phi,H'}(\pi^{1:H})$ is a truncation of $\overline{\mathcal{P}}_{\phi,H''}(\pi^{1:H})$ (Lemma F.1).

Finally, the third item of the lemma statement follows from the fact that for all $h \in [H]$ and $a \in \mathcal{A}$, $\mathbb{T}_h^{\overline{\mathcal{P}}_{\phi,H'}(\pi^{1:H})}(s^{\mathrm{sink}}|s^{\mathrm{sink}}, a) = 1$. $\square$

For a non-negative vector $\mathbf{v} \in \mathbb{R}_{\geq 0}^d$, we define $\mathfrak{n}(\mathbf{v}) \in \Delta([d])$ by, for $i \in [d]$, $\mathfrak{n}(\mathbf{v})(i) = \frac{\mathbf{v}(i)}{\sum_{j=1}^d \mathbf{v}(j)}$. Lemma F.3 below can be viewed as a "one-sided" corollary of the belief contraction result in Theorem B.2, applied to a truncated POMDP $\overline{\mathcal{P}}$. It makes a somewhat weaker assumption on the distribution $\mathscr{D}$ in Theorem B.2 (for a particular type of choice for $\mathscr{D}$), at the cost of only obtaining a one-sided inequality on the difference between the true beliefs $\mathbf{b}_h^{\overline{\mathcal{P}}}$ and the approximate beliefs $\widetilde{\mathbf{b}}_h^{\pi'}$.

**Lemma F.3.** *Fix any $\phi > 0$, $h \in [H]$, POMDP $\overline{\mathcal{P}}$ with extended state and observation spaces $\overline{\mathcal{S}}, \overline{\mathcal{O}}$ which is a truncation of $\mathcal{P}$, and general policy $\pi' \in \Pi^{\mathrm{gen}}$. Suppose that $\mathbb{T}_{h'}^{\overline{\mathcal{P}}} = \mathbb{T}_{h'}^{\mathcal{P}}$ for $h' \geq h - L$ and $\mathbb{O}_{h'}^{\overline{\mathcal{P}}} = \mathbb{O}_{h'}^{\mathcal{P}}$ for all $h' \in [H]$. Suppose further that $\pi'$ satisfies the following: for each $s \in \mathcal{S}$, if $h > L$ and there is some policy $\pi \in \Pi^{\mathrm{gen}}$ so that $d_{\mathsf{S},h-L}^{\overline{\mathcal{P}},\pi}(s) > 0$, then $s \notin \mathcal{U}_{\phi,h-L}^{\overline{\mathcal{P}}}(\pi')$. Then for general policies $\pi$, it holds that*

$$\mathbb{E}_{a_{1:h-1},o_{2:h}\sim\pi}^{\overline{\mathcal{P}}} \left[ \sum_{s\in\mathcal{S}} \left| \mathbf{b}_h^{\overline{\mathcal{P}}}(a_{1:h-1}, o_{2:h})(s) - \widetilde{\mathbf{b}}_h^{\pi'}(a_{h-L:h-1}, o_{h-L+1:h})(s) \right| \right] \leq \epsilon.$$

*Proof of Lemma F.3.* Set $\underline{h} := \max\{h - L, 1\}$. Furthermore, define $\mathscr{D} = d_{\mathsf{S},\underline{h}}^{\mathcal{P},\pi'}$. Note that, if $\underline{h} = 1$, then we have $\mathscr{D} = b_1$, the initial state distribution of $\mathcal{P}$. By definition of $\widetilde{\mathbf{b}}_h^{\pi'}$ and by our assumption that $\mathbb{T}_{h'}^{\overline{\mathcal{P}}} = \mathbb{T}_{h'}^{\mathcal{P}}$ and $\mathbb{O}_{h'+1}^{\overline{\mathcal{P}}} = \mathbb{O}_{h'+1}^{\mathcal{P}}$ for all $h' \geq h - L$, we have that

$$\widetilde{\mathbf{b}}_h^{\pi'}(a_{\underline{h}:h-1}, o_{\underline{h}+1:h}) = \mathbf{b}_h^{\mathrm{apx},\mathcal{P}}(a_{\underline{h}:h-1}, o_{\underline{h}+1:h}; \mathscr{D}) = \mathbf{b}_h^{\mathrm{apx},\overline{\mathcal{P}}}(a_{\underline{h}:h-1}, o_{\underline{h}+1:h}; \mathscr{D}).$$

Consider any general policy $\pi$, and fix any sequence $(a_{1:\underline{h}-1}, o_{2:\underline{h}}) \in (\mathcal{A} \times \mathcal{O})^{\underline{h}-1}$ that occurs with positive probability under $(\overline{\mathcal{P}}, \pi)$. If $o_{\underline{h}} = o^{\text{sink}}$, then for all sequences $(a_{\underline{h}:h-1}, o_{\underline{h}+1:h})$ that occur with positive probability under $(\overline{\mathcal{P}}, \pi)$ conditioned on $(a_{1:\underline{h}-1}, o_{2:\underline{h}})$ (which in particular implies that $o_{h'} = o^{\text{sink}}$ for all $\underline{h} + 1 \leq h' \leq h$), we have

$$\mathbf{b}_h^{\overline{\mathcal{P}}}(a_{1:h-1}, o_{2:h})(s) = \widetilde{\mathbf{b}}_h^{\pi'}(a_{\underline{h}:h-1}, o_{\underline{h}+1:h})(s) = 0$$

for all $s \in \mathcal{S}$. This uses our convention to define $\widetilde{\mathbf{b}}_h^{\pi'}(a_{\underline{h}:h-1}, o_{\underline{h}+1:h})(s^{\text{sink}}) = 1$ (see Section B) when any of $o_{\underline{h}+1}, \ldots, o_h$ are equal to $o^{\text{sink}}$.

Next suppose that $o_{\underline{h}} \neq o^{\text{sink}}$; then $\mathbf{b}_h^{\overline{\mathcal{P}}}(a_{1:\underline{h}-1}, o_{2:\underline{h}})(s^{\text{sink}}) = 0$. If $\underline{h} > 1$, then for any $s \in \overline{\mathcal{S}}$ so that $\mathbf{b}_{\underline{h}}^{\overline{\mathcal{P}}}(a_{1:\underline{h}-1}, o_{2:\underline{h}})(s) > 0$, we must have that $d_{\mathsf{S},\underline{h}}^{\overline{\mathcal{P}},\pi}(s) > 0$ and $s \in \mathcal{S}$, and therefore, $s \notin \mathcal{U}_{\phi,\underline{h}}^{\overline{\mathcal{P}}}(\pi')$ (by assumption), i.e., $\mathscr{D}(s) = d_{\mathsf{S},\underline{h}}^{\mathcal{P},\pi'}(s) \geq d_{\mathsf{S},\underline{h}}^{\overline{\mathcal{P}},\pi'}(s) \geq \phi$. On the other hand, if $\underline{h} = 1$, it holds that $\mathscr{D} = b_1 = \mathbf{b}_1^{\overline{\mathcal{P}}}(\emptyset)$. Therefore, by Theorem B.2 applied to the POMDP $\overline{\mathcal{P}}$, we have

$$\mathbb{E}_{\substack{s_{\underline{h}} \sim \mathbf{b}_{\underline{h}}^{\overline{\mathcal{P}}}(a_{1:\underline{h}-1}, o_{2:\underline{h}}), \\ a_{\underline{h}:h-1}, o_{\underline{h}+1:h} \sim \pi}}^{\overline{\mathcal{P}}} \left[ \left\| \mathbf{b}_h^{\overline{\mathcal{P}}}(a_{1:h-1}, o_{2:h}) - \widetilde{\mathbf{b}}_h^{\pi'}(a_{\underline{h}:h-1}, o_{\underline{h}+1:h}) \right\|_1 \right] \leq \epsilon.$$

Taking expectation over $(a_{1:\underline{h}-1}, o_{2:\underline{h}}) \sim \pi$ under the POMDP $\overline{\mathcal{P}}$, and using the fact that the conditional distribution of $s_{\underline{h}}$ given $a_{1:\underline{h}-1}, o_{2:\underline{h}}$ is exactly $\mathbf{b}_{\underline{h}}^{\overline{\mathcal{P}}}(a_{1:\underline{h}-1}, o_{2:\underline{h}})$, we get

$$\mathbb{E}_{a_{1:h-1}, o_{2:h} \sim \pi}^{\overline{\mathcal{P}}} \left[ \sum_{s \in \mathcal{S}} \left| \mathbf{b}_h^{\overline{\mathcal{P}}}(a_{1:h-1}, o_{2:h})(s) - \widetilde{\mathbf{b}}_h^{\pi'}(a_{\underline{h}:h-1}, o_{\underline{h}+1:h})(s) \right| \right] \leq \epsilon. \tag{18}$$

as desired. $\square$

Recall that Lemma E.1 shows that the empirical MDP $\widehat{\mathcal{M}}(\pi^{1:H})$ is close to $\widetilde{\mathcal{M}}(\pi^{1:H})$ at $\mathcal{Z}$-states which are not in $\mathcal{Z}_{h,\zeta}^{\text{low}}(\pi')$, i.e., those which are not visited too infrequently under the dynamics of $\mathcal{P}$ (recall Definition E.1). Thus, to effectively apply Lemma E.1, we need to bound the probability of reaching such states; Lemma F.4 does so, with respect to the dynamics of any truncated POMDP $\overline{\mathcal{P}}$.

**Lemma F.4.** *Fix any $\zeta > 0$, $\delta \geq 0$, $h \leq H$ and consider any policies $\pi', \pi \in \Pi^{\text{gen}}$, as well as any POMDP $\overline{\mathcal{P}}$ with extended state and observation spaces $\overline{\mathcal{S}}, \overline{\mathcal{O}}$ which is a truncation of $\mathcal{P}$. Suppose that $\pi'$ takes uniformly random actions at each step $h' \in \{\max\{h - L, 1\}, \ldots, h\}$, each chosen independently of all previous states, actions, and observations $(s_{1:h'}, a_{1:h'-1}, o_{2:h'})$. Then*

$$d_{\mathsf{Z},h}^{\overline{\mathcal{P}},\pi}(\mathcal{Z}_{h,\zeta}^{\text{low}}(\pi')) \leq \frac{A^{2L} O^L \zeta}{\phi} + \mathbb{1}[h > L] \cdot d_{\mathsf{S},h-L}^{\overline{\mathcal{P}},\pi}(\mathcal{U}_{\phi,h-L}^{\overline{\mathcal{P}}}(\pi')).$$

*Proof.* By linearity of expectation, we may assume that $\pi$ is a deterministic policy, i.e., $\pi \in \Pi^{\text{det}}$.

Fix any $h$ as in the lemma statement, and write $\underline{h} := \max\{1, h - L\}$. By definition of $\mathcal{U}_{\phi,\underline{h}}^{\overline{\mathcal{P}}}(\pi')$, for any $s \notin \mathcal{U}_{\phi,\underline{h}}^{\overline{\mathcal{P}}}(\pi')$, it holds that $d_{\mathsf{S},\underline{h}}^{\overline{\mathcal{P}},\pi'}(s) \geq \phi$. For convenience we write

$$\mathcal{U} := \begin{cases} \mathcal{U}_{\phi,\underline{h}}^{\overline{\mathcal{P}}}(\pi') & : \underline{h} > 1 \\ \emptyset & : \underline{h} = 1. \end{cases}$$

throughout the proof of the lemma.

Suppose $\underline{h} = 1$: since $d_{\mathsf{S},\underline{h}}^{\overline{\mathcal{P}},\pi}(s) \leq 1$ for all $s \in \mathcal{S}$, it then holds that $\frac{d_{\mathsf{S},\underline{h}}^{\overline{\mathcal{P}},\pi}(s)}{d_{\mathsf{S},\underline{h}}^{\overline{\mathcal{P}},\pi'}(s)} \leq \frac{1}{\phi}$ for all $s \in \mathcal{S} \backslash \mathcal{U}$. On the other hand, when $\underline{h} = 1$, it holds that $\frac{d_{\mathsf{S},h}^{\overline{\mathcal{P}},\pi}(s)}{d_{\mathsf{S},\underline{h}}^{\overline{\mathcal{P}},\pi'}(s)} = 1 \leq \frac{1}{\phi}$ for *all* $s \in \mathcal{S} = \mathcal{S} \backslash \mathcal{U}$.

We next prove by induction that for all $h'$ satisfying $\underline{h} \leq h' \leq h$, it holds that

$$\substack{\forall a'_{\underline{h}:h'-1}, o'_{\underline{h}+1:h'}, s'_{\underline{h}:h'} \\ \text{such that } s'_{\underline{h}} \notin \mathcal{U}}, \quad \frac{\mathbb{P}_\pi^{\overline{\mathcal{P}}}((a_{\underline{h}:h'-1}, o_{\underline{h}+1:h'}, s_{\underline{h}:h'}) = (a'_{\underline{h}:h'-1}, o'_{\underline{h}+1:h'}, s'_{\underline{h}:h'}))}{\mathbb{P}_{\pi'}^{\overline{\mathcal{P}}}((a_{\underline{h}:h'-1}, o_{\underline{h}+1:h'}, s_{\underline{h}:h'}) = (a'_{\underline{h}:h'-1}, o'_{\underline{h}+1:h'}, s'_{\underline{h}:h'}))} \leq \frac{A^{h'-\underline{h}}}{\phi}. \tag{19}$$

We first note that (19) holds for $h' = \underline{h}$ because $\frac{d_{\mathsf{S},\underline{h}}^{\pi,\overline{\mathcal{P}}}(s)}{d_{\mathsf{S},\underline{h}}^{\pi',\overline{\mathcal{P}}}(s)} \leq \frac{1}{\phi}$ for all $s \in \mathcal{S}\backslash\mathcal{U}$.

If (19) holds at some step $h'$, then we proceed to show that (19) holds at step $h' + 1$: conditioned on any history $a_{1:h'-1}, o_{2:h'}, s_{\underline{h}:h'}$, we have by assumption that $\pi'$ chooses a uniformly random action at step $h'$, meaning that with probability $1/A$, $\pi'$ chooses the action $\pi_{h'}(a_{a:h'-1}, o_{2:h'})$ at step $h'$. (Recall that we have assumed that $\pi$ is a deterministic policy.) Thus, we have that for all $a' \in \mathcal{A}$, $s' \in \mathcal{S}$, $o' \in \mathcal{O}$,

$$\frac{\mathbb{P}_\pi^{\overline{\mathcal{P}}}(a_{h'} = a', s_{h'+1} = s', o_{h'+1} = o'|a_{\underline{h}:h'-1}, o_{\underline{h}+1:h'}, s_{\underline{h}:h'})}{\mathbb{P}_{\pi'}^{\overline{\mathcal{P}}}(a_{h'} = a', s_{h'+1} = s', o_{h'+1} = o'|a_{\underline{h}:h'-1}, o_{\underline{h}+1:h'}, s_{\underline{h}:h'})} \leq A.$$

Then using (19) at step $h'$ gives (19) at step $h' + 1$.

Given the validity of (19) at step $h' = h$, we sum over all possible values of $s'_{\underline{h}:h} \in \mathcal{S}^{L+1}$ and $(a'_{h-L:h-1}, o'_{h-L+1:h}) \in \mathcal{Z}_{h,\zeta}^{\text{low}}(\pi')$ to obtain that

$$\sum_{(a'_{h-L:h-1}, o'_{h-L+1:h}) \in \mathcal{Z}_{h,\zeta}^{\text{low}}(\pi')} \mathbb{P}_\pi^{\overline{\mathcal{P}}}((a_{\underline{h}:h-1}, o_{\underline{h}+1:h}) = (a'_{\underline{h}:h-1}, o'_{\underline{h}+1:h}))$$

$$\leq \sum_{(a'_{h-L:h-1}, o'_{h-L+1:h}) \in \mathcal{Z}_{h,\zeta}^{\text{low}}(\pi')} \mathbb{P}_{\pi'}^{\overline{\mathcal{P}}}((a_{\underline{h}:h-1}, o_{\underline{h}+1:h}) = (a'_{\underline{h}:h-1}, o'_{\underline{h}+1:h})) \cdot \frac{A^L}{\phi} + \mathbb{P}_\pi^{\overline{\mathcal{P}}}(s_{\underline{h}} \in \mathcal{U}).$$

$$(20)$$

(Note that in the case that $h - L \leq 0$, we are multi-counting each value of $(a'_{\underline{h}:h-1}, o'_{\underline{h}+1:h})$ on both sides of the equation.) Consider any $z = (a'_{h-L:h-1}, o'_{h-L+1:h}) \in \mathcal{Z}_{h,\zeta}^{\text{low}}(\pi')$. Since $\overline{\mathcal{P}}$ is a truncation of $\mathcal{P}$ (Definition F.1), we have that

$$\mathbb{P}_{\pi'}^{\overline{\mathcal{P}}}((a_{\underline{h}:h-1}, o_{\underline{h}+1:h}) = (a'_{\underline{h}:h-1}, o'_{\underline{h}+1:h})) \leq \mathbb{P}_{\pi'}^{\mathcal{P}}((a_{\underline{h}:h-1}, o_{\underline{h}+1:h}) = (a'_{\underline{h}:h-1}, o'_{\underline{h}+1:h}))$$

$$= d_{\mathsf{Z},h}^{\mathcal{P},\pi'}(z) \leq \zeta.$$

Since $|\mathcal{Z}_{h,\zeta}^{\text{low}}(\pi')| \leq (OA)^L$, it follows from (20) that $d_{\mathsf{Z},h}^{\overline{\mathcal{P}},\pi}(\mathcal{Z}_{h,\zeta}^{\text{low}}(\pi')) \leq \frac{A^{2L}O^L\zeta}{\phi} + \sum_{s \in \mathcal{U}} d_{\mathsf{S},\underline{h}}^{\overline{\mathcal{P}},\pi}(s) \leq \frac{A^{2L}O^L\zeta}{\phi} + d_{\mathsf{S},\underline{h}}^{\overline{\mathcal{P}},\pi}(\mathcal{U})$, as desired (recall that for the case $\underline{h} = 1$, we have $\mathcal{U} = \emptyset$, so that $d_{\mathsf{S},\underline{h}}^{\overline{\mathcal{P}},\pi}(\mathcal{U}) = 0$). $\qquad\qquad\square$

# G    One-sided Comparison: Relating $\overline{\mathcal{P}}_{\phi,H'}(\pi^{1:H})$ and $\widehat{\mathcal{M}}(\pi^{1:H})$.

In this section we develop several bounds on the distance between the distributions of action-observation sequences under the truncated POMDPs $\overline{\mathcal{P}}_{\phi,H'}(\pi^{1:H})$ and the MDPs $\widehat{\mathcal{M}}^{(k)}$ constructed in the course of Algorithm 3. We will do this by relating both to the MDPs $\widetilde{\mathcal{M}}(\pi^{1:H})$ defined in Section E.1. Roughly speaking, we show that, if we attach a non-negative reward function to $\overline{\mathcal{P}}_{\phi,H'}(\pi^{1:H})$ and $\widehat{\mathcal{M}}(\pi^{1:H})$, then the value of any policy $\pi$ in $\overline{\mathcal{P}}_{\phi,H'}(\pi^{1:H})$ cannot be much greater than the value of $\pi$ in $\widehat{\mathcal{M}}(\pi^{1:H})$ (but the inequality in the other direction need not necessarily hold).

Our first lemma, Lemma G.1, can be viewed as a modification of Lemma B.1, which expresses the difference in value functions between two contextual decision processes in terms of the differences between their transitions and rewards. Lemma G.1 in particular applies to the case where one of the contextual decision processes is one of the truncated POMDPs $\overline{\mathcal{P}}_{\phi,H'-1}(\pi^{1:H})$, for some choice of $H', \phi, \pi^{1:H}$ (see Section E.3), the rewards are identical, and the value function is always approximately non-negative. In particular, Lemma G.1 derives an *upper bound* on the value function of this truncated POMDP; the one-sided nature of this upper bound should not be surprising in light of the fact that the truncated POMDPs divert some mass to the sink state, which has 0 value, while the value elsewhere is (approximately) non-negative. Importantly, the upper bound is phrased in terms of the difference between the transition functions of $\overline{\mathcal{P}}_{\phi,h}(\pi^{1:H})$ and $\mathcal{P}'$, for values of $h \leq H' - 1$; this particular detail is needed to be able to apply Lemma F.3 to further upper bound this expression.

**Lemma G.1.** *Fix the POMDP $\mathcal{P}$, and consider any reward-free contextual decision process $\mathcal{P}' = (\mathcal{A}, \overline{\mathcal{O}}, H, \mathbb{P})$. Fix any $H' \in [H]$, and consider any collection of reward functions $R = (R_1, \ldots, R_H)$, $R_h : \overline{\mathcal{O}} \to \mathbb{R}$ so that:*

- *$R_{h'} \equiv 0$ for all $h' \neq H'$;*

- *$R_{h'}(o^{\mathrm{sink}}) = 0$ for all $h' \in [H]$;*

- *For some $\psi > 0$, $V_h^{\pi, \mathcal{P}', R}(a_{1:h-1}, o_{2:h}) \geq -\psi$ for all $h \in [H]$ and $(a_{1:h-1}, o_{2:h}) \in \mathscr{H}_h$;*

- *For any policy $\pi \in \Pi^{\mathrm{gen}}$ and any trajectory $(a_{1:H-1}, o_{2:H})$ with positive probability under $(\pi, \mathcal{P}')$, if $o_h = o^{\mathrm{sink}}$ for some $h$, then $o_{h'} = o^{\mathrm{sink}}$ for all $h' > h$.*

*Consider any sequence of general policies $\pi^1, \ldots, \pi^H \in \Pi^{\mathrm{gen}}$ and $\phi > 0$. Then*

$$
V_1^{\pi, \overline{\mathcal{P}}_{H'-1}, R}(\emptyset) - V_1^{\pi, \mathcal{P}', R}(\emptyset) \leq \psi H + \sum_{h=1}^{H'-2} \mathbb{E}_{(a_{1:h}, o_{2:h}) \sim \pi}^{\overline{\mathcal{P}}_h} \left[ \left( \left( \mathbb{P}_h^{\overline{\mathcal{P}}_h} - \mathbb{P}_h^{\mathcal{P}'} \right) V_{h+1}^{\pi, \mathcal{P}', R} \right) (a_{1:h}, o_{2:h}) \right]
$$

$$
+ \mathbb{E}_{(a_{1:H'-1}, o_{2:H'-1}) \sim \pi}^{\overline{\mathcal{P}}_{H'-1}} \left[ \left( \left( \mathbb{P}_{H'-1}^{\overline{\mathcal{P}}_{H'-1}} - \mathbb{P}_{H'-1}^{\mathcal{P}'} \right) R_{H'} \right) (a_{1:H'-1}, o_{2:H'-1}) \right],
$$

*where we have written $\overline{\mathcal{P}}_h = \overline{\mathcal{P}}_{\phi,h}(\pi^{1:H})$ in the above expression.*

*Proof.* Fix a deterministic policy $\pi \in \Pi^{\mathrm{det}}$. For $h \in [H]$, write $\overline{\mathcal{P}}_h := \overline{\mathcal{P}}_{\phi,h}(\pi^{1:H})$. For $h \in [H]$ satisfying $h > L$, write $\mathcal{U}_{h-L} := \mathcal{U}_{\phi,h-L}^{\overline{\mathcal{P}}_{h-1}}(\pi^h)$.

The definition of $\overline{\mathcal{P}}_h$, for each $h \in [H]$, ensures that for any $g \geq h$, and any $a_{1:g-1} \in \mathcal{A}^{g-1}, o_{2:g} \in \mathcal{O}^{g-1}, s_{1:g} \in \mathcal{S}^g$ so that for all $h' \leq h - L$, $s_{h'} \notin \mathcal{U}_{h'}$, it holds that $\mathbb{P}_\pi^{\mathcal{P}}(s_{1:g}, a_{1:g-1}, o_{2:g}) = \mathbb{P}_\pi^{\overline{\mathcal{P}}_h}(s_{1:g}, a_{1:g-1}, o_{2:g})$. Therefore, for any $h, g \in [H]$ with $g \geq h$, we have, for all $(a_{1:g-1}, o_{2:g}) \in (\mathcal{A} \times \mathcal{O})^{g-1}$,

$$
\mathbb{P}_\pi^{\mathcal{P}}(a_{1:g-1}, o_{2:g} \text{ and } \forall h' \leq h - L, \ s_{h'} \notin \mathcal{U}_{h'}) = \mathbb{P}_\pi^{\overline{\mathcal{P}}_h}(a_{1:g-1}, o_{2:g}). \tag{21}
$$

Thus, letting $a_g = \pi_g(a_{1:g-1}, o_{2:g})$, we have

$$
\mathbb{P}_g^{\overline{\mathcal{P}}_h}(o_{g+1} | a_{1:g}, o_{2:g}) = \mathbb{P}_\pi^{\mathcal{P}}(o_{g+1} | a_{1:g}, o_{2:g} \text{ and } \forall h' \leq h - L, \ s_{h'} \notin \mathcal{U}_{h'}). \tag{22}
$$

For $h \leq H'$ and $(a_{1:h-1}, o_{2:h}) \in (\mathcal{A} \times \mathcal{O})^{h-1}$, write

$$
U_h^\pi(a_{1:h-1}, o_{2:h}) := \mathbb{E}_\pi^{\mathcal{P}} \left[ R_{H'}(o_{H'}) \cdot \mathbb{1}[\forall h' \leq H' - L - 1, \ s_{h'} \notin \mathcal{U}_{h'}] \right.
$$
$$
\left. | \ a_{1:h-1}, o_{2:h}, \ \forall h' \leq h - L - 1, \ s_{h'} \notin \mathcal{U}_{h'} \right].
$$

From (21) and the fact that $R_{H'}(o^{\mathrm{sink}}) = 0$, we have that $U_1^\pi(\emptyset) = V_1^{\pi, \overline{\mathcal{P}}_{H'-1}, R}(\emptyset)$. Furthermore, for all $a_{1:H'-1}, o_{2:H'}$ so that $\mathbb{P}_\pi^{\mathcal{P}}(a_{1:H'-1}, o_{2:H'}, \forall h' \leq H' - L - 1, \ s_{h'} \notin \mathcal{U}_{h'}) > 0$, we have $U_{H'}^\pi(a_{1:H'-1}, o_{2:H'}) = R_{H'}(o_{H'})$.

Next for any $h \leq H' - 1$ and $(a_{1:h-1}, o_{2:h}) \in (\mathcal{A} \times \mathcal{O})^{h-1}$, letting $a_h = \pi_h(a_{1:h-1}, o_{2:h})$, we have

$$
U_h^\pi(a_{1:h-1}, o_{2:h})
$$
$$
= \sum_{o_{h+1} \in \mathcal{O}} \mathbb{P}_\pi^{\mathcal{P}} \left( \begin{smallmatrix} (a_{1:h}, o_{2:h+1}) \text{ and} \\ s_{h-L} \notin \mathcal{U}_{h-L} \end{smallmatrix} \ \Big| \ \begin{smallmatrix} a_{1:h}, o_{2:h} \text{ and} \\ \forall h' \leq h-L-1, s_{h'} \notin \mathcal{U}_{h'} \end{smallmatrix} \right) \cdot U_{h+1}^\pi(a_{1:h}, o_{2:h+1})
$$
$$
= \mathbb{P}_\pi^{\mathcal{P}} \left( s_{h-L} \notin \mathcal{U}_{h-L} \ \Big| \ \begin{smallmatrix} a_{1:h}, o_{2:h} \text{ and} \\ \forall h' \leq h-L-1, s_{h'} \notin \mathcal{U}_{h'} \end{smallmatrix} \right) \cdot \sum_{o_{h+1} \in \mathcal{O}} \mathbb{P}_\pi^{\mathcal{P}} \left( o_{h+1} \ \Big| \ \begin{smallmatrix} a_{1:h}, o_{2:h} \text{ and} \\ \forall h' \leq h-L, s_{h'} \notin \mathcal{U}_{h'} \end{smallmatrix} \right) \cdot U_{h+1}^\pi(a_{1:h}, o_{2:h+1})
$$
$$
= \mathbb{P}_\pi^{\mathcal{P}} \left( s_{h-L} \notin \mathcal{U}_{h-L} \ \Big| \ \begin{smallmatrix} a_{1:h}, o_{2:h} \text{ and} \\ \forall h' \leq h-L-1, s_{h'} \notin \mathcal{U}_{h'} \end{smallmatrix} \right) \cdot \sum_{o_{h+1} \in \mathcal{O}} \mathbb{P}_\pi^{\overline{\mathcal{P}}_h}(o_{h+1} \ | \ a_{1:h}, o_{2:h}) \cdot U_{h+1}^\pi(a_{1:h}, o_{2:h+1}),
$$

$$\tag{23}$$

where the first equality follows from the definition of $U_h^\pi$, the second equality follows from rearranging, and the third equality follows from (22).

For each $h \le H'$ and $(a_{1:h-1}, o_{2:h}) \in (\mathcal{A} \times \mathcal{O})^{h-1}$, define

$$\widetilde{V}_h^\pi(a_{1:h-1}, o_{2:h}) := \begin{cases} V_h^{\pi,\mathcal{P}',R}(a_{1:h-1}, o_{2:h}) & : h \le H' - 1 \\ R_{H'}(o_{H'}) & : h = H'. \end{cases}$$

Moreover, from the definition of $V_h^{\pi,\mathcal{P}'}(\cdot)$, we have (again for $h \le H' - 1$)

$$\widetilde{V}_h^\pi(a_{1:h-1}, o_{2:h}) = \sum_{o_{h+1} \in \mathcal{O}} \mathbb{P}_h^{\mathcal{P}'}(o_{h+1}|a_{1:h}, o_{2:h}) \cdot \widetilde{V}_{h+1}^\pi(a_{1:h}, o_{2:h+1}). \tag{24}$$

The above is immediate for $h < H' - 1$ by definition of $V_h^{\pi,\mathcal{P}',R}(\cdot)$ and the fact that $R_{h'} \equiv 0$ for $h' \ne H'$, whereas for $h = H' - 1$, the fact that $V_{H'}^{\pi,\mathcal{P}',R} \equiv 0$ means that $V_h^{\pi,\mathcal{P}',R}(a_{1:h-1}, o_{2:h}) = (\mathbb{P}_h^{\mathcal{P}'}(R_{h+1} + V_{h+1}^{\pi,\mathcal{P}',R}))(a_{1:h}, o_{2:h}) = (\mathbb{P}_h^{\mathcal{P}'} R_{h+1})(a_{1:h}, o_{2:h}) = (\mathbb{P}_h^{\mathcal{P}'} \widetilde{V}_{h+1}^\pi)(a_{1:h}, o_{2:h})$, which is exactly the content of (24). Furthermore, in both cases it is permissible to take the sum over $\mathcal{O}$ (as opposed to $\overline{\mathcal{O}}$) since if $o_{h+1} = o^{\mathrm{sink}}$, then $\widetilde{V}_{h+1}^\pi(a_{1:h}, o_{2:h+1}) = 0$ by our assumptions in the lemma statement.

Now, for any $h \in [H' - 1]$, we may bound

$$\sum_{a_{1:h}, o_{2:h}} \mathbb{P}_\pi^{\mathcal{P}} \left( \begin{smallmatrix} (a_{1:h}, o_{2:h}) \text{ and} \\ \forall h' \le h-L-1,\ s_{h'} \notin \mathcal{U}_{h'} \end{smallmatrix} \right) \cdot \left( U_h^\pi(a_{1:h-1}, o_{2:h}) - \widetilde{V}_h^\pi(a_{1:h-1}, o_{2:h}) \right)$$

$$\le \psi + \sum_{a_{1:h}, o_{2:h}} \mathbb{P}_\pi^{\mathcal{P}} \left( \begin{smallmatrix} (a_{1:h}, o_{2:h}) \text{ and} \\ \forall h' \le h-L-1,\ s_{h'} \notin \mathcal{U}_{h'} \end{smallmatrix} \right) \cdot \mathbb{P}_\pi^{\mathcal{P}} \left( s_{h-L} \notin \mathcal{U}_{h-L} \mid \begin{smallmatrix} a_{1:h}, o_{2:h} \text{ and} \\ \forall h' \le h-L-1, s_{h'} \notin \mathcal{U}_{h'} \end{smallmatrix} \right)$$

$$\cdot \left( \sum_{o_{h+1} \in \mathcal{O}} \mathbb{P}_h^{\overline{\mathcal{P}}_h}(o_{h+1}|a_{1:h}, o_{2:h}) \cdot U_{h+1}^\pi(a_{1:h}, o_{2:h+1}) - \sum_{o_{h+1} \in \mathcal{O}} \mathbb{P}_h^{\mathcal{P}'}(o_{h+1}|a_{1:h}, o_{2:h}) \cdot \widetilde{V}_{h+1}^\pi(a_{1:h}, o_{2:h+1}) \right) \tag{25}$$

$$= \psi + \sum_{a_{1:h}, o_{2:h}} \mathbb{P}_\pi^{\mathcal{P}} \left( \begin{smallmatrix} (a_{1:h}, o_{2:h}) \text{ and} \\ \forall h' \le h-L,\ s_{h'} \notin \mathcal{U}_{h'} \end{smallmatrix} \right) \cdot \left( \sum_{o_{h+1} \in \mathcal{O}} \left( \mathbb{P}_h^{\overline{\mathcal{P}}_h}(o_{h+1}|a_{1:h}, o_{2:h}) - \mathbb{P}_h^{\mathcal{P}'}(o_{h+1}|a_{1:h}, o_{2:h}) \right) \cdot \widetilde{V}_{h+1}^\pi(a_{1:h}, o_{2:h+1}) \right)$$

$$+ \sum_{a_{1:h}, o_{2:h}} \mathbb{P}_\pi^{\mathcal{P}} \left( \begin{smallmatrix} (a_{1:h}, o_{2:h}) \text{ and} \\ \forall h' \le h-L,\ s_{h'} \notin \mathcal{U}_{h'} \end{smallmatrix} \right) \cdot \sum_{o_{h+1} \in \mathcal{O}} \mathbb{P}_h^{\overline{\mathcal{P}}_h}(o_{h+1}|a_{1:h}, o_{2:h}) \cdot \left( U_{h+1}^\pi(a_{1:h}, o_{2:h+1}) - \widetilde{V}_{h+1}^\pi(a_{1:h}, o_{2:h+1}) \right)$$

$$= \psi + \sum_{a_{1:h}, o_{2:h}} \mathbb{P}_\pi^{\mathcal{P}} \left( \begin{smallmatrix} (a_{1:h}, o_{2:h}) \text{ and} \\ \forall h' \le h-L,\ s_{h'} \notin \mathcal{U}_{h'} \end{smallmatrix} \right) \cdot \left( \sum_{o_{h+1} \in \mathcal{O}} \left( \mathbb{P}_h^{\overline{\mathcal{P}}_h}(o_{h+1}|a_{1:h}, o_{2:h}) - \mathbb{P}_h^{\mathcal{P}'}(o_{h+1}|a_{1:h}, o_{2:h}) \right) \cdot \widetilde{V}_{h+1}^\pi(a_{1:h}, o_{2:h+1}) \right)$$

$$+ \sum_{a_{1:h+1}, o_{2:h+1}} \mathbb{P}_\pi^{\mathcal{P}} \left( \begin{smallmatrix} (a_{1:h+1}, o_{2:h+1}) \text{ and} \\ \forall h' \le h-L,\ s_{h'} \notin \mathcal{U}_{h'} \end{smallmatrix} \right) \cdot \left( U_{h+1}^\pi(a_{1:h}, o_{2:h+1}) - \widetilde{V}_{h+1}^\pi(a_{1:h}, o_{2:h+1}) \right), \tag{26}$$

where (25) uses (23), (24), and the fact that $\widetilde{V}_h^\pi(a_{1:h-1}, o_{2:h}) \ge -\psi$ for all $h \le H' - 1$ and all $(a_{1:h-1}, o_{2:h}) \in \mathscr{H}_h$. Furthermore, (26) uses (22) and the fact that $\mathbb{P}_\pi^{\mathcal{P}} \left( \begin{smallmatrix} (a_{1:h}, o_{2:h+1}) \text{ and} \\ \forall h' \le h-L,\ s_{h'} \notin \mathcal{U}_{h'} \end{smallmatrix} \right) = \mathbb{P}_\pi^{\mathcal{P}} \left( \begin{smallmatrix} (a_{1:h+1}, o_{2:h+1}) \text{ and} \\ \forall h' \le h-L,\ s_{h'} \notin \mathcal{U}_{h'} \end{smallmatrix} \right)$ for $a_{h+1} = \pi_{h+1}(a_{1:h}, o_{2:h+1})$ and $\mathbb{P}_\pi^{\mathcal{P}} \left( \begin{smallmatrix} (a_{1:h+1}, o_{2:h+1}) \text{ and} \\ \forall h' \le h-L,\ s_{h'} \notin \mathcal{U}_{h'} \end{smallmatrix} \right) = 0$ whenever $a_{h+1} \ne \pi_{h+1}(a_{1:h}, o_{2:h+1})$.

Thus, by adding the display (26) for $1 \le h \le H' - 1$, we see that

$$U_1^\pi(\emptyset) - V_1^{\pi, \mathcal{P}', R}(\emptyset)$$

$$= U_1^\pi(\emptyset) - \widetilde{V}_1^\pi(\emptyset)$$

$$\le \sum_{h=1}^{H'-1} \sum_{a_{1:h}, o_{2:h}} \mathbb{P}_\pi^\mathcal{P} \left( \begin{smallmatrix} (a_{1:h}, o_{2:h}) \text{ and} \\ \forall h' \le h-L, \ s_{h'} \notin \mathcal{U}_{h'} \end{smallmatrix} \right) \cdot \left( \sum_{o_{h+1} \in \mathcal{O}} \left( \mathbb{P}_h^{\overline{\mathcal{P}}_h}(o_{h+1}|a_{1:h}, o_{2:h}) - \mathbb{P}_h^{\mathcal{P}'}(o_{h+1}|a_{1:h}, o_{2:h}) \right) \cdot \widetilde{V}_{h+1}^\pi(a_{1:h}, o_{2:h+1}) \right)$$

$$+ \sum_{a_{1:H'}, o_{2:H'}} \mathbb{P}_\pi^\mathcal{P} \left( \begin{smallmatrix} (a_{1:H'}, o_{2:H'}) \text{ and} \\ \forall h' \le H'-L-1, \ s_{h'} \notin \mathcal{U}_{h'} \end{smallmatrix} \right) \cdot \left( U_{H'}^\pi(a_{1:H'-1}, o_{2:H'}) - \widetilde{V}_{H'}^\pi(a_{1:H'-1}, o_{2:H'}) \right) + \psi H$$

$$= \sum_{h=1}^{H'-2} \sum_{a_{1:h}, o_{2:h}} \mathbb{P}_\pi^{\overline{\mathcal{P}}_h}(a_{1:h}, o_{2:h}) \cdot \left( \sum_{o_{h+1} \in \mathcal{O}} \left( \mathbb{P}_h^{\overline{\mathcal{P}}_h}(o_{h+1}|a_{1:h}, o_{2:h}) - \mathbb{P}_h^{\mathcal{P}'}(o_{h+1}|a_{1:h}, o_{2:h}) \right) \cdot \widetilde{V}_{h+1}^\pi(a_{1:h}, o_{2:h+1}) \right)$$

$$+ \sum_{a_{1:H'}, o_{2:H'}} \mathbb{P}_\pi^\mathcal{P} \left( \begin{smallmatrix} (a_{1:H'}, o_{2:H'}) \text{ and} \\ \forall h' \le H'-L-1, \ s_{h'} \notin \mathcal{U}_{h'} \end{smallmatrix} \right) \cdot \left( R_{H'}(o_{H'}) - R_{H'}(o_{H'}) \right) + \psi H$$

$$= \psi H + \sum_{h=1}^{H'-2} \mathbb{E}_{(a_{1:h}, o_{2:h}) \sim \pi}^{\overline{\mathcal{P}}_h} \left[ \left( \left( \mathbb{P}_h^{\overline{\mathcal{P}}_h} - \mathbb{P}_h^{\mathcal{P}'} \right) V_{h+1}^{\pi, \mathcal{P}', R} \right)(a_{1:h}, o_{2:h}) \right]$$

$$+ \mathbb{E}_{(a_{1:H'-1}, o_{2:H'-1}) \sim \pi}^{\overline{\mathcal{P}}_{H'-1}} \left[ \left( \left( \mathbb{P}_{H'-1}^{\overline{\mathcal{P}}_{H'-1}} - \mathbb{P}_{H'-1}^{\mathcal{P}'} \right) R_{H'} \right)(a_{1:H'-1}, o_{2:H'-1}) \right].$$

where the final equality follows from the fact that for $(a_{1:h-1}, o_{2:h}) \in \mathscr{H}_h \backslash (\mathcal{A} \times \mathcal{O})^{h-1}$, we have $V_{h+1}^{\pi, \mathcal{P}', R}(a_{1:h}, o_{2:h+1}) = 0$ for all $o_{h+1} \in \overline{\mathcal{O}}$ by our assumptions in the lemma statement. $\qquad \square$

Lemma G.2 shows that that the transition function of the MDP $\widehat{\mathcal{M}}(\pi^{1:H})$ approximates that of the truncated POMDP $\overline{\mathcal{P}}_{\phi,h}(\pi^{1:H})$ at step $h$. In light of the fact that $\overline{\mathcal{P}}_{\phi,h}(\pi^{1:H})$ diverts a potentially large amount of mass to the sink state $s^{\text{sink}}$, it may be somewhat surprising that the result below is two-sided in nature, i.e., we get an upper bound on the total variation distance between $\mathbb{P}_h^{\overline{\mathcal{P}}_{\phi,h}(\pi^{1:H})}(\cdot|z_{1:h}, a_h)$ and $\mathbb{P}_h^{\widehat{\mathcal{M}}(\pi^{1:H})}(\cdot|z_h, a_h)$. This may be explained by the fact that the roll-in trajectory $(z_{1:h}, a_h)$ is drawn from the truncated POMDP $\overline{\mathcal{P}}_{\phi,h}(\pi^{1:H})$, and the (latent state) transition functions of $\overline{\mathcal{P}}_{\phi,h}(\pi^{1:H})$ and $\mathcal{P}$ are identical at step $h - L$ and thereafter. Thus, trajectories $(z_{1:h}, a_h)$ which $\overline{\mathcal{P}}_{\phi,h}(\pi^{1:H})$ does not reach do not contribute to the left-hand side of (27), and trajectories which are reached are not truncated.

**Lemma G.2.** *Consider any sequence of general policies $\pi^1, \ldots, \pi^H \in \Pi^{\text{gen}}$ and $\phi > 0$. Suppose that, in the construction of $\widehat{\mathcal{M}}(\pi^{1:H})$, the event $\mathcal{E}^{\text{low}}$ of Lemma E.1 holds. Then for general policies $\pi$ and all $h \in [H]$,*

$$\mathbb{E}_{(z_{1:h}, a_h) \sim \pi}^{\overline{\mathcal{P}}_{\phi,h}(\pi^{1:H})} \sum_{z_{h+1} \in \overline{\mathcal{Z}}} \left| \mathbb{P}_h^{\overline{\mathcal{P}}_{\phi,h}(\pi^{1:H})}(z_{h+1}|z_{1:h}, a_h) - \mathbb{P}_h^{\widehat{\mathcal{M}}(\pi^{1:H})}(z_{h+1}|z_h, a_h) \right| \le \epsilon + \theta + \frac{A^{2L} O^L \zeta}{\phi}.$$

(27)

*Proof.* We write $\overline{\mathcal{P}} := \overline{\mathcal{P}}_{\phi,h}(\pi^{1:H})$, $\widetilde{\mathcal{M}} := \widetilde{\mathcal{M}}(\pi^{1:H})$, $\widehat{\mathcal{M}} := \widehat{\mathcal{M}}(\pi^{1:H})$. Throughout the proof of the lemma we write $z_{h'} = (a_{h'-L:h'-1}, o_{h'-L+1:h'})$ for $h' \in [h]$.

First note that

$$\mathbb{E}_{(z_{1:h}, a_h) \sim \pi}^{\overline{\mathcal{P}}} \sum_{z_{h+1} \in \overline{\mathcal{Z}}} \left| \mathbb{P}_h^{\overline{\mathcal{P}}}(z_{h+1}|z_{1:h}, a_h) - \mathbb{P}_h^{\widehat{\mathcal{M}}}(z_{h+1}|z_h, a_h) \right|$$

$$\le \mathbb{E}_{(z_{1:h}, a_h) \sim \pi}^{\overline{\mathcal{P}}} \sum_{z_{h+1} \in \overline{\mathcal{Z}}} \left| \mathbb{P}_h^{\overline{\mathcal{P}}}(z_{h+1}|z_{1:h}, a_h) - \mathbb{P}_h^{\widetilde{\mathcal{M}}}(z_{h+1}|z_h, a_h) \right|$$

$$+ \mathbb{E}_{(z_{1:h}, a_h) \sim \pi}^{\overline{\mathcal{P}}} \sum_{z_{h+1} \in \overline{\mathcal{Z}}} \left| \mathbb{P}_h^{\widetilde{\mathcal{M}}}(z_{h+1}|z_{1:h}, a_h) - \mathbb{P}_h^{\widehat{\mathcal{M}}}(z_{h+1}|z_h, a_h) \right|.$$

Therefore, to prove the statement of the lemma, it suffices to show that

$$\mathbb{E}^{\overline{\mathcal{P}}}_{(z_{1:h},a_h)\sim\pi}\sum_{z_{h+1}\in\overline{\mathcal{Z}}}\left|\mathbb{P}^{\overline{\mathcal{P}}}_h(z_{h+1}|z_{1:h},a_h)-\mathbb{P}^{\widetilde{\mathcal{M}}}_h(z_{h+1}|z_h,a_h)\right|\leq\epsilon \tag{28}$$

$$\mathbb{E}^{\overline{\mathcal{P}}}_{(z_{1:h},a_h)\sim\pi}\sum_{z_{h+1}\in\overline{\mathcal{Z}}}\left|\mathbb{P}^{\widetilde{\mathcal{M}}}_h(z_{h+1}|z_h,a_h)-\mathbb{P}^{\widehat{\mathcal{M}}}_h(z_{h+1}|z_h,a_h)\right|\leq\theta+\frac{A^{2L}O^L\zeta}{\phi}. \tag{29}$$

Note that for any $s\in\mathcal{S}$, if $h>L$ and there is some policy $\pi\in\Pi^{\mathrm{gen}}$ so that $d^{\overline{\mathcal{P}},\pi}_{\mathsf{S},h-L}(s)>0$, then the fact that $\overline{\mathcal{P}}=\overline{\mathcal{P}}_{\phi,h}(\pi^{1:H})$ and Lemma E.2 ensures that $s\notin\mathcal{U}^{\overline{\mathcal{P}}}_{\phi,h-L}(\pi^h)$. Furthermore, we have $\mathbb{T}^{\overline{\mathcal{P}}}_{h'}=\mathbb{T}^{\mathcal{P}}_{h'}$ for all $h'\geq h-L$ and $\mathbb{O}^{\overline{\mathcal{P}}}_{h'}=\mathbb{O}^{\mathcal{P}}_{h'}$ for all $h'\in[H]$. Thus we may apply Lemma F.3 to the POMDP $\overline{\mathcal{P}}$ with $\pi'=\pi^h$, which gives that for general policies $\pi$,

$$\mathbb{E}^{\overline{\mathcal{P}}}_{a_{1:h-1},o_{2:h}\sim\pi}\left[\sum_{s\in\mathcal{S}}\left|\mathbf{b}^{\overline{\mathcal{P}}}_h(a_{1:h-1},o_{2:h})(s)-\widetilde{\mathbf{b}}^{\pi^h}_h(a_{h-L:h-1},o_{h-L+1:h})(s)\right|\right]\leq\epsilon. \tag{30}$$

Now, for any choice of $(z_{1:h},a_h)\in\mathcal{Z}^h\times\mathcal{A}$ (which is equivalent to the data of $((a_{1:h-1},o_{2:h}),a_h)$) we have that, for $o_{h+1}\in\mathcal{O}$, letting $z_{h+1}=(a_{h-L+1:h},o_{h-L+2:h+1})$,

$$\mathbb{P}^{\overline{\mathcal{P}}}_h(z_{h+1}|z_{1:h},a_h)=e^{\top}_{o_{h+1}}\cdot\mathbb{O}^{\mathcal{P}}_{h+1}\cdot\mathbb{T}^{\mathcal{P}}_h(a_h)\cdot\mathbf{b}^{\overline{\mathcal{P}}}_h(a_{1:h-1},o_{2:h}).$$

On the other hand, (10) gives

$$\mathbb{P}^{\widetilde{\mathcal{M}}}_h(z_{h+1}|z_h,a_h):=e^{\top}_{o_{h+1}}\cdot\mathbb{O}^{\mathcal{P}}_{h+1}\cdot\mathbb{T}^{\mathcal{P}}_h(a_h)\cdot\widetilde{\mathbf{b}}^{\pi^h}_h(a_{h-L:h-1},o_{h-L+1:h}).$$

Moreover, for $z_{h+1}\in\overline{\mathcal{Z}}\backslash\mathcal{Z}$, we have $\mathbb{P}^{\overline{\mathcal{P}}}_h(z_{h+1}|z_{1:h},a_h)=\mathbb{P}^{\widetilde{\mathcal{M}}}_h(z_{h+1}|z_h,a_h)=0$. Thus, we get that

$$\sum_{z_{h+1}\in\overline{\mathcal{Z}}}\left|\mathbb{P}^{\overline{\mathcal{P}}}_h(z_{h+1}|z_{1:h},a_{1:h})-\mathbb{P}^{\widetilde{\mathcal{M}}}_h(z_{h+1}|z_h,a_h)\right|$$

$$\leq\sum_{o_{h+1}\in\mathcal{O}}\left|e^{\top}_{o_{h+1}}\cdot\mathbb{O}^{\mathcal{P}}_{h+1}\cdot\mathbb{T}^{\mathcal{P}}_h(a_h)\cdot\left(\mathbf{b}^{\overline{\mathcal{P}}}_h(a_{1:h-1},o_{2:h})-\widetilde{\mathbf{b}}^{\pi^h}_h(a_{h-L:h-1},o_{h-L+1:h})\right)\right| \tag{31}$$

$$\leq\sum_{s\in\mathcal{S}}\left|\mathbf{b}^{\overline{\mathcal{P}}}_h(a_{1:h-1},o_{2:h})(s)-\widetilde{\mathbf{b}}^{\pi^h}_h(a_{h-L:h-1},o_{h-L+1:h})(s)\right|, \tag{32}$$

where (32) uses the data processing inequality for total variation distance (together with the fact that, for $(z_{1:h},a_h)\in\mathcal{Z}^h\times\mathcal{A}$, we have $\mathbf{b}^{\overline{\mathcal{P}}}_h(a_{1:h-1},o_{2:h})(s^{\mathrm{sink}})=\widetilde{\mathbf{b}}^{\pi^h}_h(a_{h-L:h-1},o_{h-L+1:h})(s^{\mathrm{sink}})=0$). The above display, together with (30) and the fact that for any $z_h\in\overline{\mathcal{Z}}\backslash\mathcal{Z}$, $\mathbb{P}^{\overline{\mathcal{P}}}_h(o^{\mathrm{sink}}|z_{1:h},a_h)=\mathbb{P}^{\widetilde{\mathcal{M}}}_h(o^{\mathrm{sink}}|z_h,a_h)=1$, gives (28).

Next, we remark that for all $z_h\in\mathcal{Z},a_h\in\mathcal{A}$, it holds that

$$\sum_{z_{h+1}\in\overline{\mathcal{Z}}}\left|\mathbb{P}^{\widetilde{\mathcal{M}}}_h(z_{h+1}|z_h,a_h)-\mathbb{P}^{\widehat{\mathcal{M}}}_h(z_{h+1}|z_h,a_h)\right|=\frac{1}{2}\left\|\mathbb{P}^{\widetilde{\mathcal{M}}}_h(\cdot|z_h,a_h)-\mathbb{P}^{\widehat{\mathcal{M}}}_h(\cdot|z_h,a_h)\right\|_1\leq\theta+\mathbb{1}[z_h\in\mathcal{Z}^{\mathrm{low}}_{h,\zeta}(\widehat{\pi}^h)],$$

where the inequality follows from Lemma E.1 (and assumption that $\mathcal{E}^{\mathrm{low}}$ holds). Taking expectation over $(z_{1:h},a_{1:h})\sim(\overline{\mathcal{P}},\pi)$, we obtain that

$$\mathbb{E}^{\overline{\mathcal{P}}}_{(z_{1:h},a_{1:h})\sim\pi}\sum_{z_{h+1}\in\overline{\mathcal{Z}}}\left|\mathbb{P}^{\widetilde{\mathcal{M}}}_h(z_{h+1}|z_h,a_h)-\mathbb{P}^{\widehat{\mathcal{M}}}_h(z_{h+1}|z_h,a_h)\right|$$

$$\leq\theta+\mathbb{P}^{\overline{\mathcal{P}}}_{(z_{1:h},a_{1:h})\sim\pi}\left(z_h\in\mathcal{Z}^{\mathrm{low}}_{h,\zeta}(\widehat{\pi}^h)\right)$$

$$=\theta+d^{\overline{\mathcal{P}},\pi}_{\mathsf{Z},h}(\mathcal{Z}^{\mathrm{low}}_{h,\zeta}(\widehat{\pi}^h))$$

$$\leq\theta+\frac{A^{2L}O^L\zeta}{\phi},$$

where the final inequality follows by Lemma F.4 with $\pi'=\pi^h$ and the fact that if $h>L$, for all $s$ with $d^{\overline{\mathcal{P}},\pi}_{\mathsf{S},h-L}(s)>0$, it holds that $s\notin\mathcal{U}^{\overline{\mathcal{P}}}_{\phi,h-L}(\pi^h)$ (Lemma E.2), meaning that $d^{\overline{\mathcal{P}},\pi}_{\mathsf{S},h-L}(\mathcal{U}^{\overline{\mathcal{P}}}_{\phi,h-L}(\pi^h))=0$. $\qquad\square$

Lemma G.3 shows that for any reward function of a particular form, the associated value function of $\widehat{\mathcal{M}}(\pi^{1:H})$, under any general policy, is nonnegative. Roughly speaking, this holds because the reward functions considered in Lemma G.3 belong to the convex hull of indicators of subsets of (latent) states. Since $\widehat{\mathcal{M}}(\pi^{1:H})$ is an approximation of $\mathcal{P}$ and the value function for such rewards would be non-negative in $\mathcal{P}$, it is too in $\widehat{\mathcal{M}}(\pi^{1:H})$.

**Lemma G.3.** *Consider any sequence of general policies $\pi^1, \ldots, \pi^H \in \Pi^{\mathrm{gen}}$ and $\phi > 0$. Fix any $H' \in [H]$. Consider any collection of reward functions $R = (R_1, \ldots, R_H)$, $R_h : \overline{\mathcal{O}} \to \mathbb{R}$ so that $R_{h'} \equiv 0$ for all $h' > H'$, and so that for each $h' \leq H'$, there are weights $\alpha_{h',s} \in [0,1]$, $s \in \mathcal{S}$ so that $R_{h'}(o) = \sum_{s \in \mathcal{S}} \alpha_{h',s} \cdot (\mathbb{O}_{h'}^{\mathcal{P}})_{s,o}^{\dagger}$ for all $o \in \mathcal{O}$. Suppose that in the construction of $\widehat{\mathcal{M}}(\pi^{1:H})$ the event $\mathcal{E}^{\mathrm{low}}$ of Lemma E.1 holds. Then for all $\pi \in \Pi^{\mathrm{gen}}$, $h \in [H]$, and $z_h \in \mathcal{Z}$,*

$$V_h^{\pi, \widehat{\mathcal{M}}(\pi^{1:H}), R}(z_h) \geq -\frac{H(H-h)\sqrt{S}}{\gamma} \cdot \theta.$$

*Proof.* Write $\widehat{\mathcal{M}} = \widehat{\mathcal{M}}(\pi^{1:H})$. Note that by our assumption on $R$ and Lemma B.3, we have that $|R_h(o)| \leq \frac{\sqrt{S}}{\gamma}$ for all $h, o$, meaning that $|V_h^{\pi, \widehat{\mathcal{M}}, R}(z_h)| \leq \frac{H\sqrt{S}}{\gamma}$ for all $h, z_h$.

Note that for all $h \in [H]$ and $z_h \in \overline{\mathcal{Z}} \backslash \mathcal{Z}$, we have $V_h^{\pi, \widehat{\mathcal{M}}, R}(z_h) = 0$, so it suffices to consider the case that $z_h \in \mathcal{Z}$; we will do so via reverse induction on $h$. For the base case, we have that $V_{H'}^{\pi, \widehat{\mathcal{M}}, R} \equiv 0$ (which follows from $R_{h'} \equiv 0$ for all $h' > H'$). Now fix some $h < H'$, assume that $V_{h+1}^{\pi, \widehat{\mathcal{M}}, R}(z_{h+1}) \geq -\frac{H(H-h-1)\sqrt{S}}{\gamma} \cdot \theta$ for all $z_{h+1} \in \overline{\mathcal{Z}}$, and consider any $z_h = (a_{h-L:h-1}, o_{h-L+1:h}) \in \overline{\mathcal{Z}}$. Then we have, for any $z_h \in \mathcal{Z}$:

$$V_h^{\pi, \widehat{\mathcal{M}}, R}(z_h) = \mathbb{E}_{z_{h+1} \sim \mathbb{P}_h^{\widehat{\mathcal{M}}}(\cdot | z_h, a_h)} \left[ R_{h+1}(\mathsf{o}(z_{h+1})) + V_{h+1}^{\pi, \widehat{\mathcal{M}}, R}(z_{h+1}) \right]$$

$$\geq \mathbb{E}_{z_{h+1} \sim \mathbb{P}_h^{\widehat{\mathcal{M}}}(\cdot | z_h, a_h)} \left[ R_{h+1}(\mathsf{o}(z_{h+1})) + V_{h+1}^{\pi, \widehat{\mathcal{M}}, R}(z_{h+1}) \right] - \frac{H\sqrt{S}}{\gamma} \cdot \theta \tag{33}$$

$$\geq \mathbb{E}_{z_{h+1} \sim \mathbb{P}_h^{\widehat{\mathcal{M}}}(\cdot | z_h, a_h)} \left[ R_{h+1}(\mathsf{o}(z_{h+1})) \right] - \frac{H(H-h)\sqrt{S}}{\gamma} \cdot \theta \tag{34}$$

$$= \sum_{o_{h+1} \in \mathcal{O}} \mathbb{P}_h^{\widehat{\mathcal{M}}} \left( (a_{h-L+1:h}, o_{h-L+2:h+1}) | z_h, a_h \right) \cdot R_{h+1}(o_{h+1}) - \frac{H(H-h)\sqrt{S}}{\gamma} \cdot \theta$$

$$= \sum_{s \in \mathcal{S}} \alpha_{h+1,s} \cdot \sum_{o_{h+1} \in \mathcal{O}} (\mathbb{O}_{h+1}^{\mathcal{P}})_{s,o_{h+1}}^{\dagger} \cdot e_{o_{h+1}}^{\top} \cdot \mathbb{O}_{h+1}^{\mathcal{P}} \cdot \mathbb{T}_h^{\mathcal{P}}(a_h) \cdot \widetilde{\mathbf{b}}_h^{\pi^h}(a_{h-L:h-1}, o_{h-L+1:h})$$

$$\qquad - \frac{H(H-h)\sqrt{S}}{\gamma} \cdot \theta \tag{35}$$

$$= \sum_{s \in \mathcal{S}} \alpha_{h+1,s} \cdot e_s^{\top} \cdot \mathbb{T}_h^{\mathcal{P}}(a_h) \cdot \widetilde{\mathbf{b}}_h^{\pi^h}(a_{h-L:h-1}, o_{h-L+1:h})$$

$$\qquad - \frac{H(H-h)\sqrt{S}}{\gamma} \cdot \theta \tag{36}$$

$$\geq -\frac{H(H-h)\sqrt{S}}{\gamma} \cdot \theta, \tag{37}$$

where:

- (33) follows because if $z_h, a_h$ are so that $\left\| \mathbb{P}_h^{\widehat{\mathcal{M}}}(\cdot | z_h, a_h) - \mathbb{P}_h^{\widetilde{\mathcal{M}}}(\cdot | z_h, a_h) \right\|_1 > \theta$, then by Lemma E.1 (and assumption that $\mathcal{E}^{\mathrm{low}}$ holds) we have that with probability 1 over $z_{h+1} \sim \mathbb{P}_h^{\widehat{\mathcal{M}}}(\cdot | z_h, a_h)$, it holds that $z_{h+1} \notin \mathcal{Z}$, meaning that $R_{h+1}(z_{h+1}) + V_{h+1}^{\pi, \widehat{\mathcal{M}}, R}(z_{h+1}) = 0$ with probability 1.

On the other hand, if $\left\|\mathbb{P}_h^{\widehat{\mathcal{M}}}(\cdot|z_h, a_h) - \mathbb{P}_h^{\widetilde{\mathcal{M}}}(\cdot|z_h, a_h)\right\|_1 \leq \theta$, then the fact that $\left|R_{h+1}(\mathsf{o}(z_{h+1})) + V_{h+1}^{\pi,\widehat{\mathcal{M}},R}(z_{h+1})\right| \leq \frac{H\sqrt{S}}{\gamma}$ for all $z_{h+1} \in \mathcal{Z}$ establishes (33).

- (34) follows from the inductive hypothesis;

- (35) follows from (10);

- the final inequality in (37) follows since all $\alpha_{h+1,s} \geq 0$ and the entries of $\mathbb{T}_h^{\mathcal{P}}(a_h)$, $\widetilde{\mathbf{b}}_h^{\pi^h}(a_{h-L:h-1}, o_{h-L+1:h})$ are all non-negative.

(37) completes the inductive step, and thus the proof of the lemma. $\qquad \square$

Using Lemmas G.2 and G.3, we next establish Lemma G.4, the main result of the section, which shows that for any reward function which is a special case of the type used in Lemma G.3, the value function of $\overline{\mathcal{P}}_{\phi,H''}(\pi^{1:H})$ is approximately upper bounded by that of $\widehat{\mathcal{M}}(\pi^{1:H})$ (for appropriate choices of $H''$). Intuitively, the lemma holds since, as established in Lemma G.2, the transitions of $\overline{\mathcal{P}}_{\phi,h}(\pi^{1:H})$ and $\widehat{\mathcal{M}}(\pi^{1:H})$ are close at step $h$. Furthermore, Lemma G.1 ensures that the difference in value functions of $\overline{\mathcal{P}}_{\phi,H''}(\pi^{1:H})$ and $\widehat{\mathcal{M}}(\pi^{1:H})$ may be bounded above by that difference in transitions at step $h$.

**Lemma G.4.** *Consider any sequence of general policies $\pi^1, \ldots, \pi^H \in \Pi^{\mathrm{gen}}$ and $\phi > 0$. Fix any $H' \in [H]$. Consider any collection of reward functions $R = (R_1, \ldots, R_H)$, $R_h : \overline{\mathcal{O}} \to \mathbb{R}$ so that $R_h \equiv 0$ for all $h \neq H'$, and so that there are weights $\alpha_s \in [0,1]$, $s \in \mathcal{S}$ so that $R_{H'}(o) = \sum_{s \in \mathcal{S}} \alpha_s \cdot (\mathbb{O}_{H'}^{\mathcal{P}})_{s,o}^{\dagger}$ for all $o \in \mathcal{O}$. Suppose that, in the construction of $\widehat{\mathcal{M}}(\pi^{1:H})$, the event $\mathcal{E}^{\mathrm{low}}$ of Lemma E.1 holds. Then for all general policies $\pi$ and all $H'' \geq H' - 1$,*

$$V_1^{\pi,\overline{\mathcal{P}}_{\phi,H''}(\pi^{1:H}),R}(\emptyset) - V_1^{\pi,\widehat{\mathcal{M}}(\pi^{1:H}),R}(\emptyset) \leq \frac{H^2\sqrt{S}\epsilon}{\gamma} + \frac{2H^3\theta\sqrt{S}}{\gamma} + \frac{H^2\sqrt{S}A^{2L}O^L\zeta}{\phi\gamma} \tag{38}$$

In the above lemma statement we consider $R$ to be a reward function for the MDP $\widehat{\mathcal{M}}(\pi^{1:H})$ in the natural way, i.e., via abuse of notation, we use the induced mapping $R_h : \overline{\mathcal{Z}} \to \mathbb{R}$ defined by $R_h(z) = R_h(\mathsf{o}(z))$. Note also that for all $h' \in [H]$ we have $R_{h'}(o^{\mathrm{sink}}) = 0$, since $(\mathbb{O}_{h'}^{\mathcal{P}})_{s,o^{\mathrm{sink}}}^{\dagger} = 0$ for all $s \in \mathcal{S}$.

*Proof.* For $h \in [H]$, we write $\overline{\mathcal{P}}_h := \overline{\mathcal{P}}_{\phi,h}(\pi^{1:H})$, and furthermore write $\widetilde{\mathcal{M}} := \widetilde{\mathcal{M}}(\pi^{1:H})$, and $\widehat{\mathcal{M}} := \widehat{\mathcal{M}}(\pi^{1:H})$. By Jensen's inequality, it suffices to consider the case that $\pi$ is a deterministic policy.

Since $\overline{\mathcal{P}}_{H''}$ is a truncation of $\overline{\mathcal{P}}_{H'-1}$ (Lemma F.1), we have that, for any $s \in \mathcal{S}$, $\mathbb{P}_\pi^{\overline{\mathcal{P}}_{H''}}[s_h = s] \leq \mathbb{P}_\pi^{\overline{\mathcal{P}}_{H'-1}}[s_h = s]$. In light of the fact that $V_1^{\pi,\overline{\mathcal{P}}_h,R}(\emptyset) = \mathbb{E}_\pi^{\overline{\mathcal{P}}_h}\left[\sum_{s \in \mathcal{S}} \alpha_s \cdot \mathbb{1}[s_h = s]\right]$ for each $h \in [H]$, it follows that $V_1^{\pi,\overline{\mathcal{P}}_{H''},R}(\emptyset) \leq V_1^{\pi,\overline{\mathcal{P}}_{H'-1},R}(\emptyset)$. Thus it suffices to upper bound $V_1^{\pi,\overline{\mathcal{P}}_{H'-1},R}(\emptyset) - V_1^{\pi,\widehat{\mathcal{M}},R}(\emptyset)$.

Write $\psi := \frac{H^2\theta\sqrt{S}}{\gamma}$. We will now apply Lemma G.1 to the POMDP $\mathcal{P}$ with $\mathcal{P}' = \widehat{\mathcal{M}}$. Its preconditions are verified as follows: certainly $R_{h'} \equiv 0$ for $h' \neq H'$, and moreover $R_{H'}(o^{\mathrm{sink}}) = 0$ since $(\mathbb{O}_{H'}^{\mathcal{P}})_{s,o^{\mathrm{sink}}}^{\dagger} = 0$ for all $s \in \mathcal{S}$. By Lemma G.3, $V_{h+1}^{\pi,\widehat{\mathcal{M}},R}(a_{1:h-1}, o_{2:h}) \geq -\psi$ for all $h$ and $a_{1:h-1}, o_{2:h} \in (\mathcal{A} \times \mathcal{O})^{h-1}$. Furthermore, for all $(a_{1:H-1}, o_{2:H}) \in \mathscr{H}_H$ occuring with positive probability undner $(\sigma, \widehat{\mathcal{M}})$ for any policy $\sigma$, the definition of $\widehat{\mathcal{M}}(\pi^{1:H})$ ensures that if $o_h = o^{\mathrm{sink}}$ for

some $h \in [H]$, then $o_{h'} = o^{\text{sink}}$ for all $h' > h$. Thus, we may apply Lemma G.1, giving that

$$
\begin{aligned}
&V_1^{\pi, \overline{\mathcal{P}}_{H'-1}, R}(\emptyset) - V_1^{\pi, \widehat{\mathcal{M}}, R}(\emptyset) \\
&\leq \sum_{h=1}^{H'-2} \mathbb{E}_{(a_{1:h}, o_{2:h}) \sim \pi}^{\overline{\mathcal{P}}_h} \left[ ((\mathbb{P}_h^{\overline{\mathcal{P}}_h} - \mathbb{P}_h^{\widehat{\mathcal{M}}})(V_{h+1}^{\pi, \widehat{\mathcal{M}}, R}))(a_{1:h}, o_{2:h}) \right] \\
&\quad + \mathbb{E}_{(a_{1:H'-1}, o_{2:H'-1}) \sim \pi}^{\overline{\mathcal{P}}_{H'-1}} \left[ \left( (\mathbb{P}_{H'-1}^{\overline{\mathcal{P}}_{H'-1}} - \mathbb{P}_{H'-1}^{\widehat{\mathcal{M}}}) R_{H'} \right) (a_{1:H'-1}, o_{2:H'-1}) \right] + \psi H \\
&\leq \psi H + \frac{H\sqrt{S}}{\gamma} \cdot \sum_{h=1}^{H'-1} \mathbb{E}_{(a_{1:h}, o_{2:h}) \sim \pi}^{\overline{\mathcal{P}}_h} \left[ \left\| \mathbb{P}_h^{\overline{\mathcal{P}}_h}(\cdot | a_{1:h}, o_{2:h}) - \mathbb{P}_h^{\widehat{\mathcal{M}}}(\cdot | a_{1:h}, o_{2:h}) \right\|_1 \right] \\
&\leq \frac{H^3 \theta \sqrt{S}}{\gamma} + \frac{H^2 \sqrt{S}}{\gamma} \cdot \left( \epsilon + \theta + \frac{A^{2L} O^L \zeta}{\phi} \right) \\
&\leq \frac{H^2 \sqrt{S} \epsilon}{\gamma} + \frac{2H^3 \theta \sqrt{S}}{\gamma} + \frac{H^2 \sqrt{S} A^{2L} O^L \zeta}{\phi \gamma}, \tag{39}
\end{aligned}
$$

where the third-to-last inequality follows from the fact that $\left| V_h^{\pi, \widehat{\mathcal{M}}, R}(a_{1:h-1}, o_{2:h}) \right| \leq \frac{H\sqrt{S}}{\gamma}$ for all $h \in [H]$ and $(a_{1:h-1}, o_{2:h}) \in \mathscr{H}_h$, and the second-to-last inequality follows from Lemma G.2, applied for each $h \in [H'-1]$. $\qquad \square$

# H Two-sided comparison inequalities

In this section, we prove several inequalities which relate $\mathcal{P}$ and $\widehat{\mathcal{M}}(\pi^{1:H})$, which are similar in spirit to those in Section G (which related $\overline{\mathcal{P}}_{\phi, H'}(\pi^{1:H})$ and $\widehat{\mathcal{M}}(\pi^{1:H})$), but are of a *two-sided* nature. Accordingly, the upper bounds we show on the difference in the value functions for any policy $\pi$ depend on the probability that $\pi$ visits the underexplored sets (Definition E.2) associated to the policies $\pi^{1:H}$.

Given $S \in \mathbb{N}$ and a distribution $Q \in \Delta([S])$, as well as a subset $\mathcal{S}' \subset [S]$, let $\mathcal{R}_{\mathcal{S}'}(Q) \in \mathbb{R}_{\geq 0}^S$ be the vector which is identical to $Q$ except that for $s \in \mathcal{S}'$, $\mathcal{R}_{\mathcal{S}'}(Q)(s) = 0$. Furthermore, recall that for a non-negative vector $\mathbf{v} \in \mathbb{R}_{\geq 0}^d$, we define $\mathfrak{n}(\mathbf{v}) \in \Delta([d])$ by, for $i \in [d]$, $\mathfrak{n}(\mathbf{v})(i) = \frac{\mathbf{v}(i)}{\sum_{j=1}^d \mathbf{v}(j)}$.

**Lemma H.1.** *If $Q \in \Delta^S$, $\mathcal{S}' \subset [S]$, then $\| Q - \mathfrak{n}(\mathcal{R}_{\mathcal{S}'}(Q)) \|_1 = 2 \cdot Q(\mathcal{S}')$.*

*Proof.* We compute

$$
\begin{aligned}
\| Q - \mathfrak{n}(\mathcal{R}_{\mathcal{S}'}(Q)) \|_1 &= Q(\mathcal{S}') + \sum_{s \notin \mathcal{S}'} \left| Q(s) - \frac{Q(s)}{1 - Q(\mathcal{S}')} \right| \\
&= Q(\mathcal{S}') + \sum_{s \notin \mathcal{S}'} Q(\mathcal{S}') \cdot \left| \frac{Q(s)}{1 - Q(\mathcal{S}')} \right| \\
&= 2 \cdot Q(\mathcal{S}').
\end{aligned}
$$

$\qquad \square$

Lemma H.2 is a corollary of the belief contraction theorem (Theorem B.2), which relaxes the constraint of Theorem B.2 that $\frac{\mathscr{D}'(s)}{\mathscr{D}(s)} \leq \frac{1}{\phi}$ for all $s \in \mathcal{S}$ (for particular types of choices for $\mathscr{D}', \mathscr{D}$). The cost of doing so is that the upper bound contains a term denoting the probability that $\pi$ visits a certain set of underexplored states.

**Lemma H.2.** *There is a constant $C \geq 1$ so that the following holds. For any $\epsilon > 0, L \in \mathbb{N}$ so that $L \geq C \cdot \min \left\{ \frac{\log(1/(\epsilon\phi)) \log(\log(1/\phi)/\epsilon)}{\gamma^2}, \frac{\log(1/(\epsilon\phi))}{\gamma^4} \right\}$, it holds that, for all general policies*

$\pi, \pi' \in \Pi^{\mathrm{gen}}$,

$$\mathbb{E}^{\mathcal{P}}_{a_{1:h-1}, o_{2:h} \sim \pi} \left\| \mathbf{b}_h(a_{1:h-1}, o_{2:h}) - \widetilde{\mathbf{b}}^{\pi'}_h(a_{h-L:h-1}, o_{h-L+1:h}) \right\|_1$$
$$\leq \epsilon + \mathbb{1}[h > L] \cdot 6 \cdot d^{\mathcal{P},\pi}_{\mathsf{S},h-L}(\mathcal{U}^{\mathcal{P}}_{\phi,h-L}(\pi')).$$

*Proof.* By linearity of expectation it suffices to prove the result for the case that $\pi$ is a deterministic policy, i.e., we have $\pi = (\pi_1, \ldots, \pi_H)$ where $\pi_h : \mathcal{A}^{h-1} \times \mathcal{O}^{h-1} \to \mathcal{A}$. Fix any such deterministic policy $\pi$ and a general policy $\pi' \in \Pi^{\mathrm{gen}}$.

Note that the lemma statement is immediate in the case that $h \leq L$, as we have $\mathbf{b}_h(a_{1:h-1}, o_{2:h}) = \widetilde{\mathbf{b}}^{\pi'}_h(a_{h-L:h-1}, o_{h-L+1:h})$ (see Section E.1). Thus, for the remainder of the proof, we may assume $h > L$.

Set $\underline{h} := h - L$. Fix any choice of $\tau := (a_{1:\underline{h}-1}, o_{2:\underline{h}}) \in \mathcal{A}^{\underline{h}-1} \times \mathcal{O}^{\underline{h}-1}$. Let $\pi^\tau$ be the policy defined at steps $\underline{h}$ onward, given the history $\tau$; in particular, for $h' \geq h$, define $\pi^\tau_{h'} : \mathcal{A}^{h'-h+L} \times \mathcal{O}^{h'-h+L} \to \mathcal{A}$ by $\pi^\tau_{h'}(a_{\underline{h}:h'-1}, o_{\underline{h}+1:h'}) = \pi_{h'}(a_{1:h'-1}, o_{2:h'})$.

Theorem B.2 gives that for any distribution $Q \in \Delta(\mathcal{S})$ so that $\frac{Q(s)}{d^{\mathcal{P},\pi'}_{\mathsf{S},\underline{h}}(s)} \leq \frac{1}{\phi}$ for all $s \in \mathcal{S}$ (with $\frac{0}{0}$ interpreted as 0), it holds that

$$\mathbb{E}^{\mathcal{P}}_{s_{\underline{h}} \sim Q, (a_{\underline{h}:h-1}, o_{\underline{h}+1:h}) \sim \pi^\tau} \left\| \mathbf{b}^{\mathrm{apx}}_h(a_{\underline{h}:h-1}, o_{\underline{h}+1:h}; Q) - \widetilde{\mathbf{b}}^{\pi'}_h(a_{\underline{h}:h-1}, o_{\underline{h}+1:h}) \right\|_1 \leq \epsilon. \qquad (40)$$

Now set $Q^\tau := \mathfrak{n}\left( \mathcal{R}_{\mathcal{U}^{\mathcal{P}}_{\phi,\underline{h}}(\pi')}(\mathbf{b}_{\underline{h}}(a_{1:\underline{h}-1}, o_{2:\underline{h}})) \right)$. By Lemma H.1, $\frac{1}{2} \left\| \mathbf{b}_{\underline{h}}(a_{1:\underline{h}-1}, o_{2:\underline{h}}) - Q^\tau \right\|_1 \leq \epsilon^\tau$, where $\epsilon^\tau = \sum_{s \in \mathcal{U}^{\mathcal{P}}_{\phi,\underline{h}}(\pi')} \mathbf{b}_{\underline{h}}(a_{1:\underline{h}-1}, o_{2:\underline{h}})(s)$. We now see that

$$\mathbb{E}^{\mathcal{P}}_{s_{\underline{h}} \sim \mathbf{b}_{\underline{h}}(a_{1:\underline{h}-1}, o_{2:\underline{h}}), (a_{\underline{h}:h-1}, o_{\underline{h}+1:h}) \sim \pi^\tau} \left\| \mathbf{b}_h(a_{1:h-1}, o_{2:h}) - \widetilde{\mathbf{b}}^{\pi'}_h(a_{\underline{h}:h-1}, o_{\underline{h}+1:h}) \right\|_1$$

$$\leq 2\epsilon^\tau + \mathbb{E}^{\mathcal{P}}_{s_{\underline{h}} \sim Q^\tau, (a_{\underline{h}:h-1}, o_{\underline{h}+1:h}) \sim \pi^\tau} \left\| \mathbf{b}_h(a_{1:h-1}, o_{2:h}) - \widetilde{\mathbf{b}}^{\pi'}_h(a_{\underline{h}:h-1}, o_{\underline{h}+1:h}) \right\|_1$$

$$\leq 2\epsilon^\tau + \mathbb{E}^{\mathcal{P}}_{s_{\underline{h}} \sim Q^\tau, (a_{\underline{h}:h-1}, o_{\underline{h}+1:h}) \sim \pi^\tau} \left\| \mathbf{b}^{\mathrm{apx}}_h(a_{\underline{h}:h-1}, o_{\underline{h}+1:h}; Q^\tau) - \widetilde{\mathbf{b}}^{\pi'}_h(a_{\underline{h}:h-1}, o_{\underline{h}+1:h}) \right\|_1$$

$$+ \mathbb{E}^{\mathcal{P}}_{s_{\underline{h}} \sim Q^\tau, (a_{\underline{h}:h-1}, o_{\underline{h}+1:h}) \sim \pi^\tau} \left\| \mathbf{b}_h(a_{1:h-1}, o_{2:h}) - \mathbf{b}^{\mathrm{apx}}_h(a_{\underline{h}:h-1}, o_{\underline{h}+1:h}; Q^\tau) \right\|_1$$

$$\leq 6\epsilon^\tau + \mathbb{E}^{\mathcal{P}}_{s_{\underline{h}} \sim Q^\tau, (a_{\underline{h}:h-1}, o_{\underline{h}+1:h}) \sim \pi^\tau} \left\| \mathbf{b}^{\mathrm{apx}}_h(a_{\underline{h}:h-1}, o_{\underline{h}+1:h}; Q^\tau) - \widetilde{\mathbf{b}}^{\pi'}_h(a_{\underline{h}:h-1}, o_{\underline{h}+1:h}) \right\|_1 \qquad (41)$$

$$\leq 6\epsilon^\tau + \epsilon, \qquad (42)$$

where (42) uses (40) together with the fact that $\frac{Q^\tau(s)}{d^{\pi',\mathcal{P}}_{\mathsf{S},\underline{h}}(s)} \leq \frac{1}{\phi}$ for all $s \in \mathcal{S}$ (as those $s \in \mathcal{S}$ with $d^{\pi',\mathcal{P}}_{\mathsf{S},\underline{h}}(s) < \phi$ have been "zeroed out" from $\mathbf{b}_{\underline{h}}(a_{1:\underline{h}-1}, o_{2:\underline{h}})$), and (41) uses the fact that

$$\mathbb{E}^{\mathcal{P}}_{s_{\underline{h}} \sim Q^\tau, (a_{\underline{h}:h-1}, o_{\underline{h}+1:h}) \sim \pi^\tau} \left\| \mathbf{b}_h(a_{1:h-1}, o_{2:h}) - \mathbf{b}^{\mathrm{apx}}_h(a_{\underline{h}:h-1}, o_{\underline{h}+1:h}; Q^\tau) \right\|_1$$

$$\leq 2 \cdot \min \left\{ \mathbb{E}^{\mathcal{P}}_{s_{\underline{h}} \sim Q^\tau, (a_{\underline{h}:h-1}, o_{\underline{h}+1:h}) \sim \pi^\tau} \left( \left\| \frac{\mathbf{b}^{\mathrm{apx}}_h(a_{\underline{h}:h-1}, o_{\underline{h}+1:h}; Q^\tau)}{\mathbf{b}_h(a_{1:h-1}, o_{2:h})} \right\|_\infty - 1 \right), 1 \right\} \qquad (43)$$

$$\leq 2 \cdot \min \left\{ \left( \left\| \frac{Q^\tau}{\mathbf{b}_{\underline{h}}(a_{1:\underline{h}-1}, o_{2:\underline{h}})} \right\|_\infty - 1 \right), 1 \right\} \qquad (44)$$

$$\leq 2 \cdot \min \left\{ \frac{\epsilon^\tau}{1 - \epsilon^\tau}, 1 \right\} \leq 4\epsilon^\tau,$$

where (43) uses the fact that for $P, Q \in \Delta^S$, $\|P - Q\|_1 \leq 2 \cdot \left( \left\| \frac{P}{Q} \right\|_\infty - 1 \right)$, and (44) uses [GMR22, Lemma 4.6].

Note that $\mathbb{E}^{\mathcal{P}}_{(a_{1:\underline{h}-1}, o_{2:\underline{h}}) \sim \pi}[\mathbf{b}_{\underline{h}}(a_{1:\underline{h}-1}, o_{2:\underline{h}})] = d^{\mathcal{P},\pi}_{\mathsf{S},\underline{h}}$, meaning that $\mathbb{E}^{\mathcal{P}}_\pi[\epsilon^\tau] = d^{\mathcal{P},\pi}_{\mathsf{S},\underline{h}}(\mathcal{U}^{\mathcal{P}}_{\phi,\underline{h}}(\pi'))$. Furthermore, note that the conditional distribution of $s_{\underline{h}}$ given $(a_{1:\underline{h}-1}, o_{2:\underline{h}})$ is exactly $\mathbf{b}_{\underline{h}}(a_{1:\underline{h}-1}, o_{2:\underline{h}})$.

Thus, taking an expectation of the display (42) over $(a_{1:\underline{h}-1}, o_{2:\underline{h}}) \sim \pi$, we get that

$$\mathbb{E}^{\mathcal{P}}_{(a_{1:h-1}, o_{2:h}) \sim \pi} \left\| \mathbf{b}_h(a_{1:h-1}, o_{2:h}) - \widetilde{\mathbf{b}}_h^{\pi'}(a_{\underline{h}:h-1}, o_{\underline{h}+1:h}) \right\|_1$$

$$\leq \epsilon + \mathbb{E}^{\mathcal{P}}_{\pi}[6\epsilon^\tau] = \epsilon + 6 \cdot d^{\mathcal{P},\pi}_{\mathsf{S},\underline{h}}(\mathcal{U}^{\mathcal{P}}_{\phi,\underline{h}}(\pi')),$$

as desired. □

Lemma H.3 is the main result of this section, which uses Lemma H.2 to show an upper bound on the absolute value of the difference between the value functions of $\mathcal{P}$ and $\widehat{\mathcal{M}}(\pi^{1:H})$, for any policy $\pi$.

**Lemma H.3.** *Consider any sequence of general policies $\pi^1, \ldots, \pi^H \in \Pi^{\mathrm{gen}}$ and $\phi > 0$. Fix any $H' \in [H]$. Consider any collection of reward functions $R = (R_1, \ldots, R_H)$, $R_h : \overline{\mathcal{O}} \to [-1, 1]$ so that $R_{h'} \equiv 0$ for all $h' > H'$. Suppose that in the construction of $\widehat{\mathcal{M}}(\pi^{1:H})$, the event $\mathcal{E}^{\mathrm{low}}$ of Lemma E.1 holds. Then for all general policies $\pi \in \Pi^{\mathrm{gen}}$,*

$$\left| V_1^{\pi,\mathcal{P},R}(\emptyset) - V_1^{\pi,\widehat{\mathcal{M}}(\pi^{1:H}),R}(\emptyset) \right| \leq H^2 \epsilon + H^2 \theta + \frac{H^2 A^{2L} O^L \zeta}{\phi} + 4H \cdot \sum_{h=L+1}^{H'-1} d^{\mathcal{P},\pi}_{\mathsf{S},h-L}(\mathcal{U}^{\mathcal{P}}_{\phi,h-L}(\pi^h)).$$

*Proof.* Fix general policies $\pi^{1:H}$, and for $h \in [H]$, let $\widehat{\pi}^h$ be the policy which follows $\pi^h$ up to step $\max\{h - L - 1, 0\}$ and thereafter chooses actions uniformly at random. Write $\widehat{\mathcal{M}} = \widehat{\mathcal{M}}(\pi^{1:H})$ and $\widetilde{\mathcal{M}} = \widetilde{\mathcal{M}}(\pi^{1:H})$. By Jensen's inequality, it suffices to consider the case that $\pi$ is a deterministic policy. Using the assumption that $R_{h'}$ is identically 0 for $h' > H'$, we have from Lemma B.1 that for all general policies $\pi$, writing $z_h = (a_{h-L:h-1}, o_{h-L+1:h})$,

$$\left| V_1^{\pi,\mathcal{P},R}(\emptyset) - V_1^{\pi,\widehat{\mathcal{M}},R}(\emptyset) \right| = \left| \mathbb{E}^{\mathcal{P}}_{(a_{1:H-1}, o_{2:H}) \sim \pi} \left[ \sum_{h=1}^{H} ((\mathbb{P}^{\mathcal{P}}_h - \mathbb{P}^{\widehat{\mathcal{M}}}_h)(V_{h+1}^{\pi,\widehat{\mathcal{M}},R} + R_{h+1}))(a_{1:h}, o_{2:h}) \right] \right|$$

$$= \left| \mathbb{E}^{\mathcal{P}}_{(a_{1:H-1}, o_{2:H}) \sim \pi} \left[ \sum_{h=1}^{H'-1} ((\mathbb{P}^{\mathcal{P}}_h - \mathbb{P}^{\widehat{\mathcal{M}}}_h)(V_{h+1}^{\pi,\widehat{\mathcal{M}},R} + R_{h+1}))(a_{1:h}, o_{2:h}) \right] \right|$$

$$\leq \frac{H}{2} \cdot \mathbb{E}^{\mathcal{P}}_{(a_{1:H-1}, o_{2:H}) \sim \pi} \left[ \sum_{h=1}^{H'-1} \left\| \mathbb{P}^{\mathcal{P}}_h(\cdot|a_{1:h}, o_{2:h}) - \mathbb{P}^{\widehat{\mathcal{M}}}_h(\cdot|z_h, a_h) \right\|_1 \right] \tag{45}$$

$$\leq \frac{H}{2} \cdot \mathbb{E}^{\mathcal{P}}_{(a_{1:H-1}, o_{2:H}) \sim \pi} \left[ \sum_{h=1}^{H'-1} \left\| \mathbb{P}^{\mathcal{P}}_h(\cdot|a_{1:h}, o_{2:h}) - \mathbb{P}^{\widetilde{\mathcal{M}}}_h(\cdot|z_h, a_h) \right\|_1 \right]$$

$$+ \frac{H}{2} \cdot \mathbb{E}^{\mathcal{P}}_{(a_{1:H-1}, o_{2:H}) \sim \pi} \left[ \sum_{h=1}^{H'-1} \left\| \mathbb{P}^{\widehat{\mathcal{M}}}_h(\cdot|z_h, a_h) - \mathbb{P}^{\widetilde{\mathcal{M}}}_h(\cdot|z_h, a_h) \right\|_1 \right], \tag{46}$$

where (45) uses the fact that $\left| \left( V_{h+1}^{\pi,\widehat{\mathcal{M}},R} + R_{h+1} \right)(a_{1:h}, o_{2:h+1}) \right| \leq H$ for all $a_{1:h}, o_{2:h+1}$. For all $h \in [H]$, $a_{1:h} \in \mathcal{A}^h$ and $o_{2:h} \in \mathcal{O}^{h-1}$, we have (with $z_h = (a_{h-L:h-1}, o_{h-l+1:h})$)

$$\mathbb{P}^{\mathcal{P}}_h(o_{h+1}|a_{1:h}, o_{2:h}) = e_{o_{h+1}}^\top \cdot \mathbb{O}^{\mathcal{P}}_{h+1} \cdot \mathbb{T}^{\mathcal{P}}_h(a_h) \cdot \mathbf{b}^{\mathcal{P}}_h(a_{1:h-1}, o_{2:h})$$

$$\mathbb{P}^{\widetilde{\mathcal{M}}}_h(o_{h+1}|z_h, a_h) = e_{o_{h+1}}^\top \cdot \mathbb{O}^{\mathcal{P}}_{h+1} \cdot \mathbb{T}^{\mathcal{P}}_h(a_h) \cdot \widetilde{\mathbf{b}}^{\pi^h}_h(a_{h-L:h-1}, o_{h-L+1:h}), \tag{47}$$

where (47) follows from (10). Thus, by the data processing inequality, for all $a_{1:h}, o_{2:h}$, (again writing $z_h = (a_{h-L:h-1}, o_{h-l+1:h})$)

$$\left\| \mathbb{P}^{\mathcal{P}}_h(\cdot|a_{1:h}, o_{2:h}) - \mathbb{P}^{\widetilde{\mathcal{M}}}_h(\cdot|z_h, a_h) \right\|_1 \leq \left\| \mathbf{b}^{\mathcal{P}}_h(a_{1:h-1}, o_{2:h}) - \widetilde{\mathbf{b}}^{\pi^h}_h(a_{h-L:h-1}, o_{h-L+1:h}) \right\|_1. \tag{48}$$

Next, by Lemma E.1 and the assumption that $\mathcal{E}^{\mathrm{low}}$ holds, we have that for all $h \in [H]$, $z_h \in \mathcal{Z}$, $a_h \in \mathcal{A}$,

$$\left\| \mathbb{P}_h^{\widehat{\mathcal{M}}}(\cdot | z_h, a_h) - \mathbb{P}_h^{\widetilde{\mathcal{M}}}(\cdot | z_h, a_h) \right\|_1 \leq \theta + 2\mathbb{1}[z_h \in \mathcal{Z}_{h,\zeta}^{\mathrm{low}}(\widehat{\pi}^h)]. \tag{49}$$

Moreover, by (48), we have

$$\mathbb{E}_{(a_{1:H-1}, o_{2:H}) \sim \pi}^{\mathcal{P}} \left[ \sum_{h=1}^{H'-1} \left\| \mathbb{P}_h^{\mathcal{P}}(\cdot | a_{1:h}, o_{2:h}) - \mathbb{P}_h^{\widetilde{\mathcal{M}}}(\cdot | a_{1:h}, o_{2:h}) \right\|_1 \right]$$

$$\leq \mathbb{E}_{(a_{1:H-1}, o_{2:H}) \sim \pi}^{\mathcal{P}} \left[ \sum_{h=1}^{H'-1} \left\| \mathbf{b}_h^{\mathcal{P}}(a_{1:h-1}, o_{2:h}) - \widetilde{\mathbf{b}}_h^{\pi^h}(a_{h-L:h-1}, o_{h-L+1:h}) \right\|_1 \right]$$

$$\leq H\epsilon + 6 \cdot \sum_{h=L+1}^{H'-1} d_{\mathsf{S},h-L}^{\mathcal{P},\pi}(\mathcal{U}_{\phi,h-L}^{\mathcal{P}}(\pi^h)), \tag{50}$$

where (50) uses Lemma H.2 for each $h$ satisfying $L+1 \leq h \leq H'-1$ (in particular, for each such value of $h$, the policy $\pi'$ in Lemma H.2 is set to $\pi^h$).

Next, using (49), we compute

$$\mathbb{E}_{(a_{1:H-1}, o_{2:H}) \sim \pi}^{\mathcal{P}} \left[ \sum_{h=1}^{H'-1} \left\| \mathbb{P}_h^{\widehat{\mathcal{M}}}(\cdot | a_{1:h}, o_{2:h}) - \mathbb{P}_h^{\widetilde{\mathcal{M}}}(\cdot | a_{1:h}, o_{2:h}) \right\|_1 \right]$$

$$\leq \theta H + 2 \cdot \sum_{h=1}^{H'-1} \mathbb{P}_{z_h \sim \pi}^{\mathcal{P}} \left( z_h \in \mathcal{Z}_{h,\zeta}^{\mathrm{low}}(\widehat{\pi}^h) \right)$$

$$\leq \theta H + 2 \cdot \frac{HA^{2L}O^L\zeta}{\phi} + 2 \cdot \sum_{h=L+1}^{H'-1} d_{\mathsf{S},h-L}^{\mathcal{P},\pi}(\mathcal{U}_{\phi,h-L}^{\mathcal{P}}(\pi^h)), \tag{51}$$

where (51) uses Lemma F.4 for each $1 \leq h \leq H'-1$: in particular, for each value of $h$, the lemma is applied with $\overline{\mathcal{P}} = \mathcal{P}$ and $\pi' = \widehat{\pi}^h$.

Combining (46), (50), and (51), we obtain that

$$\left| V_1^{\pi,\mathcal{P},R}(\emptyset) - V_1^{\pi,\widehat{\mathcal{M}},R}(\emptyset) \right| \leq H^2\epsilon + H^2\theta + \frac{H^2 A^{2L}O^L\zeta}{\phi} + 4H \cdot \sum_{h=L+1}^{H'-1} d_{\mathsf{S},h-L}^{\mathcal{P},\pi}(\mathcal{U}_{\phi,h-L}^{\mathcal{P}}(\pi^h)),$$

as desired. $\qquad \square$

# I   Analysis of `BaSeCAMP` (Algorithm 3)

In this section we prove Theorem 3.1. We first show the following technical lemma, which relates the underexplored sets (Definition E.2) of the truncated POMDPs $\overline{\mathcal{P}}_{\phi,h}(\pi^{1:H})$ and $\mathcal{P}$.

**Lemma I.1.** *Consider any sequence of general policies $\pi^{1:H,0}$, and define $\pi^h := \frac{1}{H-h+1}\sum_{h'=h}^{H} \pi^{h',0}$ for each $h \in [H]$. Then, for each $h$ satisfying $L < h \leq H$ and any $\phi > 0$,*

$$\mathcal{U}_{\phi,h-L}^{\overline{\mathcal{P}}_{\phi,h}(\pi^{1:H})}(\pi^h) \subset \mathcal{U}_{\phi \cdot H^2 S,h-L}^{\mathcal{P}}(\pi^h).$$

*Proof.* Fix $\phi$ and $\pi^{1:H}$, and write $\overline{\mathcal{P}}_h := \overline{\mathcal{P}}_{\phi,h}(\pi^{1:H})$.

Consider any $h > L$, and any $s \in \mathcal{U}_{\phi,h-L}^{\overline{\mathcal{P}}_h}(\pi^h)$; in particular, we have $d_{\mathsf{S},h-L}^{\overline{\mathcal{P}}_h,\pi^h}(s) \leq \phi$. By item 1 of Lemma F.2, it holds that

$$d_{\mathsf{S},h-L}^{\mathcal{P},\pi^h}(s) \leq d_{\mathsf{S},h-L}^{\overline{\mathcal{P}}_h,\pi^h}(s) + d_{\mathsf{S},h-L}^{\overline{\mathcal{P}}_h,\pi^h}(s^{\mathrm{sink}}). \tag{52}$$

For each $H' > L$, write $\mathcal{U}_{H'-L} := \mathcal{U}_{\phi,H'-L}^{\overline{\mathcal{P}}_{H'-1}}(\pi^{H'})$; in words, $\mathcal{U}_{H'-L}$ is the set of underexplored states in $\overline{\mathcal{P}}_{H'-1}$ at step $H' - L$ which all mass is diverted away from in the construction of $\overline{\mathcal{P}}_{H'}$ (see Section E.3). It holds that

$$d_{\mathsf{S},h-L}^{\overline{\mathcal{P}}_h,\pi^h}(s^{\mathrm{sink}}) = \sum_{H'=L+1}^{h} d_{\mathsf{S},H'-L}^{\overline{\mathcal{P}}_{H'-1},\pi^h}(\mathcal{U}_{H'-L})$$

$$\leq H \cdot \sum_{H'=L+1}^{h} d_{\mathsf{S},H'-L}^{\overline{\mathcal{P}}_{H'-1},\pi^{H'}}(\mathcal{U}_{H'-L}) \tag{53}$$

$$\leq H \cdot \sum_{H'=L+1}^{h} S\phi \leq H(H-1)S\phi, \tag{54}$$

where (53) follows from the fact that for all $s \in \mathcal{S}$ and any $H' \leq h$,

$$d_{\mathsf{S},H'-L}^{\overline{\mathcal{P}}_{H'-1},\pi^h}(s) = \frac{1}{H-h+1} \sum_{h'=h}^{H} d_{\mathsf{S},H'-L}^{\overline{\mathcal{P}}_{H'-1},\pi^{h',0}}(s)$$

$$\leq H \cdot \frac{1}{H-H'+1} \sum_{h'=H'}^{H} d_{\mathsf{S},H'-L}^{\overline{\mathcal{P}}_{H'-1},\pi^{h',0}}(s)$$

$$= H \cdot d_{\mathsf{S},H'-L}^{\overline{\mathcal{P}}_{H'-1},\pi^{H'}}(s),$$

and (54) follows by definition of $\mathcal{U}_{H'-L}$. From (52) it follows that $d_{\mathsf{S},h-L}^{\mathcal{P},\pi^h}(s) \leq \phi + H(H-1)S\phi \leq H^2 S\phi$, i.e., $s \in \mathcal{U}_{\phi \cdot H^2 S,h-L}^{\mathcal{P}}(\pi^h)$, as desired. $\qquad\square$

Lemma I.2 is the main technical lemma in the proof of Theorem 3.1, and shows that for each iteration $k$ of BaSeCAMP, one of two options must hold: option 1 implies that the policies $\overline{\pi}^{k,1:H}$ of BaSeCAMP have sufficiently explored $\mathcal{P}$ and will be used later to show that the optimal policy for $\widehat{\mathcal{M}}^{(k)}$ is an approximately optimal policy for $\mathcal{P}$. Option 2 states that progress is made by BaSeCAMP at iteration $k$ in sense that some new latent state $(s, h)$ of $\mathcal{P}$ must be explored. The proof of the lemma proceeds by combining the results of Section G and H which show that the true POMDP $\mathcal{P}$ is "sufficiently close" to the approximating MDPs $\widehat{\mathcal{M}}^{(k)}$, with the result of Section D which states that the policy obtained from a barycentric spanner (as in Algorithm 2) has desirable exploration properties.

**Lemma I.2** ("Progress Lemma"; formal version of Lemma 5.1). *Consider the execution of Algorithm 3 given the values of the parameters defined in Section C.1. Fix any $k \in [K]$ in step 3 of Algorithm 3 and suppose that, in the construction of $\widehat{\mathcal{M}}^{(k)}$ in step 5, the event $\mathcal{E}^{\mathrm{low}}$ of Lemma E.1 holds. Then one of the two following statements holds:*

1. *For all general policies $\pi$, it holds that $d_{\mathsf{S},H-L}^{\overline{\mathcal{P}}_H(\overline{\pi}^{k,1:H}),\pi}(s^{\mathrm{sink}}) \leq \delta H$.*

2. *There is $H' \in [H]$ and $(h,s) \in [H'] \times \mathcal{S}$, so that $h > L$ and the following holds:*

$$(h-L,s) \in \mathcal{U}_{\phi \cdot H^2 S,h-L}^{\mathcal{P}}(\overline{\pi}^{k,h}) \quad and \quad d_{\mathsf{S},h-L}^{\mathcal{P},\pi^{k+1,H',0}}(s) \geq \frac{\delta' \cdot \gamma}{56 H S^{3/2} O^2}.$$

*Proof.* Fix $k \in [K]$. For $h \in [H]$, we will write $\pi^h := \overline{\pi}^{k,h}$ for the course of the proof. Furthermore, according to our convention, let $\widehat{\pi}^h$ denote the policy which follows $\pi^h$ for the first $\max\{h - L - 1, 0\}$ steps and thereafter chooses uniformly random actions. For each $H' \in [H]$, define $\overline{\mathcal{P}}_{H'} := \overline{\mathcal{P}}_{\phi,H'}(\pi^{1:H})$ and $\widehat{\mathcal{M}} = \widehat{\mathcal{M}}(\pi^{1:H}) = \widehat{\mathcal{M}}^{(k)}$.

We consider two cases:

**Case 1.** First, suppose that for all general policies $\pi \in \Pi^{\mathrm{gen}}$ and all $H' \in [H]$ satisfying $H' > L$, it holds that

$$d_{\mathsf{S},H'-L}^{\overline{\mathcal{P}}_{H'-1},\pi}\left(\mathcal{U}_{\phi,H'-L}^{\overline{\mathcal{P}}_{H'-1}}(\pi^{H'})\right) \leq \delta. \tag{55}$$

We now claim that

$$d_{\mathsf{S},H-L}^{\overline{\mathcal{P}}_H,\pi}\left(s^{\mathrm{sink}}\right) \leq \delta \cdot H. \tag{56}$$

To see that (56) holds, we argue by induction, noting that $d_{\mathsf{S},L}^{\overline{\mathcal{P}}_L,\pi}(s^{\mathrm{sink}}) = 0$ since $\overline{\mathcal{P}}_L = \mathcal{P}$. Next, we note that for each $H' \geq L+1$,

$$\begin{aligned}
d_{\mathsf{S},H'-L}^{\overline{\mathcal{P}}_{H'},\pi}\left(s^{\mathrm{sink}}\right) &= d_{\mathsf{S},H'-L-1}^{\overline{\mathcal{P}}_{H'},\pi}(s^{\mathrm{sink}}) + d_{\mathsf{S},H'-L}^{\overline{\mathcal{P}}_{H'-1},\pi}\left(\mathcal{U}_{\phi,H'-L}^{\overline{\mathcal{P}}_{H'-1}}(\pi^{H'})\right) \\
&= d_{\mathsf{S},H'-L-1}^{\overline{\mathcal{P}}_{H'-1},\pi}(s^{\mathrm{sink}}) + d_{\mathsf{S},H'-L}^{\overline{\mathcal{P}}_{H'-1},\pi}\left(\mathcal{U}_{\phi,H'-L}^{\overline{\mathcal{P}}_{H'-1}}(\pi^{H'})\right) \\
&\leq \delta \cdot H' + \delta = \delta \cdot (H'+1),
\end{aligned}$$

where the first equality holds by definition of $\overline{\mathcal{P}}_{H'+1}$ (see Section E.3), the second equality holds since the transitions at steps $1, 2, \ldots, H' - L - 2$ of $\overline{\mathcal{P}}_{H'-1}, \overline{\mathcal{P}}_{H'}$ are identical, and the inequality follows by assumption (55) and the inductive hypothesis. Thus, in this case, we have established that (56) holds (i.e., item 1 in the lemma statement holds).

**Case 2.** Next suppose that the previous case does not hold, and choose $H' > L$ as small as possible so that, for some $\pi^\star \in \Pi^{\mathrm{gen}}$,

$$d_{\mathsf{S},H'-L}^{\overline{\mathcal{P}}_{H'-1},\pi^\star}\left(\mathcal{U}_{\phi,H'-L}^{\overline{\mathcal{P}}_{H'-1}}(\pi^{H'})\right) > \delta. \tag{57}$$

We will now apply Lemma G.4 with the value of $H'$ in Lemma G.4 set to $H' - L$, the value of $H''$ in Lemma G.4 set to $H' - 1$, the value of $\phi$ in Lemma G.4 set to the value of $\phi$ defined in Section C.1, $\pi^{1:H}$ set to the present value of $\pi^{1:H}$, and $\widehat{\mathcal{M}}(\pi^{1:H})$ set to $\widehat{\mathcal{M}} = \widehat{\mathcal{M}}^{(k)}$. We also need to define the rewards $R = (R_1, \ldots, R_H)$, where each $R_h : \overline{\mathcal{O}} \to \mathbb{R}$. We set each $R_h$, $h \neq H' - L$, to be identically 0. Furthermore, for $o \in \overline{\mathcal{O}}$, we define

$$R_{H'-L}(o) := \sum_{s \in \mathcal{S}} \mathbb{1}\left[s \in \mathcal{U}_{\phi,H'-L}^{\overline{\mathcal{P}}_{H'-1}}(\pi^{H'})\right] \cdot \left(\mathbb{O}_{H'-L}^{\mathcal{P}}\right)_{s,o}^\dagger.$$

Note that the rewards $R$ satisfy the requirements of Lemma G.4. The definition of $R$ ensures that, for all $\pi \in \Pi^{\mathrm{gen}}$,

$$d_{\mathsf{S},H'-L}^{\overline{\mathcal{P}}_{H'-1},\pi}(\mathcal{U}_{\phi,H'-L}^{\overline{\mathcal{P}}_{H'-1}}(\pi^{H'})) = \sum_{s \in \mathcal{U}_{\phi,H'-L}^{\overline{\mathcal{P}}_{H'-1}}(\pi^{H'})} \begin{cases} \left\langle e_s, \left(\mathbb{O}_{H'-L}^{\mathcal{P}}\right)^\dagger \cdot d_{\mathsf{O},H'-L}^{\widehat{\mathcal{M}},\pi}\right\rangle = V_1^{\pi,\widehat{\mathcal{M}},R}(\emptyset) \\[2mm] \left\langle e_s, \left(\mathbb{O}_{H'-L}^{\mathcal{P}}\right)^\dagger \cdot d_{\mathsf{O},H'-L}^{\overline{\mathcal{P}}_{H'-1},\pi}\right\rangle = V_1^{\pi,\overline{\mathcal{P}}_{H'-1},R}(\emptyset). \end{cases}$$

Thus, by Lemma G.4, the fact that $\mathcal{E}^{\mathrm{low}}$ holds for the construction of $\widehat{\mathcal{M}}$, and (57), it follows that

$$\begin{aligned}
&\sum_{s \in \mathcal{U}_{\phi,H'-L}^{\overline{\mathcal{P}}_{H'-1}}(\pi^{H'})} \left\langle e_s, \left(\mathbb{O}_{H'-L}^{\mathcal{P}}\right)^\dagger \cdot d_{\mathsf{O},H'-L}^{\widehat{\mathcal{M}},\pi^\star}\right\rangle \\
&= V_1^{\pi^\star,\widehat{\mathcal{M}},R}(\emptyset) \\
&\geq V_1^{\pi^\star,\overline{\mathcal{P}}_{H'-1},R}(\emptyset) - \left(\frac{H^2\sqrt{S}\epsilon}{\gamma} + \frac{2H^3\sqrt{S}\theta}{\gamma} + \frac{H^2 A^{2L} O^L \sqrt{S}\zeta}{\phi\gamma}\right) \\
&\geq \delta - \left(\frac{H^2\sqrt{S}\epsilon}{\gamma} + \frac{2H^3\sqrt{S}\theta}{\gamma} + \frac{H^2 A^{2L} O^L \sqrt{S}\zeta}{\phi\gamma}\right) \geq \delta',
\end{aligned}$$

where we have used the definitions of the parameters given in Section C.1: in particular, since $\theta = \epsilon$ and $\frac{\zeta \cdot A^{2L} O^L}{\phi} \leq \epsilon$, we need that $\delta - \frac{H^3\sqrt{S}}{\gamma} \cdot 4\epsilon \geq \delta'$, which holds since $\delta = 2\delta' = C^\star \cdot \frac{O^4 H^3 \sqrt{S}}{\gamma} \cdot \epsilon$ for some constant $C^\star > 1$ and $C^\star$ is sufficiently large.

Since $\widehat{\mathcal{M}}$ is an MDP on the state space $\overline{\mathcal{Z}}$, and any MDP has a deterministic Markov policy achieving its optimal value, there must be some $\sigma^\star \in \Pi_{\mathsf{Z}}^{\mathrm{markov}}$ so that

$$\sum_{s \in \mathcal{U}_{\phi, H'-L}^{\overline{\mathcal{P}}_{H'-1}}(\pi^{H'})} \left\langle e_s, \left(\mathbb{O}_{H'-L}^{\mathcal{P}}\right)^\dagger \cdot d_{\mathsf{O}, H'-L}^{\widehat{\mathcal{M}}, \sigma^\star} \right\rangle = V_1^{\sigma^\star, \widehat{\mathcal{M}}, R}(\emptyset) \geq \delta'. \tag{58}$$

We will now apply Lemma D.2 with the following settings (importantly, the values of $\mathcal{X}, \mathbb{M}$ below are the same as the settings used in step 4 of Algorithm 2 in the construction of the policies $\pi^{k+1, H', 0}$):

$$\eta = \delta', \quad \mathcal{X} = \left\{ d_{\mathsf{O}, H'-L}^{\widehat{\mathcal{M}}, \pi} : \pi \in \Pi_{\mathsf{Z}}^{\mathrm{markov}} \right\}, \quad \mathbb{M} = \left(\mathbb{O}_{H'-L}^{\mathcal{P}}\right)^\dagger.$$

We must first verify that the preconditions of the lemma hold: in particular, it suffices to show that for any subset $\mathcal{S}' \subset \mathcal{S}$, and any $\pi \in \Pi_{\mathsf{Z}}^{\mathrm{markov}} \subset \Pi^{\mathrm{gen}}$,

$$\left\langle \sum_{s \in \mathcal{S}'} e_s, \left(\mathbb{O}_{H'-L}^{\mathcal{P}}\right)^\dagger \cdot d_{\mathsf{O}, H'-L}^{\widehat{\mathcal{M}}, \pi} \right\rangle \geq -\frac{\delta'}{4O^2}. \tag{59}$$

To do so, we will again apply Lemma G.4, with the same parameter settings as previously, except with different reward functions: in particular, we define the rewards $R = (R_1, \ldots, R_H)$, for $R_h : \overline{\mathcal{O}} \to \mathbb{R}$, as follows. We set each $R_h$, $h \neq H' - L$, to be identically 0. Furthermore, for $o \in \overline{\mathcal{O}}$, we define

$$R_{H'-L}(o) := \sum_{s \in \mathcal{S}} \mathbb{1}\left[s \in \mathcal{S}'\right] \cdot \left(\mathbb{O}_{H'-L}^{\mathcal{P}}\right)_{s, o}^\dagger.$$

Then the following equalities hold true:

$$V_1^{\pi, \widehat{\mathcal{M}}, R}(\emptyset) = \left\langle \sum_{s \in \mathcal{S}'} e_s, \left(\mathbb{O}_{H'-L}^{\mathcal{P}}\right)^\dagger \cdot d_{\mathsf{O}, H'-L}^{\widehat{\mathcal{M}}, \pi} \right\rangle \tag{60}$$

$$V_1^{\pi, \overline{\mathcal{P}}_{H'-1}, R}(\emptyset) = \left\langle \sum_{s \in \mathcal{S}'} e_s, \left(\mathbb{O}_{H'-L}^{\mathcal{P}}\right)^\dagger \cdot d_{\mathsf{O}, H'-L}^{\overline{\mathcal{P}}_{H'-1}, \pi} \right\rangle.$$

Since, for each $s \in \mathcal{S}' \subset \mathcal{S}$, it holds that

$$\left\langle e_s, \left(\mathbb{O}_{H'-L}^{\mathcal{P}}\right)^\dagger \cdot d_{\mathsf{O}, H'-L}^{\overline{\mathcal{P}}_{H'-1}, \pi} \right\rangle = \left\langle e_s, d_{\mathsf{S}, H'-L}^{\overline{\mathcal{P}}_{H'-1}, \pi} \right\rangle = d_{\mathsf{S}, H'-L}^{\overline{\mathcal{P}}_{H'-1}, \pi}(s) \geq 0,$$

we have that $V_1^{\pi, \overline{\mathcal{P}}_{H'-1}, R}(\emptyset) \geq 0$, which means, by Lemma G.4 and (60), that (59) holds as long as

$$\frac{\delta'}{4O^2} \geq \frac{H^2 \sqrt{S} \epsilon}{\gamma} + \frac{2H^3 \sqrt{S} \theta}{\gamma} + \frac{H^2 A^{2L} O^L \sqrt{S} \zeta}{\phi \gamma},$$

which is true by our parameter settings in Section C.1 (in particular, since $\theta = \epsilon$ and $\frac{\zeta A^{2L} O^L}{\phi} = \epsilon$, we need that $\frac{\delta'}{4O^2} \geq \frac{H^3 \sqrt{S}}{\gamma} \cdot 4\epsilon$, which holds as long as $C^\star$ is sufficiently large).

By Lemma D.2 and (58), it holds that the policy $\pi^{k+1, H', 0}$ output by Algorithm 2 satisfies

$$\sum_{s \in \mathcal{U}_{\phi, H'-L}^{\overline{\mathcal{P}}_{H'-1}}(\pi^{H'})} \left\langle e_s, \left(\mathbb{O}_{H'-L}^{\mathcal{P}}\right)^\dagger \cdot d_{\mathsf{O}, H'-L}^{\widehat{\mathcal{M}}, \pi^{k+1, H', 0}} \right\rangle \geq \frac{\delta'}{4O^2}. \tag{61}$$

We next apply Lemma H.3, with the present values of $H'$, $\pi^{1:H}$ and $\phi$ (namely, the $\phi$ defined in Section C.1), and the following setting of $R = (R_1, \ldots, R_H)$: $R_{h'} \equiv 0$ for all $h' \neq H' - L$, and, for $o \in \overline{\mathcal{O}}$,

$$R_{H'-L}(o) := \sum_{s \in \mathcal{U}_{\phi, H'-L}^{\overline{\mathcal{P}}_{H'-1}}(\pi^{H'})} \left(\mathbb{O}_{H'-L}^{\mathcal{P}}\right)_{s, o}^\dagger.$$

Since $|R_{H'-L}(o)| \leq \frac{\sqrt{S}}{\gamma}$ for all $o \in \overline{\mathcal{O}}$ (which follows from Lemma B.3), we have from Lemma H.3 (and the fact that $\mathcal{E}^{\mathrm{low}}$ holds in the construction of $\widehat{\mathcal{M}}$) that

$$\left| d_{\mathsf{S},H'-L}^{\mathcal{P},\pi^{k+1,H',0}}(\mathcal{U}_{\phi,H'-L}^{\overline{\mathcal{P}}_{H'-1}}(\pi^{H'})) - \sum_{s \in \mathcal{U}_{\phi,H'-L}^{\overline{\mathcal{P}}_{H'-1}}(\pi^{H'})} \left\langle e_s, (\mathbb{O}_{H'-L}^{\mathcal{P}})^{\dagger} \cdot d_{\mathsf{O},H'-L}^{\widehat{\mathcal{M}},\pi^{k+1,H',0}} \right\rangle \right|$$

$$= \left| V_1^{\pi^{k+1,H',0},\mathcal{P},R}(\emptyset) - V_1^{\pi^{k+1,H',0},\widehat{\mathcal{M}},R}(\emptyset) \right|$$

$$\leq \frac{\sqrt{S}}{\gamma} \cdot \left( H^2 \epsilon + H^2 \theta + \frac{H^2 A^{2L} O^L \zeta}{\phi} + 4H \cdot \sum_{h=L+1}^{H'-1} d_{\mathsf{S},h-L}^{\pi^{k+1,H',0},\mathcal{P}}(\mathcal{U}_{\phi,h-L}^{\mathcal{P}}(\pi^h)) \right)$$

$$\leq \frac{\delta'}{80^2} + \frac{4H\sqrt{S}}{\gamma} \cdot \sum_{h=L+1}^{H'-1} d_{\mathsf{S},h-L}^{\mathcal{P},\pi^{k+1,H',0}}(\mathcal{U}_{\phi,h-L}^{\mathcal{P}}(\pi^h)),$$

where the final inequality follows since

$$\frac{\sqrt{S}}{\gamma} \cdot \left( H^2 \epsilon + H^2 \theta + \frac{H^2 A^{2L} O^L \zeta}{\phi} \right) \leq \frac{\delta'}{8O^2},$$

which holds by our parameter settings in Section C.1. In particular, since we have $\theta = \epsilon$ and $\frac{A^{2L} O^L \zeta}{\phi} \leq \epsilon$, it suffices to have $\frac{\delta'}{8O^2} \geq \frac{\sqrt{S}H^2}{\gamma} \cdot 3\epsilon$, which holds as long as the constant $C^{\star}$ is sufficiently large. Thus, using (61), we get that

$$d_{\mathsf{S},H'-L}^{\mathcal{P},\pi^{k+1,H',0}}(\mathcal{U}_{\phi,H'-L}^{\overline{\mathcal{P}}_{H'-1}}(\pi^{H'})) \geq \frac{\delta'}{8O^2} - \frac{4H\sqrt{S}}{\gamma} \cdot \sum_{h=L+1}^{H'-1} d_{\mathsf{S},h-L}^{\mathcal{P},\pi^{k+1,H',0}}(\mathcal{U}_{\phi,h-L}^{\mathcal{P}}(\pi^h)). \qquad (62)$$

We next apply Lemma I.1, with $\pi^{h,0}$ in the lemma statement set to $\overline{\pi}^{k,h,0} := \frac{1}{k}\sum_{k'=1}^{k} \pi^{k',h,0}$. Then we have

$$\pi^h = \overline{\pi}^{k,h} = \frac{1}{k}\sum_{k'=1}^{k} \frac{1}{H-h+1} \sum_{h'=h}^{H} \pi^{k',h',0} = \frac{1}{H-h+1}\sum_{h'=h}^{H} \overline{\pi}^{k,h',0},$$

so Lemma I.1 gives that $\mathcal{U}_{\phi,H'-L}^{\overline{\mathcal{P}}_{H'-1}}(\pi^{H'}) \subset \mathcal{U}_{\phi \cdot H^2 S, H'-L}^{\mathcal{P}}(\pi^{H'})$, so rearranging (62), we get

$$d_{\mathsf{S},H'-L}^{\mathcal{P},\pi^{k+1,H',0}}(\mathcal{U}_{\phi \cdot H^2 S, H'-L}^{\mathcal{P}}(\pi^{H'})) + \frac{4H\sqrt{S}}{\gamma} \cdot \sum_{h=L+1}^{H'-1} d_{\mathsf{S},h-L}^{\mathcal{P},\pi^{k+1,H',0}}(\mathcal{U}_{\phi,h-L}^{\mathcal{P}}(\pi^h)) \geq \frac{\delta'}{8O^2}.$$

Using that $\mathcal{U}_{\phi,h-L}^{\mathcal{P}}(\pi^h) \subset \mathcal{U}_{\phi \cdot H^2 S, h-L}^{\mathcal{P}}(\pi^h)$ for all $h$ and simplifying gives

$$\frac{4H\sqrt{S}}{\gamma} \cdot \sum_{h=L+1}^{H'} d_{\mathsf{S},h-L}^{\mathcal{P},\pi^{k+1,H',0}}(\mathcal{U}_{\phi \cdot H^2 S, h-L}^{\mathcal{P}}(\pi^h)) \geq \frac{\delta'}{8O^2}.$$

It then follows that there is some $(h,s) \in [H'] \times \mathcal{S}$ so that $h > L$, $(h-L, s) \in \mathcal{U}_{\phi \cdot H^2 S}^{\mathcal{P}}(\pi^h)$ and

$$d_{\mathsf{S},h-L}^{\mathcal{P},\pi^{k+1,H',0}}(s) \geq \frac{\delta' \cdot \gamma}{32H^2 S^{3/2} O^2},$$

which means that item (2) of the lemma statement holds. $\qquad \square$

Recall that in Section C.1 we set $K = 2HS$. Lemma I.3 formalizes the intuition that item 2 of Lemma I.2 can only hold for $HS$ iterations, showing that indeed item 1 must hold for some iteration $k \in [K]$ (using that $K > HS$).

**Lemma I.3.** *Suppose that the event $\mathcal{E}^{\mathrm{low}}$ (of Lemma E.1) holds for the construction of $\widehat{\mathcal{M}}^{(k)}$, for all rounds $k$ in Algorithm 3. Then at least $K/2$ iterations over $k$ in Algorithm 3 satisfy the following: for all $\pi \in \Pi^{\mathrm{gen}}$, $d_{\mathsf{S},H-L}^{\overline{\mathcal{P}}_{\phi,H}(\overline{\pi}^{k,1:H}),\pi}(s^{\mathrm{sink}}) \leq \delta H$.*

*Proof.* Let the set of $k \in [K]$ so that there is some $\pi \in \Pi^{\text{gen}}$ with $d^{\overline{\mathcal{P}}_{\phi,H}(\overline{\pi}^{k,1:H}),\pi}_{\mathsf{S},H-L}(s^{\text{sink}}) > \delta H$ be denoted by $\mathcal{K}_0 \subset [K]$. Lemma I.2 gives that for each $k \in \mathcal{K}_0$, there is $H' \in [H]$, $(h,s) \in [H'] \times \mathcal{S}$ so that $h > L$ and $(h-L,s) \in \mathcal{U}^{\mathcal{P}}_{\phi \cdot H^2 S, h-L}(\overline{\pi}^{k,h})$ so that $d^{\mathcal{P},\pi^{k+1,H',0}}_{\mathsf{S},h-L}(s) \geq \frac{\delta' \cdot \gamma}{56 H S^{3/2} O^2}$. For all $k' \in [K]$ with $k' \geq k+1$, $\overline{\pi}^{k',h}$ is a mixture policy which puts weight at least $\frac{1}{K} \cdot \frac{1}{H} = \frac{1}{2H^2 S}$ on $\pi^{k+1,H',0}$, meaning that for all such $k'$, it holds that

$$d^{\mathcal{P},\overline{\pi}^{k',h}}_{\mathsf{S},h-L}(s) \geq \frac{1}{2H^2 S} \cdot \frac{\delta' \cdot \gamma}{56 H S^{3/2} O^2} > \phi \cdot H^2 S,$$

which means that $(h-L,s) \notin \mathcal{U}^{\mathcal{P}}_{\phi \cdot H^2 S, h-L}(\overline{\pi}^{k',h})$. The second inequality above holds since we have, in Section C.1, set $\phi = \frac{\epsilon \cdot \gamma}{C^\star \cdot H^5 S^{7/2} O^2} < \frac{\delta' \cdot \gamma}{C^\star \cdot H^5 S^{7/2} O^2}$ for a sufficiently large constant $C^\star$.

Thus, if $|\mathcal{K}_0| > HS$, then letting $k^\star$ be the largest element of $\mathcal{K}_0$, we get that for all $h \in [H]$ with $h > L, \mathcal{U}^{\mathcal{P}}_{\phi \cdot H^2 S, h-L}(\overline{\pi}^{k^\star,h})$ must be empty, which contradicts Lemma I.2. Hence $|\mathcal{K}_0| \leq HS = K/2$, as desired. $\square$

Lemma I.4 shows that, if the property in item 1 of Lemma I.2 is satisfied, then the value functions of $\mathcal{P}$ and $\widehat{\mathcal{M}}(\pi^{1:H})$ are close under any policy. This result is similar to Lemma H.3 (which is used in its proof), but unlike Lemma H.3, the upper bound does not depend on the probability of reaching any underexplored set. Instead, such a probability is upper bounded by the assumption that item 1 of Lemma I.2 holds.

**Lemma I.4.** *Let $\phi > 0$ and suppose $\pi^{1:H}$ are general policies so that $\overline{\mathcal{P}}_{\phi,H}(\pi^{1:H})$ satisfies the following, for some $\delta > 0$: for all general policies $\pi$, $d^{\overline{\mathcal{P}}_{\phi,H}(\pi^{1:H}),\pi}_{\mathsf{S},H-L}(s^{\text{sink}}) \leq \delta$. Suppose that, in the construction of $\widehat{\mathcal{M}}(\pi^{1:H})$, the event $\mathcal{E}^{\text{low}}$ of Lemma E.1 holds. Then, for all $h \in [H]$ with $h > L$, we have*

$$d^{\mathcal{P},\pi}_{\mathsf{S},h-L}(\mathcal{U}^{\mathcal{P}}_{\phi,h-L}(\pi^h)) \leq S\delta. \tag{63}$$

*Furthermore, for any reward functions $R = (R_1, \ldots, R_H)$, $R_h : \overline{\mathcal{O}} \to [0,1]$, it holds that*

$$\left| V^{\pi,\mathcal{P},R}_1(\emptyset) - V^{\pi,\widehat{\mathcal{M}}(\pi^{1:H}),R}_1(\emptyset) \right| \leq H^2 \epsilon + H^2 \theta + \frac{H^2 A^{2L} O^L \zeta}{\phi} + 4H^2 S\delta.$$

*Proof.* Since $\mathcal{E}^{\text{low}}$ holds, Lemma H.3 with the given values of $\phi, \pi^{1:H}$ and $H' = H$ gives that, for all general policies $\pi$,

$$\left| V^{\pi,\mathcal{P},R}_1(\emptyset) - V^{\pi,\widehat{\mathcal{M}}(\pi^{1:H}),R}_1(\emptyset) \right| \leq H^2 \epsilon + H^2 \theta + \frac{H^2 A^{2L} O^L \zeta}{\phi} + 4H \cdot \sum_{h=L+1}^{H} d^{\mathcal{P},\pi}_{\mathsf{S},h-L}(\mathcal{U}^{\mathcal{P}}_{\phi,h-L}(\pi^h)).$$

Thus it suffices to bound $d^{\mathcal{P},\pi}_{\mathsf{S},h-L}(\mathcal{U}^{\mathcal{P}}_{\phi,h-L}(\pi^h))$ for each $h > L$. For this purpose, we first use item 1 of Lemma F.2 to get that for all $s \in \mathcal{S}$ and $h > L$, and all $\pi \in \Pi^{\text{gen}}$,

$$d^{\mathcal{P},\pi}_{\mathsf{S},h-L}(s) \leq d^{\overline{\mathcal{P}}_{\phi,h}(\pi^{1:H}),\pi}_{\mathsf{S},h-L}(s) + d^{\overline{\mathcal{P}}_{\phi,h}(\pi^{1:H}),\pi}_{\mathsf{S},h-L}(s^{\text{sink}}). \tag{64}$$

By items 2 and 3 of Lemma F.2 and our assumption in the statement of the present lemma that $d^{\overline{\mathcal{P}}_{\phi,H}(\pi^{1:H}),\pi}_{\mathsf{S},H-L}(s^{\text{sink}}) \leq \delta$, it holds that

$$d^{\overline{\mathcal{P}}_{\phi,h}(\pi^{1:H}),\pi}_{\mathsf{S},h-L}(s^{\text{sink}}) \leq d^{\overline{\mathcal{P}}_{\phi,H}(\pi^{1:H}),\pi}_{\mathsf{S},H-L}(s^{\text{sink}}) \leq \delta.$$

Lemma F.1 gives that for all $h \in [H], \mathcal{U}^{\mathcal{P}}_{\phi,h-L}(\pi^h) \subset \mathcal{U}^{\overline{\mathcal{P}}_{\phi,h}(\pi^{1:H})}_{\phi,h-L}(\pi^h)$. But Lemma E.2 with $H' = h$ ensures that for any $s \in \mathcal{U}^{\overline{\mathcal{P}}_{\phi,h}(\pi^{1:H})}_{\phi,h-L}(\pi^h)$, we have $d^{\overline{\mathcal{P}}_{\phi,h}(\pi^{1:H}),\pi}_{\mathsf{S},h-L}(s) = 0$. Thus, from (64) we have

$$d^{\mathcal{P},\pi}_{\mathsf{S},h-L}(\mathcal{U}^{\mathcal{P}}_{\phi,h-L}(\pi^h)) \leq S\delta,$$

which gives the desired result.

$\square$

Finally, using Lemma I.4, the proof of Theorem 3.1 is straightforward.

*Proof of Theorem 3.1.* Let $\mathcal{E}^1$ be the event that the event $\mathcal{E}^{\text{low}}$ of Lemma E.1 holds in the construction of $\widehat{\mathcal{M}}^{(k)}$ in Algorithm 3 for all $k \in [K]$. Then, by Lemma E.1, the probability of $\mathcal{E}^1$ is at least $1 - pK \geq 1 - \beta/2$ (where we have used $p = \beta/(2K)$ as defined in Section C.1).

By Lemma I.3, under the event $\mathcal{E}^1$, there must be some iteration $k$ in Algorithm 3 so that, for all $\pi \in \Pi^{\text{gen}}$, $d_{\mathsf{S},H-L}^{\overline{\mathcal{P}}_{\phi,H}(\overline{\pi}^{k,1:H}),\pi}(s^{\text{sink}}) \leq \delta H$. Then by Lemma I.4, for any reward function $R$ with values in $[0,1]$, it holds that, under $\mathcal{E}^1$,

$$\left| V_1^{\pi,\mathcal{P},R}(\emptyset) - V_1^{\pi,\widehat{\mathcal{M}}^{(k)},R}(\emptyset) \right| \leq H^2\epsilon + H^2\theta + \frac{H^2 A^{2L} O^L \zeta}{\phi} + 4H^3 S\delta \leq \frac{\alpha}{8}, \qquad (65)$$

where the final inequality follows by definition of $\alpha$ in Section C.1, assuming the constant $C^\star$ is large enough.

For each $h \in [H]$, let $\pi^h := \overline{\pi}^{k,h}$, and $\widehat{\pi}^h$ denote the policy which follows $\overline{\pi}^{k,h}$ up to step $\max\{h - L - 1, 0\}$ and thereafter chooses uniformly random actions. Denote the reward function of $\widehat{\mathcal{M}}^{(k)}$ as constructed in Algorithm 1 by $\widehat{R}^{(k)} := R^{\widehat{\mathcal{M}}^{(k)}}$, and denote the true reward function of $\mathcal{P}$ by $R^{\mathcal{P}}$. By Lemma E.1, under the event $\mathcal{E}^1$, for any $h \in [H]$ and any trajectory $a_{1:h-1}, o_{2:h}$ for which $(a_{h-L:h-1}, o_{h-L+1:h}) \notin \mathcal{Z}_{h,\zeta}^{\text{low}}(\widehat{\pi}^h)$, it holds that $\widehat{R}^{(k)}(o_h) = R_h^{\mathcal{P}}(o_h)$. Thus, by the definition of the value function,

$$\left| V_1^{\pi,\mathcal{P},R^{\mathcal{P}}}(\emptyset) - V_1^{\pi,\mathcal{P},\widehat{R}^{(k)}}(\emptyset) \right|$$

$$= \left| \mathbb{E}_{(a_{1:H-1},o_{2:H})\sim\pi}^{\mathcal{P}} \left[ \sum_{h=2}^{H} R_h^{\mathcal{P}}(o_h) - \widehat{R}^{(k)}(o_h) \right] \right|$$

$$\leq \mathbb{E}_{(a_{1:H-1},o_{2:H})\sim\pi}^{\mathcal{P}} \left[ \sum_{h=2}^{H} \mathbb{1}[(a_{h-L:h-1}, o_{h-L+1:h}) \notin \mathcal{Z}_{h,\zeta}^{\text{low}}(\widehat{\pi}^h)] \right]$$

$$\leq \frac{HA^{2L}O^L\zeta}{\phi} + \sum_{h=2}^{H} d_{\mathsf{S},h-L}^{\mathcal{P},\pi}(\mathcal{U}_{\phi,h-L}^{\mathcal{P}}(\pi^h)) \qquad (66)$$

$$\leq \frac{HA^{2L}O^L\zeta}{\phi} + HS\delta \leq \alpha/8, \qquad (67)$$

where (66) uses Lemma F.4 for each $2 \leq h \leq H$, applied to the POMDP $\mathcal{P}$ with $\pi' = \widehat{\pi}^h$, and (67) uses (63) of Lemma I.4.

Combining (65) and (67) gives that, for all general policies $\pi$,

$$\left| V_1^{\pi,\mathcal{P},R^{\mathcal{P}}}(\emptyset) - V_1^{\pi,\widehat{\mathcal{M}}^{(k)},\widehat{R}^{(k)}}(\emptyset) \right| \leq \frac{\alpha}{4}. \qquad (68)$$

The above inequality holds in particular for the optimal policy $\pi^\star$ of $\mathcal{P}$. Since $\pi_\star^k$ (as computed in step 10 of BaSeCAMP) is the optimal policy of $\widehat{\mathcal{M}}^{(k)}$, we have (using (68))

$$V_1^{\pi_\star^k,\mathcal{P},R^{\mathcal{P}}}(\emptyset) \geq V_1^{\pi_\star^k,\widehat{\mathcal{M}}^{(k)},\widehat{R}^{(k)}}(\emptyset) - \frac{\alpha}{4}$$

$$\geq V_1^{\pi^\star,\widehat{\mathcal{M}}^{(k)},\widehat{R}^{(k)}}(\emptyset) - \frac{\alpha}{4}$$

$$\geq V_1^{\pi^\star,\mathcal{P},R^{\mathcal{P}}}(\emptyset) - \frac{\alpha}{2}.$$

Denote by $\mathcal{E}^2$ the event that, for all $k \in [K]$, the estimate $\widehat{r}^k$ produced in step 11 of Algorithm 3 is within $\alpha/4$ of the true value of $\pi_\star^k$, namely $V_1^{\pi_\star^k,\mathcal{P},R^{\mathcal{P}}}(\emptyset)$. By Hoeffding's inequality and the union bound, $\mathcal{E}^2$ occurs with probability at least $1 - \beta/2$. Thus, under the event $\mathcal{E}^1 \cap \mathcal{E}^2$ (which occurs with probability at least $1 - \beta$, the policy $\pi_\star^{k^\star}$ output by BaSeCAMP (Algorithm 3) satisfies

$$V_1^{\pi_\star^{k^\star},\mathcal{P},R^{\mathcal{P}}}(\emptyset) \geq V_1^{\pi^\star,\mathcal{P},R^{\mathcal{P}}}(\emptyset) - \alpha,$$

as desired.

Finally, we analyze the runtime of Algorithm 3. Let $T_1$ be the running time for a single call of ConstructMDP($L, N_0, N_1, \overline{\pi}^{k,1:H}$), and let $T_2$ be the running time for a single call of BarySpannerPolicy($\widehat{\mathcal{M}}^{(k)}, h$). Since it takes time $\tilde{O}(|\mathcal{Z}|^2 AH)$ to compute an optimal policy of $\widehat{\mathcal{M}}^{(k)}$ for each $k \in [K]$ and $K = 2SH \leq 2OH$, the total running time is bounded above by

$$2OH \cdot \left( T_1 + T_2 + \widetilde{O}(|\mathcal{Z}|^2 AH) + \widetilde{O}\left(\frac{H^2}{\alpha^2}\right) \right),$$

where $\widetilde{O}(\cdot)$ denotes that logarithmic factors (in $H, O, A, 1/\alpha$) are suppressed (above the $\widetilde{O}(\cdot)$ accounts for bit arithmetic in the algorithm's operations).

Next, from the definition of ConstructMDP (Algorithm 1), it is straightforward to see that

$$T_1 = \widetilde{O}\left( H^2 N_0 |\mathcal{Z}| AO \right).$$

Next we bound $T_2$. The linear optimization oracle $\mathcal{O}$ defined in step 3 of Algorithm 2 can be implemented in time $\widetilde{O}(|\mathcal{Z}| \cdot HO)$ using dynamic programming, since the problem $\arg\max_{\pi \in \Pi_{\mathcal{Z}}^{\mathrm{markov}}} \langle r, d_{\mathsf{O}, h-L}^{\pi, \widehat{\mathcal{M}}} \rangle$ is equivalent to finding an optimal policy in $\widehat{\mathcal{M}}$ when the reward function is given by $r$ at step $h - L$ and is 0 otherwise. Step 4 requires $O(O^2 \log O)$ calls to $\mathcal{O}$, meaning that a single run of BarySpannerPolicy($\widehat{\mathcal{M}}, h$) (Algorithm 2) takes time

$$T_2 = \widetilde{O}\left( O^3 H \cdot |\mathcal{Z}| \right). \tag{69}$$

Altogether, since $|\mathcal{Z}| = (AO)^L$, and using the definition of $N_0$ in Section C.1, the running time (and number of samples) taken by BaSeCAMP is $(AO)^{CL} \cdot \log(1/\beta)$, for some constant $C > 1$. □