# OpenReview forum: "Learning in Observable POMDPs, without Computationally Intractable Oracles"
_NeurIPS.cc/2022/Conference — NeurIPS 2022 Accept_

### Official Review · Reviewer_9ek8 · 2022-07-10

**Rating:** 7
**Confidence:** 4
**Soundness:** 4 excellent
**Presentation:** 3 good
**Contribution:** 4 excellent

**Summary:**

This paper proposes a novel algorithm for learning tabular POMDPs based on belief contraction and barycentric spanners.
Both its sample complexity and computational complexity scales exponentially w.r.t the observable parameter $1/\gamma$ and quasi-polynomially w.r.t $S,A,O,H$ and $1/\alpha$.



**Questions:**


1.  Given that poly sample complexity is achievable  [LCSJ22], do you think one can combine the techniques in this paper and  [LCSJ22] to obtain poly sample and quasi-poly computation algorithm?

2. The belief contraction result hints that a POMDP can be approximated by an MDP by viewing the most recent M-step observations-actions as states. So I am curious in stead of using the barycentric spanners, can we simply run UCRL or other tabular MDP algorithms on this MDP?

**Strengths And Weaknesses:**

This is a nice paper with interesting results and novel analysis. I enjoy reading it.

Strength

1. To my knowledge, this is the first quasi-polynomial time algorithm for learning tabular MDPs under reasonable assumptions.

2. The application of barycentric spanners is novel and the analysis is nontrivial.


Weakness

1. The sample complexity is exponential in $1/\gamma$ and quasi-polynomially in all other parameters. If I understand correctly, the algorithms in [LCSJ22] only require a polynomial number of samples to learn a near-optimal policy under the same observable condition. (Nonetheless, the algorithms in [LCSJ22]  seem to require exponential time to implement as pointed out in this work. ) I believe these differences should be pointed out explicitly in the main paper.

2. The observable assumption can never be satisfied when the number of states is larger than the number of observations. Any comments on how to address this limitation and generalize the results in this paper?


 [LCSJ22] Qinghua Liu, Alan Chung, Csaba Szepesvári, and Chi Jin. When is partially observable reinforcement learning not scary? arXiv preprint arXiv:2204.08967, 2022.

---

> ### Author Response · Authors · 2022-08-02
> **Response to reviewer 9ek8**
>
> We thank the reviewer for their time and insightful questions. We respond to the questions below:
>
> Extending observability to the overcomplete setting (more states than observations), perhaps via a multi-step notion of observability, is an interesting open problem. The recently-introduced ``weakly-revealing'' condition which enables sample-efficient learning for POMDPs [LCSJ '22] is very closely related, but there seem to be technical obstacles even in achieving computationally efficient planning under such a condition.
>
> Regarding achieving polynomial sample complexity (as in [LCSJ'22]) and quasipolynomial time: this is a very interesting open problem, and we do not have a strong belief as to whether it is possible or not. The techniques in our work seem stuck at quasipolynomial sample complexity, essentially since our analysis is via the approximate MDP, which has a quasipolynomial number of unknown parameters. In contrast, recent works achieving polynomial sample complexity, such as [LCSJ'22], use approaches which have only a polynomial number of parameters. These works have the disadvantage of needing to solve non-convex optimization problems, for which efficient algorithms are not known (e.g., in [LCSJ'22], it is an optimistic maximum likelhood problem).
>
>
>
> Regarding applying tabular MDP learning to the approximate MDP: the problem with that approach is the *approximation (misspecification) error* $\epsilon$. Learning under misspecification is a notorious challenge in RL, and the misspecification bound $\epsilon$ tends to show up in the suboptimality guarantee multiplied by an additional factor polynomial in the size/dimension of the approximating MDP (e.g. Theorem 4 of [JKALS '17] ``Contextual Decision Processes with low Bellman rank are PAC-Learnable''). In our setting, this factor leads to vacuous guarantees. We remark that tabular approaches such as UCRL would work if the closeness of the POMDP to an MDP held *uniformly* over all states of the MDP, but such a uniform bound on the approximation error is weaker than the notion of approximation we have (Eq. 3). Please see Appendix A in the supplementary material (specifically line 662 onwards) for a more detailed discussion of this issue.

---

> > ### Comment · Reviewer_9ek8 · 2022-08-08
> > **Re author response**
> >
> > Thanks for the detailed response! This is a nice paper and I will continue supporting its acceptance.

---

### Official Review · Reviewer_C7us · 2022-07-11

**Rating:** 7
**Confidence:** 4
**Soundness:** 4 excellent
**Presentation:** 3 good
**Contribution:** 3 good

**Summary:**

The paper proposes a quasi-polynomial-time algorithm for learning in a class of partially observable Markov decision processes (POMDPs). It builds on the recent set of results that noticed the reductions in the computational complexity of planning in POMDPs that comply with an assumption on the observability in the POMDP.

**Questions:**

* When is it easy to check the underlying assumptions?
* How much does the result rely on the complexity reduction in planning under the observability assumption and how much of it is specific to the learning setting? A discussion cannot hurt.

**Limitations:**

There is no explicit discussion of the limitations in the paper or an empirical study that would help the reader establish an intuition on the implications and limitations of the results.

**Strengths And Weaknesses:**

Strengths:
* The paper tackles a difficult problem and offers a novel perspective.
* The direction of understanding the implications of imposing practically viable assumptions on POMDPs on planning and learning is an emerging one. It makes sense to investigate problems in this direction.
* It is a relatively dense yet sufficiently clearly written paper.

Weaknesses:
* While I am sure the authors will claim that the paper is a theoretical one, a paper that argues to be not computationally intractable should provide empirical evidence of the implied traceability.
* It relies on assumptions similar to those in the recent literature. Nevertheless, it is necessary to provide additional and convincing evidence that the assumptions can be checked in practice and they will hold for interesting and large classes of problem instances. The lack of empirical evidence as mentioned in the previous bullet point hurts here as well.
* While with a different perspective, there is additional recent work that aims to cope with the computational impracticality of planning in POMDPs: https://arxiv.org/abs/2009.11459, https://arxiv.org/abs/2204.00755, and https://arxiv.org/abs/1710.10294. Contrast with this work would help and may broaden the interest in the current paper.

---

> ### Author Response · Authors · 2022-08-02
> **Response to reviewer C7us**
>
> We thank the reviewer for their time and for the interesting references. We will incorporate the references in our discussion of prior work on heuristics for planning and learning POMDPs.
>
> While we do not implement our algorithm in this paper, the primary distinction between our work and previous theoretical results on learning POMDPs (with provable guarantees) is that our algorithm *could* feasibly be implemented (in quasipolynomial time), whereas prior algorithms invoke hopelessly intractable oracles with no recipes for (even heuristically) implementing them.
>
> We believe that it is important to not settle for such a wide gap between theory and practice, where there are heuristics that sometimes work well in practice, but the only known algorithms with provable guarantees, prior to our work, either needed unrealistic assumptions or required exponential time. From that vantage point, getting quasipolynomial time is a significant milestone (moreover, lower bounds in the prior work show that our result cannot be substantially improved).
>
> To answer the reviewer's two questions:
>
> • Regarding checking the underlying assumptions, i.e., certification of $\gamma$-observability (e.g. given an observation matrix): good question; this is an interesting open problem.
>
> • The complexity reduction allowed by the observability assumption is *necessary* to achieve a computationally efficient algorithm (since learning is at least as hard as planning). However, it is in no way *sufficient*; an efficient planning algorithm does not directly yield an efficient learning algorithm. Since every observable POMDP has a quasipolynomial-sized ``approximate MDP", it's natural to hope that standard techniques for learning tabular MDPs (such as UCRL), or known techniques for learning misspecified MDPs, could be applicable. However, there are major technical obstacles to those approaches. See Appendix A of the supplementary material (line 662 onwards) as well as the last paragraph of our response to Reviewer 9ek8 for more discussion.

---

### Official Review · Reviewer_kngr · 2022-07-12

**Rating:** 7
**Confidence:** 3
**Soundness:** 3 good
**Presentation:** 3 good
**Contribution:** 3 good

**Summary:**

The paper introduces a method for approximately solving unknown POMDPs. The method relies on constructing an MDP approximation and a set of policies that incrementally improve coverage of that MDP's state space, thereby increasing the accuracy of the MDP approximation. The approach makes use of a novel exploration objective that uses the concept of barycentric spanners to ensure that every possible observation distribution can be represented as a linear combination of a small "core" set of observation distributions. The resulting algorithm has approximately optimal time complexity and quasi-polynomial sample complexity.

**Questions:**

(See numbered items above.)

**Strengths And Weaknesses:**

**Strengths:**
- The paper covers a lot of difficult theoretical ground and communicates the high-level ideas very clearly.
- The theoretical results appear to be correct, and the supplementary materials appear to be detailed enough that one could verify the claims presented in the main text, although I was only able to check them at a high level.
- The section "Overview of results" is extremely helpful for communicating the high-level ideas quickly, and for providing a roadmap of the remainder of the paper.

**Weaknesses:**

1. Lines 135-141: I'm a little worried about the way the policies are defined here. It's fine to say a general policy is a distribution over deterministic policies, but when we execute that policy, we sample a deterministic policy from it, and follow that deterministic policy for the entire episode. My concern is: does this scheme necessarily include the optimal policy for the POMDP? Are all trajectories that can be achieved with a fully stochastic policy still supported under the resulting distribution? Is this construction allowed because the policy is conditioned on the entire history?

2. I'm confused about a few things:

    a. Line 46: "A necessary first step towards solving the learning problem is having a computationally efficient planning algorithm." -> Is there a reason why a model-free approach wouldn't work?

    b. Line 120: Why is reward a function of observations $O$? I'm used to seeing it depend on states $S$.

    c. Lines 280-282: "the formal latent state distribution induced by $\pi$ is a linear combination of the formal latent state distributions with the same coefficients" -> Why is this the case?

3. In a few of the definitions and assumptions, I would have preferred to have more detail. However, I appreciate that the paper is quite short on space, so I understand that many sections have been significantly compressed already.

    a. Assumption 1.1 defines the distance between two distributions as the sum of the differences in probability for each outcome. Why use this particular definition for $\gamma$-observability?

    b. In section 2.2, while I appreciate that the notation is presented up front, it would be helpful to see an example for why we need to define the probability of an event $\mathbb{P}^\mathcal{P}_\pi(\mathcal{E})$.

    c. In section 2.4, the presentation of the product space $\mathcal{Z}$ is confusing. It took until much later for me to realize that we need this because it's the form that the POMDP -> MDP conversion requires.

    d. In Algorithm 3, lines 3-4, there are multiple definitions of $k$, which is confusing.

    e. In line 331, I could not find a definition for $e_s$ anywhere. I assumed this was a unit/one-hot vector, so the $\langle e_i, \cdot \rangle$ is equivalent to indexing the second term at position $i$. It would be helpful to have this notation explicitly defined, even if it's just in the appendix.

-----

**Review Summary:**

This paper seems useful for several reasons. First, it introduces an algorithm for learning in POMDPs that achieves optimal time complexity and quasi-polynomial sample complexity. Second, it presents a novel exploration scheme that relies on the concept of barycentric spanners, rather than the ubiquitous "optimism under uncertainty" principle. While I can only verify the correctness at a high level, the material is presented clearly and the high-level ideas are very approachable. Unfortunately, space limitations preclude a more detailed treatment in the main text, but the supplementary materials appear to more than make up for it. Accept.

---

> ### Author Response · Authors · 2022-08-02
> **Response to reviewer kngr**
>
> We thank the reviewer for their time and helpful comments. To address their questions:
>
> (1.) Any POMDP is equivalent to an (exponential-sized) MDP on the space of histories. Thus, the fact that the optimal policy for a POMDP is (without loss of generality) a deterministic but history-dependent policy follows from the fact that any MDP has an optimal policy that is deterministic. Furthermore, all trajectories that can be achieved by a fully stochastic policy can also be achieved by a distribution over deterministic policies, namely the product distribution over the stochastic policy's distribution at each history.
>
> (2a.) Certainly, a model-free approach could work. We meant this statement in a complexity-theoretic sense: planning reduces to learning (by simulating trajectories), so if there exists an efficient learning algorithm, there must exist an efficient planning algorithm. Thus, before attempting to construct a computationally efficient learning algorithm for a class of POMDPs, it is necessary to ensure that this class does not suffer from a hardness result for efficient planning.
>
> (2b.) It is standard in the literature on learning in POMDPs for the rewards to be observed. Thus, to avoid the notational clutter of a two-part observation (the ``observation'' and the reward), we simply denote the reward as a function of the observation. Note that since the latent state is not observed, we cannot make the reward a function of the latent state.
>
> (2c.) We're saying that this linear combination (of formal latent state distributions) has the same coefficients as the linear combination on line 279 (of observation distributions). The reason is that each formal latent state distribution is simply the product of the matrix $\mathbb{O}^{\dagger}_{h-L}$ with the corresponding observation distribution.
>
> (3a.) This metric is the total variation distance (up to a factor of $2$). We define $\gamma$-observability in this way because it is an intuitive notion (least $\ell_1$ singular value), and moreover it is currently the weakest assumption known to circumvent the computational intractability of planning. In contrast, the related $\ell_2$ definition (i.e. least $\ell_2$ singular value), which is used in prior works on sample-efficient (but computationally inefficient) learning, is not currently known to enable efficient planning. Understanding whether planning with a lower bound on the least $\ell_2$ singular value is computationally tractable is an interesting open problem.
>
> (3b.) See, e.g., the left hand side of Eq. 2 (line 224): here the event $\mathcal{E}$ is the event that the observation at step $h+1$ is $o_{h+1}$. Eq. 2 expresses the probability under a policy $\pi$ that the next observation is $o_{h+1}$ given the entire history as a linear transformation of the belief state at step $h$.
>
> (3cde.) Thanks for catching these omissions/typos. We'll add clarification for the purpose of the product space $\mathcal{Z}$ and the definition of $e_s$ (which is indeed the unit vector). Line 4 of Algorithm 3 should say $\bar{\pi}^{k,h} = \frac{1}{k}\sum_{i=1}^k \pi^{i,h}$.

---

> > ### Comment · Reviewer_kngr · 2022-08-03
> > **Response to authors**
> >
> > (1.) Okay, great, that makes sense. Thanks for that explanation.
> >
> > (2a.) Ah, okay. I interpreted it as a prescriptive statement about how to learn. I suppose I should have interpreted it as "unless there exists a computationally efficient planning algorithm, there cannot exist a computationally efficient learning algorithm". In other words, learning may not *require* planning, but the problem of learning is at least as hard as the problem of planning.
> >
> > (2.b.) I agree that rewards are typically observed, but my comment is about how rewards are *defined*. I would expect the underlying transition dynamics and rewards to depend on the unobserved Markov state, but then for those rewards to be fully observable once realized. The argument that "since the latent state is not observed, we cannot make the reward a function of the latent state" feels backwards. By that logic, we can't make the observations depend on the latent state either, when clearly they do. My question is: do we lose anything by defining rewards this way? For one thing, we can no longer express POMDPs where the rewards disambiguate between identical observations that come from two different underlying states.
> >
> > (2.c.) I see... We don't have access to $\mathbb{O}^{\dagger}_{h-L}$, but nevertheless it exists(?), so we can use it to get the linear combination over latent state distributions.
> >
> > (3.) Thanks for the clarifications. It would be helpful to include this justification of $\gamma$-observability in the main text.

---

> > > ### Author Response · Authors · 2022-08-03
> > > **Response about (2b)**
> > >
> > > We don't lose anything by defining rewards as a function of the observation, because there is a simple reduction from the general case. To be more precise:
> > >
> > > For any POMDP where the observations are a stochastic function of the latent states $s \mapsto o$, and the (observed) rewards are a stochastic function of the latent states $s \mapsto r$, we can (1) discretize the rewards to precision $\epsilon / H$, and then (2) define a new POMDP with observations $(o, r)$, and the rewards are a deterministic function of the observations $(o,r) \mapsto r$. This POMDP is $\epsilon$-equivalent to the original POMDP (that is, the value of any policy changes by at most $\epsilon$), and the size of the POMDP is only increased multiplicatively by a factor of $H/\epsilon$.
> > >
> > > Now suppose we are interacting with the original POMDP and want to learn a near-optimal policy. We can simulate interaction with the new POMDP by discretizing the observed rewards and then treating the observation as the pair $(o,r)$. The rewards are now a known function of the observations (i.e. a projection), so we are in exactly the setting which we describe in the paper. Also, the $\gamma$-observability constant of the new POMDP is no worse than that of the original POMDP, and can be better (e.g. in the situation you describe, where the reward disambiguates). Does this help answer your question?

---

> > > > ### Comment · Reviewer_kngr · 2022-08-06
> > > > **Response to authors**
> > > >
> > > > Yes, that's very helpful, thanks. I'm a little nervous that the discretization will multiply the observation space by the number of reward bins, but I suppose there will be many cases where the number of reward bins is much less than the size of the observation space. For example, if the reward is sparse (i.e. $r \in \\{0, 1\\}$), then you only need 2 bins, and the bounds are essentially unchanged. In any case, I don't think your reward formulation is necessarily a drawback---I was just curious why you defined things that way.

---

### Meta-Review · Area_Chair_pxAg · 2022-08-31

**Recommendation:** Accept
**Confidence:** Certain

**Metareview:**

This paper provides the first quasi-polynomial time algorithm for learning a large class of POMDPs. Part of the technics is based on a prior paper on developing quasi-polynomial time algorithm for planning in POMDPs, which achieved a subset of the tasks in this paper. Given all prior algorithms for learning POMDPs requires an exponential amount of time, we believe the contribution is significant and worth the acceptance to NeurIPS.

**Award:**

No

---

### Decision · Program_Chairs · 2022-09-14

Accept